# Flexible Language Modeling in Continuous Space with Transformer-based Autoregressive Flows

**Ruixiang Zhang, Shuangfei Zhai, Jiatao Gu, Yizhe Zhang, Huangjie Zheng, Tianrong Chen**
**Miguel Angel Bautista, Josh Susskind, Navdeep Jaitly**

Apple

`{ruixiangz, szhai, njaitly}@apple.com`

## Abstract

Autoregressive models have driven remarkable progress in language modeling. Their foundational reliance on discrete tokens, unidirectional context, and single-pass decoding, while central to their success, also inspires the exploration of a design space that could offer new axes of modeling flexibility. In this work, we explore an alternative paradigm, shifting language modeling from a discrete token space to a continuous latent space. We propose a novel framework `TarFlowLM`, that employs transformer-based autoregressive normalizing flows [73] to model these continuous representations. This approach unlocks substantial flexibility, enabling the construction of models that can capture global bi-directional context through stacked, alternating-direction autoregressive transformations, support block-wise generation with flexible token patch sizes, and facilitate a hierarchical multi-pass generation process. We further propose new mixture-based coupling transformations designed to capture complex dependencies within the latent space shaped by discrete data, and demonstrate theoretical connections to conventional discrete autoregressive models. Extensive experiments on language modeling benchmarks demonstrate strong likelihood performance and highlight the flexible modeling capabilities inherent in our framework.

## 1 Introduction

Transformer-based autoregressive models [5, 66] have emerged as the dominant paradigm for language modeling, achieving remarkable performance by predicting discrete tokens one at a time under the cross-entropy objective. By scaling both model size and training data, these models excel at next-token prediction and have become the foundation for modern natural language generation systems.

Their remarkable success provides a strong foundation and inspires further exploration into the **design space of autoregressive sequence modelling**. While the established paradigm of discrete, typically unidirectional autoregressive generation offers a powerful and well-understood framework, we consider whether alternative formulations might offer new dimensions of flexibility and open different avenues for model construction. This work investigates such a possibility: *What if autoregressive language modeling were reformulated within a continuous latent space?* Moving to continuous representations allows for the preservation of the sequential factorization familiar from autoregressive methods, while potentially unlocking new modeling capabilities.

This adoption of continuous latent spaces is a direction also pursued by other generative frameworks; for instance, diffusion models have demonstrated strong capabilities in this domain. Our investigation, however, distinguishes itself by concentrating on autoregressive normalizing flows for modeling the joint distribution of these continuous sequences. We pursue this specific avenue because normalizing

39th Conference on Neural Information Processing Systems (NeurIPS 2025).

flows offer expressive power for density estimation, and critically, their intrinsic sequential processing aligns them closely with the fundamental mechanisms of discrete autoregressive language models. This alignment provides a unique vantage point: it allows us to explore how the core principles of autoregressive modeling can be evolved and potentially augmented when transitioned into a continuous, learnable transformation framework, thereby seeking to extend and enrich this successful paradigm.

Building on this perspective, we propose a novel framework that employs Transformer-based autoregressive normalizing flows [73] to model these continuous latent representations. This shift from discrete token space to a continuous latent space is key to unlocking substantial modeling flexibility. Our formulation facilitates the construction of models capable of capturing global bi-directional context through stacked, alternating-direction autoregressive transformations. It also supports blockwise generation with adaptable token patch sizes and enables a hierarchical multi-pass generation process, allowing for the observation and influence of sequence formation through intermediate stages. These capabilities arise naturally from the continuous and invertible nature of the flow-based transformations.

To this end, we present `TarFlowLM`, a Transformer-based Auto-Regressive Flow Language Model that uses mixture-based coupling transformations to effectively model the complex, multi-modal distributions created when mapping discrete data to a continuous latent space. Our key theoretical contribution is establishing the equivalence that transforms these mixture distributions into exact, invertible normalizing flow layers. Specifically, we show that a single-dimensional Mixture of Gaussians (MoG) distribution can be realized as a 1D Mixture-CDF flow, and a multi-dimensional MoG distribution as a Mixture-Rosenblatt flow. We also draw theoretical links between our continuous approach and standard discrete autoregressive models. Experiments on language modeling benchmarks show that our method achieves strong likelihood performance and, importantly, enables greater modeling flexibility. Our work expands the possibilities of language modeling by extending autoregressive methods into the continuous domain.

## 2  Background

Normalizing flows offer a method for constructing flexible probability distributions over continuous variables [51]. The core idea is to start with a simple base distribution $p_{\text{base}}(\mathbf{u})$ defined on $\mathbb{R}^d$, often a standard Gaussian, and transform samples $\mathbf{u} \sim p_{\text{base}}(\mathbf{u})$ using an invertible and differentiable function $f : \mathbb{R}^d \to \mathbb{R}^d$, known as a diffeomorphism. This process yields a variable $\mathbf{z} = f^{-1}(\mathbf{u})$ that follows a potentially much more complex distribution $p(\mathbf{z})$. A key advantage of this approach is that the probability density function $p(\mathbf{z})$ can be computed exactly using the change of variables formula:

$$\log p(\mathbf{z}) = \log p_{\text{base}}(f(\mathbf{z})) + \log |\det J_f(\mathbf{z})| \tag{1}$$

Here, $J_f(\mathbf{z})$ is the Jacobian matrix of the transformation $f$ evaluated at $\mathbf{z}$, and $|\det(\cdot)|$ denotes the absolute value of the determinant. Complex distributions are typically modeled by composing multiple simple transformations $f = f^{(L)} \circ \cdots \circ f^{(1)}$, where each layer $f^{(\ell)}$ is designed to be easily invertible and have a tractable Jacobian determinant. The log-density then becomes a sum of log-determinant terms from each layer, plus the log-density of the final transformed variable under the base distribution:

$$\log p(\mathbf{z}) = \log p_{\text{base}}(\mathbf{u}) + \sum_{\ell=1}^{L} \log \left| \det J_{f^{(\ell)}}(\mathbf{h}^{(\ell-1)}) \right| \tag{2}$$

where $\mathbf{h}^{(0)} = \mathbf{z}$, $\mathbf{h}^{(\ell)} = f^{(\ell)}(\mathbf{h}^{(\ell-1)})$, and $\mathbf{u} = \mathbf{h}^{(L)}$.

The main challenge in designing normalizing flows lies in choosing transformations $f^{(\ell)}$ that are both expressive enough to model complex data distributions and computationally efficient, particularly regarding the calculation of the inverse $f^{-1}$ and the Jacobian determinant $\det J_f$.

A prominent and widely used class is autoregressive flows [39, 50]. These structure the transformation $f$ such that $u_i$ depends on $z_j$ with $j < i$. This ensures the Jacobian $J_f$ is triangular, making its determinant the product of diagonal entries, computable in $\mathcal{O}(d)$ time. A common choice for the element-wise transformation is the affine function: $z_i = \alpha_i u_i + \beta_i$. In the autoregressive setting, the scale $\alpha_i > 0$ and shift $\beta_i$ for dimension $i$ are functions of the preceding dimensions, e.g.,

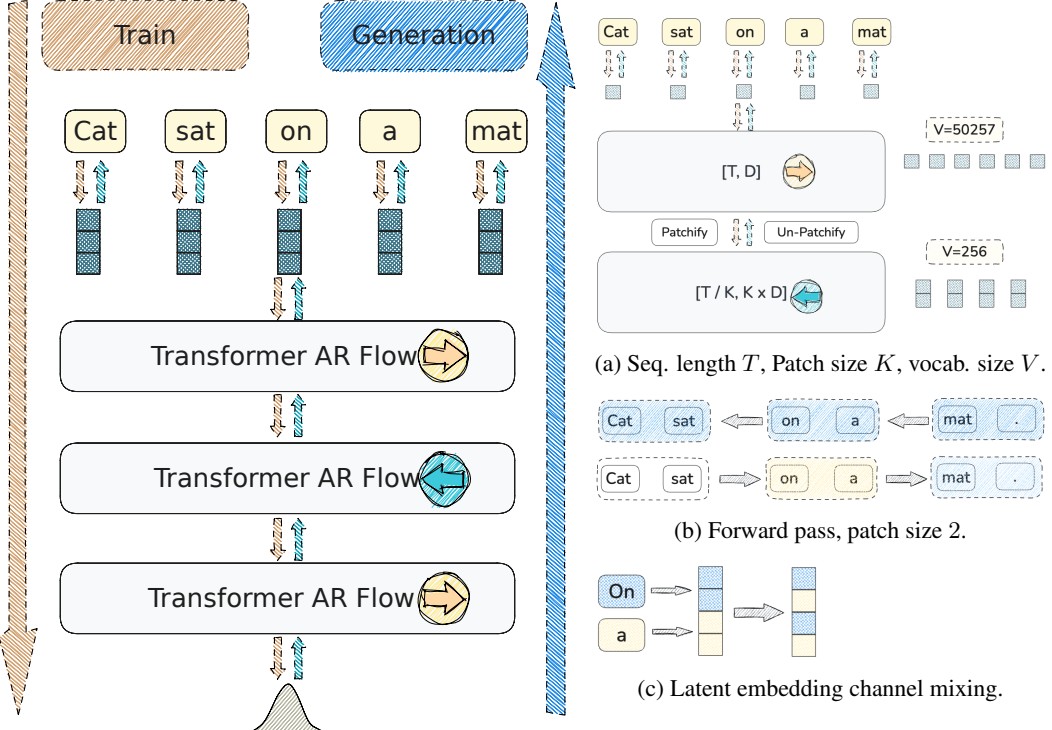

(a) Seq. length $T$, Patch size $K$, vocab. size $V$.

(b) Forward pass, patch size 2.

(c) Latent embedding channel mixing.

Figure 1: Our approach transforms discrete token sequences into a continuous latent space modeled by stacked Transformer-based autoregressive normalizing flows, creating a fully invertible pipeline for training and generation. **Left:** During training (Red top-down arrow), tokens are encoded into Gaussian latents by a VAE encoder, transformed through flow layers into standard Gaussian variables, and reconstructed by a tied decoder. At inference (Blue bottom-up arrow), samples from a standard Gaussian are inverted through the flows and decoded into text. **Right:** Key flexibility features: (a) Tokens can be grouped into patches and embedded into latent space using mixture-of-Gaussians transformations with dynamic vocabulary sizes; (b) Block-wise autoregressive flows alternate directions, enabling context exchange between initially independent tokens; (c) Channel mixing permutes latent dimensions between flows to facilitate cross-token information sharing.

$\alpha_i = \alpha_i(\mathbf{z}_{<i})$ and $\beta_i = \beta_i(\mathbf{z}_{<i})$. If the base variables $u_i$ are drawn from a standard Gaussian $\mathcal{N}(0, 1)$, this affine transformation implies that each $z_i$ conditioned on the context $\mathbf{z}_{<i}$ follows a Gaussian distribution $\mathcal{N}(z_i; \mu_i = \beta_i(\mathbf{z}_{<i}), \sigma_i^2 = \alpha_i(\mathbf{z}_{<i})^2)$. To ensure all variables influence each other, multiple autoregressive layers are typically stacked, often with variable permutations between them.

## 3 Autoregressive Language Modeling in Continuous Latent Space

**Notation.** We use italics for scalars ($x_t$), bold for vectors ($\mathbf{z}_t$), and bold with indices for sequences ($\mathbf{x}_{1:T}$, $\mathbf{z}_{1:T}$).

We propose utilizing the Variational Autoencoder (VAE) framework [38]. The core idea is to map sequences of discrete tokens $\mathbf{x}_{1:T} = (x_1, \ldots, x_T)$ into sequences of continuous latent variables $\mathbf{z}_{1:T} = (\mathbf{z}_1, \ldots, \mathbf{z}_T)$, where each $\mathbf{z}_t \in \mathbb{R}^d$. We then model the joint distribution $p(\mathbf{z}_{1:T})$, the prior, to capture sequential dependencies. This approach aims to preserve the sequential factorization structure familiar from autoregressive models while gaining the benefits of end-to-end differentiability inherent in continuous spaces.

Our model consists of three components (see Fig. 1): an encoder $q(\mathbf{z}_{1:T}|\mathbf{x}_{1:T})$ and decoder $p(\mathbf{x}_{1:T}|\mathbf{z}_{1:T})$ that map between discrete token sequences and continuous latent sequences, and

Table 1: Comparison of 1D and $d$-D Mixture of Gaussians (MoG) normalizing flow layers, transforming input $z/\mathbf{z}$ to standard Gaussian $u/\mathbf{u}$. We use $\mathcal{C}$ to denote the conditional context.

| | 1-D Mixture CDF Layer | $d$-D Mixture Rosenblatt Layer |
|---|---|---|
| Input | Scalar $z \in \mathbb{R}$ | Vector $\mathbf{z} \in \mathbb{R}^d$ |
| Output | Scalar $u \in \mathbb{R}$ | Vector $\mathbf{u} \in \mathbb{R}^d$ |
| Base PDF ($p_{\text{base}}$) | $\mathcal{N}(u; 0, 1)$ | $\mathcal{N}(\mathbf{u}; \mathbf{0}, \boldsymbol{I}_d)$ |
| Input PDF ($p(\cdot\|\mathcal{C})$) | $p_{\text{mix-1}}(z\|\mathcal{C}) = \sum_j \pi_j \mathcal{N}(z; m_j, s_j^2)$ | $p_{\text{mix-}d}(\mathbf{z}\|\mathcal{C}) = \sum_j \pi_j \mathcal{N}(\mathbf{z}; \mathbf{m}_j, s_j^2 \boldsymbol{I}_d)$ |
| Forward Map ($f$) | $u = \Phi^{-1}(F_{\text{mix}}(z; \mathcal{C}))$ | $u_i = \Phi^{-1}(F_i(z_i\|\mathcal{C}, \mathbf{z}_{<i}))$, for $i = 1..d$ |
| | $F_{\text{mix}}$: CDF of $p(z\|\mathcal{C})$ | (Rosenblatt, see Alg. 1) |
| Inverse Map ($f^{-1}$) | $z = F_{\text{mix}}^{-1}(\Phi(u); \mathcal{C})$ | Solve $F_i(z_i\| \dots) = \Phi(u_i)$ for $z_i$, for $i = 1..d$ |
| $\log \|\det J_f\|$ | $\log p_{\text{mix-1}}(z\|\mathcal{C}) - \log p_{\text{base}}(u)$ | $\log p_{\text{mix-}d}(\mathbf{z}\|\mathcal{C}) - \log p_{\text{base}}(\mathbf{u})$ |

an autoregressive prior $p(\mathbf{z}_{1:T})$ over the latent space. The model parameters (covering encoder, decoder, and prior) are learned by maximizing the Evidence Lower Bound (ELBO) on the data log-likelihood $\log p(\mathbf{x}_{1:T})$:

$$\mathcal{L}(\mathbf{x}_{1:T}) = \mathbb{E}_{\mathbf{z}_{1:T} \sim q(\cdot|\mathbf{x}_{1:T})} \underbrace{\log p(\mathbf{x}_{1:T}|\mathbf{z}_{1:T})}_{\text{decoder}} + \underbrace{\log p(\mathbf{z}_{1:T})}_{\text{prior}} - \underbrace{\log q(\mathbf{z}_{1:T}|\mathbf{x}_{1:T})}_{\text{encoder}} \tag{3}$$

### 3.1 The Bridge: Encoder and Decoder

The encoder and decoder facilitate the transition between the discrete data space of tokens and the continuous latent space of vectors $\mathbf{z}_t$.

**Encoder** $q(\mathbf{z}_{1:T}|\mathbf{x}_{1:T})$. We use a factorized encoder, $q(\mathbf{z}_{1:T}|\mathbf{x}_{1:T}) = \prod_{t=1}^{T} q(\mathbf{z}_t|x_t)$, where each token $k \in \{1, \dots, V\}$ is assigned a learnable mean $\boldsymbol{\mu}_k \in \mathbb{R}^d$ and variance $\sigma_k^2$ for an isotropic Gaussian: $q(\mathbf{z}_t|x_t = k) = \mathcal{N}(\mathbf{z}_t; \boldsymbol{\mu}_k, \sigma_k^2 \boldsymbol{I}) \equiv \mathcal{N}_k(\mathbf{z}_t)$. Thus, the codebook $\{\boldsymbol{\mu}_k, \sigma_k^2\}_{k=1}^{V}$ maps each token to a Gaussian distribution in latent space.

**Decoder** $p(\mathbf{x}_{1:T}|\mathbf{z}_{1:T})$. The decoder maps each latent vector $\mathbf{z}_t$ back to a distribution over tokens, with again a factorized structure $p(\mathbf{x}_{1:T}|\mathbf{z}_{1:T}) = \prod_{t=1}^{T} p(x_t|\mathbf{z}_t)$. We reuse the *encoder*'s Gaussian parameters $\{\boldsymbol{\mu}_k, \sigma_k^2\}_{k=1}^{V}$ for the decoder with a Bayesian posterior parameterization assuming an uniform prior: $p(x_t = k|\mathbf{z}_t) = \frac{p(x_t=k)q(\mathbf{z}_t|x_t=k)}{\sum_{j=1}^{V} p(x_t=j)q(\mathbf{z}_t|x_t=j)} = \frac{\mathcal{N}_k(\mathbf{z}_t)}{\sum_{j=1}^{V} \mathcal{N}_j(\mathbf{z}_t)}$. In other words, the probability of decoding token $k$ from $\mathbf{z}_t$ is determined by how likely $\mathbf{z}_t$ is under the $k$-th encoder Gaussian, compared to all other tokens.

### 3.2 Autoregressive Prior Modeling for $p(\mathbf{z}_{1:T})$

To model the prior $p(\mathbf{z}_{1:T})$ over latent sequences, we use the standard autoregressive factorization: $p(\mathbf{z}_{1:T}) = \prod_{t=1}^{T} p(\mathbf{z}_t|\mathbf{z}_{<t})$, where $\mathbf{z}_{<t} = (\mathbf{z}_1, \dots, \mathbf{z}_{t-1})$ is the history. The main task is to define the conditional distribution $p(\mathbf{z}_t|\mathbf{z}_{<t})$. We present two forms of parameterizing the conditional $p(\mathbf{z}_t|\mathbf{z}_{<t})$, one with dimension-wise autoregressive factorization Sec. 3.2.1, the other with token-wise autoregressive factorization Sec. 3.2.2. In each formulation, we use a mixture-based probability distribution, and show that we can equivalently convert the mixture probability density into an equivalent invertible normalizing flow layer. Finally, we show how stacking such flow layers yields an expressive and flexible prior model $p(\mathbf{z}_{1:T})$.

#### 3.2.1 Dimension-wise Autoregressive 1-D Mixture Modeling

We start from decomposing the conditional $p(\mathbf{z}_t|\mathbf{z}_{<t})$ using the chain rule across the dimensions of $\mathbf{z}_t$:

$$p(\mathbf{z}_t|\mathbf{z}_{<t}) = \prod_{i=1}^{d} p(z_{t,i}|\mathbf{z}_{<t}, \mathbf{z}_{t,<i}) \tag{4}$$

Here, $z_{t,i}$ is the $i$-th scalar component of $\mathbf{z}_t$. $\mathbf{z}_{t,<i} = (z_{t,1}, \dots, z_{t,i-1})$ are its preceding components. We parameterize each scalar conditional density $p(z_{t,i}|\mathbf{z}_{<t}, \mathbf{z}_{t,<i})$ as a mixture of $V$ 1-dimensional

| | **TEXT8 (Test BPC ↓)** | | | **OpenWebText (Validation Perplexity ↓)** | |
|---|---|---|---|---|---|
| Space | Type | Method | BPC | Method | Perplexity |
| C | Diffusion | Plaid [27] | ≤ 1.48 | Gaussian Diffusion | ≤ 27.28 |
| C | Diffusion | BFN [25] | ≤ 1.41 | — | |
| D | AR | MAC [59] | ≤ 1.40 | — | |
| D | AR | Transformer AR [1] | **1.23** | Transformer AR | **17.54** |
| D | Diffusion | SEDD Absorb [45] | ≤ 1.39 | SEDD Absorb | ≤ 24.10 |
| D | Diffusion | MD4 [58] | ≤ 1.37 | MD4 [58] | ≤ 22.13 |
| D | Diffusion | TCSM [77] | ≤ 1.25 | GenMD4 [58] | ≤ 21.80 |
| D | Diffusion | EDLM [70] | ≤ 1.24 | MDLM [56] | ≤ 23.21 |
| C | NF | Latent NF [82] | ≤ 1.61 | — | |
| C | NF | CNF [41] | ≤ 1.45 | — | |
| C | NF | Argmax Flow [31] | ≤ 1.45 | — | |
| C | NF | TarFlowLM Affine | ≤ 1.54 | TarFlowLM Affine | ≤ 148.21 |
| C | NF | TarFlowLM Mix-1 | ≤ 1.37 | TarFlowLM Mix-1 | ≤ 27.11 |
| C | NF | TarFlowLM Mix-d | ≤ 1.30 | TarFlowLM Mix-d | ≤ 22.64 |

Table 2: Performance comparison across different model types and datasets. C=Continuous space, D=Discrete space, AR=Autoregressive, NF=Normalizing Flows.

Gaussian distributions. The parameter bundle $\left[\boldsymbol{\pi}_{t,i}, \mathbf{m}_{t,i}, \boldsymbol{\sigma}^2_{t,i}\right]$ is predicted by a causal Transformer-based model from the context $(\mathbf{z}_{<t}, \mathbf{z}_{t,<i})$. The conditional density is then, dropping the subscripts $t, i$ here for notational brevity:

$$p(z_{t,i}|\mathbf{z}_{<t}, \mathbf{z}_{t,<i}) = \sum_{k=1}^{V} \boldsymbol{\pi}[k]\mathcal{N}(z_{t,i}; \mathbf{m}[k], \boldsymbol{\sigma}^2[k]), \quad \left[\boldsymbol{\pi}, \mathbf{m}, \boldsymbol{\sigma}^2\right] = \text{Transformer}(\mathbf{z}_{<t}, \mathbf{z}_{t,<i}) \quad (5)$$

We denote this 1-D mixture PDF as $p_{\text{mix-1}}(z_{t,i})$. We discuss the connection to the mixture-of-logistics coupling [29] in Sec. D.

**Transforming 1-D Mixture Density to an Invertible Flow.** The conditional 1D mixture density $p_{\text{mix-1}}(\cdot)$ from Eq. (5) naturally defines an invertible normalizing flow layer. This layer transforms an input $z_{t,i}$ into $u_{t,i}$, such that $u_{t,i}$ follows a standard normal distribution $\mathcal{N}(0, 1)$. The transformation is given by $u_{t,i} = \Phi^{-1}(F_{\text{mix-1}}(z_{t,i}; \mathbf{z}_{<t}, \mathbf{z}_{t,<i}))$, where $F_{\text{mix-1}}$ is the CDF of $p_{\text{mix-1}}$ and $\Phi^{-1}$ is the inverse standard normal CDF. This mapping is invertible and its key properties are summarized in Table 1 (see Appendix A for details).

The log absolute Jacobian determinant for this transformation, central to normalizing flows, takes the form $\log \left| \frac{\partial u_{t,i}}{\partial z_{t,i}} \right| = \log p_{\text{mix-1}}(z_{t,i}|\mathbf{z}_{<t}, \mathbf{z}_{t,<i}) - \log \mathcal{N}(u_{t,i}; 0, 1)$. This expresses the log-density of $z_{t,i}$ under the mixture model as the sum of the log-density of $u_{t,i}$ under the standard normal and the log-Jacobian:

$$\log p_{\text{mix-1}}(z_{t,i}|\mathbf{z}_{<t}, \mathbf{z}_{t,<i}) = \log \mathcal{N}(u_{t,i}; 0, 1) + \log \left| \frac{\partial u_{t,i}}{\partial z_{t,i}} \right|. \quad (6)$$

In this way, learning the mixture parameters $\left[\boldsymbol{\pi}, \mathbf{m}, \boldsymbol{\sigma}^2\right]$ is equivalent to learning a 1D Mixture-CDF flow layer. We summarize this in the following proposition, where we omit the condition $\mathbf{z}_{<t}, \mathbf{z}_{t,<i}$ for brevity.

> **Proposition 1. (Single-dim) Equivalence of mixture distribution and Mixture-CDF Flow.**
>
> Let $p_{\text{mix}-1}(z) = \sum_{k=1}^{V} \pi_k \mathcal{N}(z; m_k, \sigma_k^2)$ be a 1D MoG probability density function, and let $F_{\text{mix}-1}(z)$ be its corresponding cumulative distribution function (CDF). The transformation $f : \mathbb{R} \to \mathbb{R}$ defined by $u = f(z) = \Phi^{-1}(F_{\text{mix}-1}(z))$, where $\Phi$ is the standard normal CDF, is an exact normalizing flow between the density $p_{\text{mix}-1}(z)$ and the standard normal density $\mathcal{N}(u; 0, 1)$.
>
> *Proof.* The result follows directly from the probability integral transform theorem, see Appendix A for details. □

### 3.2.2 Token-wise Autoregressive $d$-D Mixture Modeling

Alternatively, $p(\mathbf{z}_t|\mathbf{z}_{<t})$ can be modeled directly as a $d$-dimensional distribution. We model $p(\mathbf{z}_t|\mathbf{z}_{<t})$ directly as a $d$-dimensional distribution using a mixture of Gaussians with a shared global codebook $\Phi_S = \{(\boldsymbol{\mu}_k, \sigma_k^2)\}_{k=1}^{V}$ containing $V$ Gaussian components. The network only predicts the $V$ mixture weights $\boldsymbol{\pi}_t(\mathbf{z}_{<t})$ (a $V$-dim vector) from the history $\mathbf{z}_{<t}$:

$$p(\mathbf{z}_t|\mathbf{z}_{<t}) = \sum_{k=1}^{V} \boldsymbol{\pi}_t(\mathbf{z}_{<t})[k]\mathcal{N}(\mathbf{z}_t; \boldsymbol{\mu}_k, \sigma_k^2\boldsymbol{I}), \quad \boldsymbol{\pi}_t(\mathbf{z}_{<t}) = \text{Transformer}(\mathbf{z}_{<t}) \tag{7}$$

This approach requires predicting only $V$ parameters per time step, making it computationally efficient. We denote this $d$-dimensional mixture PDF as $p_{\text{mix-}d}(\mathbf{z}_t|\mathbf{z}_{<t})$.

**Connection to Discrete AR Language Models.** This formulation is closely linked to discrete autoregressive language models; see Sec. F for details.

**Transforming $d$-D Mixture Density to an Invertible Flow.** The $d$-dimensional conditional mixture density $p_{\text{mix-}d}$ can also be realized as an invertible normalizing flow layer. Unlike the 1D case, where the CDF and its inverse are available in closed form, a direct CDF-based transformation is not tractable for general $d$-dimensional mixtures. Instead, we use a sequential transformation $\mathbf{u} = \mathbf{g}_d(\mathbf{z}; \mathbf{z}_{<t})$ based on Rosenblatt's theorem [55], which maps $\mathbf{z} \in \mathbb{R}^d$ to $\mathbf{u} \in \mathbb{R}^d$ such that $\mathbf{u}$ follows a standard Gaussian $\mathcal{N}(\mathbf{0}, \boldsymbol{I}_d)$. This process, described in Algorithm 1, proceeds dimension by dimension: at each step $i$, a 1D Mixture-CDF transform is applied to $z_i$, conditioned on the previous components $\mathbf{z}_{<i}$. The full derivation, including the inverse transformation, is given in Appendix B.

The log absolute Jacobian determinant of this transformation has a form directly analogous to the 1D case (see Appendix B): $\log|\det J_{\mathbf{g}_d}(\mathbf{z})| = \log p_{\text{mix-}d}(\mathbf{z}_t|\mathbf{z}_{<t}) - \log\mathcal{N}(\mathbf{u}; \mathbf{0}, \boldsymbol{I}_d)$. This relationship, also summarized in Table 1, shows that the log-probability of $\mathbf{z}$ under the $d$-dimensional mixture is given by the change of variables formula:

$$\log p_{\text{mix-}d}(\mathbf{z}_t|\mathbf{z}_{<t}) = \log\mathcal{N}(\mathbf{u}; \mathbf{0}, \boldsymbol{I}_d) + \log|\det J_{\mathbf{g}_d}(\mathbf{z})| \tag{8}$$

We summarize this in the following proposition, where we omit the condition $\mathbf{z}_{<t}$ for brevity.

---

**Proposition 2. (Multi-dim) Equivalence of mixture distribution and Mixture-Rosenblatt Flow.**

Let $p_{\text{mix-}d}(\mathbf{z}) = \sum_{k=1}^{V} \pi_k \mathcal{N}(\mathbf{z}; \mathbf{m}_k, \sigma_k^2 \mathbf{I}_d)$ be a $d$-dimensional MoG PDF. Let the transformation $\mathbf{g} : \mathbb{R}^d \to \mathbb{R}^d$ be defined sequentially for $i = 1, \ldots, d$ by

$$u_i = g_i(\mathbf{z}) = \Phi^{-1}\big(F_i(z_i|\mathbf{z}_{<i})\big),$$

where $F_i(z_i|\mathbf{z}_{<i})$ is the CDF of the true conditional probability distribution $p(z_i|\mathbf{z}_{<i})$ derived from $p_{\text{mix-}d}(\mathbf{z})$ (see Lemma 1). This transformation is an exact normalizing flow between $p_{\text{mix-}d}(\mathbf{z})$ and the standard $d$-dimensional normal distribution $\mathcal{N}(\mathbf{u}; \mathbf{0}, \mathbf{I}_d)$.

*Proof.* See Appendix B for details. □

---

We also provide a proof of the invertibility and differentiability of the proposed Rosenblatt Flow in Proposition 3 in Appendix Sec. B.

### 3.3 Stacking Flow Layers for Expressive Priors

To construct a flexible prior $p(\mathbf{z}_{1:T})$ over the latent sequence, we stack multiple invertible flow layers. Each flow $f^{(\ell)}$ transforms its input $\mathbf{h}_{1:T}^{(\ell-1)}$ to output $\mathbf{h}_{1:T}^{(\ell)}$, starting from $\mathbf{h}_{1:T}^{(0)} \triangleq \mathbf{z}_{1:T}$ and ending with $\mathbf{u}_{1:T} \triangleq \mathbf{h}_{1:T}^{(L)}$, which is modeled by a standard Gaussian base distribution $p_{\text{base}}(\mathbf{u}_{1:T})$.

The model is trained by maximizing the ELBO. For a single sample, the per-token objective is:

$$\mathcal{L}_t \approx -\log\left(\sum_{k=1}^{V} \mathcal{N}_k(\mathbf{z}_t)\right) + \log\mathcal{N}(\mathbf{u}_t; \mathbf{0}, \boldsymbol{I}) + \sum_{\ell=1}^{L}\left[\log p_{\text{mix},t}^{(\ell)}(\mathbf{h}_t^{(\ell-1)}|\mathcal{C}_t^{(\ell)}) - \log\mathcal{N}(\mathbf{h}_t^{(\ell)}; \mathbf{0}, \boldsymbol{I})\right] \tag{9}$$

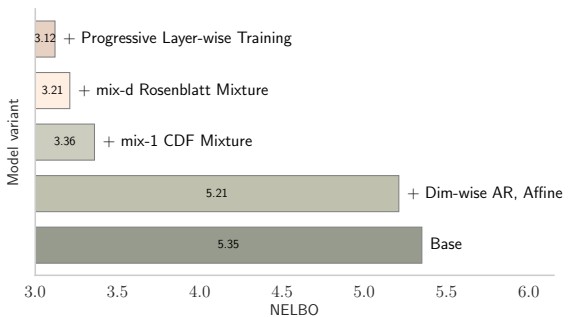

Figure 2: Ablation on model variants.

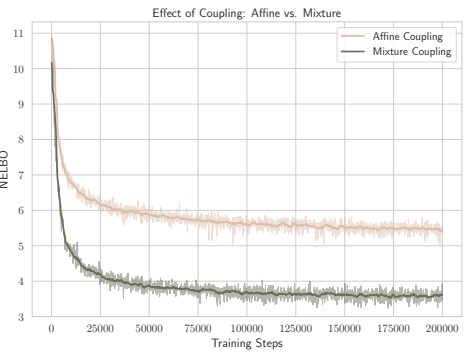

Figure 3: Affine vs. Mixture coupling.

where the first term applies when using the tied encoder-decoder ( equation 3.1). Here, $p_{\text{mix},t}^{(\ell)}$ is the mixture density for the input to layer $\ell$, and $\mathcal{C}_t^{(\ell)}$ denotes the conditioning information.

For each flow layer $f^{(\ell)}$, the objective is twofold: (a) maximize the log-likelihood of its input under the learned mixture $p_{\text{mix},t}^{(\ell)}$, and (b) encourage the output $\mathbf{h}_t^{(\ell)}$ to match the standard normal distribution. This is achieved by maximizing $\log p_{\text{mix},t}^{(\ell)}(\mathbf{h}_t^{(\ell-1)}|\mathcal{C}_t^{(\ell)}) - \log \mathcal{N}(\mathbf{h}_t^{(\ell)}; \mathbf{0}, \boldsymbol{I})$ for each layer and token.

This stacked structure enables the model to incrementally transform the latent variables and capture complex dependencies. Different flow types and autoregressive directions can be combined across layers for greater expressiveness.

**Progressive Layer-wise Training**   Our stacked normalizing flow prior naturally supports progressive layer-wise training, which simplifies optimization for deep flow models. We start by training the first block of flow layers (e.g., $f^{(1)}$ to $f^{(k_1)}$), possibly together with the VAE encoder and decoder, until their ELBO contribution (Eq. equation 9) stabilizes on a validation set. Once converged, we freeze their parameters and add the next block of layers (e.g., $f^{(k_1+1)}$ to $f^{(k_2)}$), training only the new layers to further transform the fixed output from the previous block. This process repeats, adding and training new layers while keeping earlier ones fixed, until all $L$ flow layers are trained. Progressive training improves stability, as each new layer incrementally normalizes the representation, and helps the model reach better optima in deep architectures.

## 4   Experiments

We conduct experiments to evaluate our proposed continuous latent space framework for autoregressive language modeling. The primary objectives are twofold: first, to assess its likelihood performance on standard language modeling benchmarks, and second, to demonstrate the flexible modeling capabilities enabled by the continuous and flow-based formulation.

**Datasets and Evaluation.**   We evaluate our models on standard language modeling benchmarks, specifically TEXT8 [47] and OPENWEBTEXT [22]. For character-level tasks, we report bits per character (BPC), while for word-level tasks, we use perplexity (PPL) or negative ELBO (NELBO) on the respective test sets, given the VAE-based nature of our model.

**Model Configurations.**   We conduct our experiments using the GPT2-Small architecture, following the setup in [1, 56, 58]. This model has 12 layers, a hidden size of 768, and 12 attention heads. All conditional distributions within the flow layers are parameterized by Transformers, as described in Sections 3.2.1 and 3.2.2. We primarily explore two model configurations:

- **Mix-1**: This variant employs a dimension-wise autoregressive 1-D mixture prior ($p(z_{t,i} \mid \mathbf{z}_{<t}, \mathbf{z}_{t,<i})$; see Sec. 3.2.1). The model consists of three flow layers with alternating directions (left-to-right and right-to-left), starting from the $\mathbf{z} \to \mathbf{u}$ direction. The number of Transformer layers in each flow is $[2, 2, 8]$. Unless otherwise specified in ablation studies, we use $V = 64$ mixture components for OPENWEBTEXT and $V = 27$ for TEXT8. To achieve dimension-wise autoregression, we use the same MLP-based MAF [50] in [41].

- **Mix-d**: This variant uses a token-wise autoregressive $d$-dimensional mixture prior ($p(\mathbf{z}_t \mid \mathbf{z}_{<t})$; see Sec. 3.2.2). The model is composed of three flow blocks, trained progressively

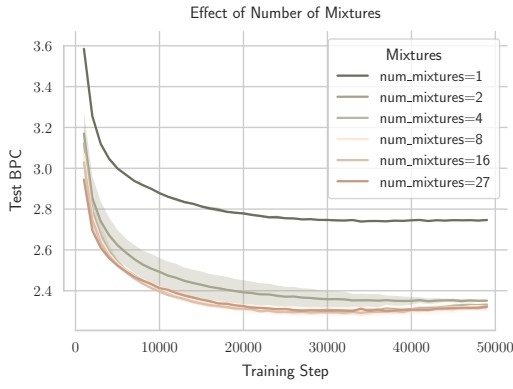

Figure 4: Effect of mixture components numbers.

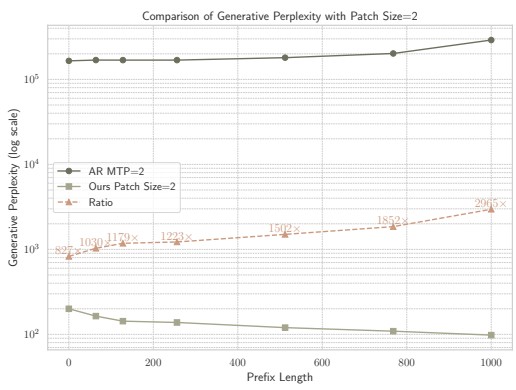

Figure 5: Generative PPL evaluation with patch size 2.

Table 3: Unconditional samples from `TarFlowLM`.

| |
|---|
| seeming that he is the only person here. He is tall. He is balding, with a face that is neither well-shaped, nor is it really human."\n"The man is a man, isn't he?" The man in the middle was curious. "How does he get there?"\n"He does. He's not an ordinary person. He's not a monster. He's a human. He's a human with extraordinary skills. He's an amazing man."\nThe other man had been watching the man, who he called human, in silence for almost an hour. He didn't look too happy, but he was still a human, so it didn't matter. He was looking at that man, not at his body, but at the man himself, which was why he was so excited. He had watched him for so long, so long |

| |
|---|
| Soul! I'm gonna be in love with my dog!" She said, giving him a big hug and then making sure she'd come home by now! (Hoping that she'd give him a hug too...) The two boys came to their parents' bedroom and went to sleep. After that, they continued to play a game of chess with their parents. (It was a bit of a joke in our world, so I'll just say...) The two boys had a long chat. A lot of them were talking about the idea of what a dog should look like... I guess I will make him a cute dog! Anyway, they started talking about things like... "Do you want to make me your dog?" Then they began to tease me, "Tell me, how long are you gonna stay?" |

as outlined in Sec. 3.3. The first flow layer uses `Mix-d` Mixture-Rosenblatt coupling, with the Gaussian mixture codebook tied to the encoder. The remaining two layers use `Mix-1` Mixture-CDF coupling. For OPENWEBTEXT, we set $V = 50257$ in the first flow layer and $V = 64$ in the next two; for TEXT8, we use $V = 27$ in the first layer and $V = 2$ in the remaining layers.

We use a latent embedding dimension of $d = 16$ for all OPENWEBTEXT experiments and $d = 5$ for all TEXT8 experiments.

**Perplexity evaluation** We assess the language modeling capabilities of our approach by training models on three standard benchmarks: TEXT8, and OPENWEBTEXT. For each dataset, we evaluate the NELBO of our trained models on their respective validation or test splits. Specifically, for TEXT8, we follow the established character-level setup and data splits, typically using fixed-length text chunks for training. For OPENWEBTEXT, we train models using the common GPT-2 tokenization and a context length of 1,024 tokens, reserving a portion of the dataset for validation, upon which NELBO is reported. We report results in Table 2.

**Results discussion.** Table 2 shows the discrete autoregressive Transformer is best (Text8 1.23 BPC; OWT 17.54 PPL). `TarFlowLM` is competitive among continuous-space/flow baselines: `Mix-d` (22.64 PPL; 1.30 BPC) outperforms several diffusion/NF methods and approaches the strongest diffusion results; `Mix-1` is on par with Gaussian diffusion. While below discrete AR on PPL/BPC, `TarFlowLM` adds bi-directional, block-wise, and hierarchical flexibility.

**Importance of Mixture Coupling** We provide empirical evidence for the effectiveness of the mixture coupling layers introduced earlier. Specifically, we compare two model variants on the OPENWEBTEXT dataset: (1) a baseline using only affine coupling layers (6 flow blocks, each with a 2-layer causal transformer, totaling 12 transformer layers), and (2) an identical architecture but with `Mix-1D` (mixture) coupling layers. As shown in Fig. 3, incorporating mixture couplings leads to a substantial improvement in negative ELBO, clearly demonstrating their importance for modeling discrete text data.

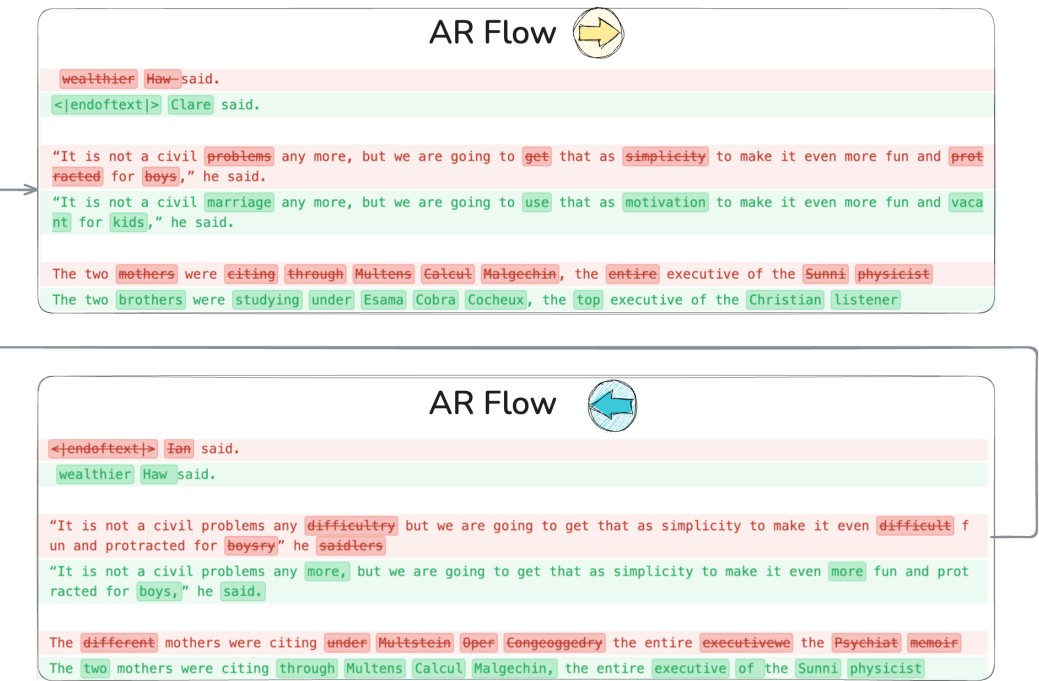

Figure 6: Demonstration of the stacked autoregressive flows for text editing in a continuous latent space. Each panel shows the transformation of the decoded text from input (green) to output (red), while also highlighting the difference (edits) made by the flow layer.

**Flexible Patch Size: Block-wise Multi-token Generation**    Our framework offers greater flexibility in sequence modeling by moving beyond the traditional single-token generation approach. Instead, it processes "patches" of multiple tokens at once, allowing for simultaneous prediction and generation. In contrast, standard discrete autoregressive language models—even those trained with multi-token objectives [21, 42]—still generate tokens strictly one at a time to maintain autoregressive consistency, since generating multiple tokens together breaks their core assumptions. We address this by using a continuous latent space and stacked normalizing flows to capture dependencies within each token patch. First, stacked autoregressive flows with alternating directions (see Fig. 1, patch size two) enable later layers to propagate and refine joint information, even if earlier blocks treat tokens as conditionally independent (as shown by the shaded area). Second, intra-patch dependencies are further modeled by mixing or permuting latent embedding dimensions between flow layers (channel mixing, Algorithm 2), allowing information to flow between token positions within a patch. Together, these mechanisms let the model effectively learn intra-patch dependencies. Block-wise generation also changes the computational landscape: it shifts some of the workload from sequence length to model depth (number of flow layers), reducing the effective sequence length. As shown in Fig. 7, this can significantly lower the total FLOPs needed to process a sequence (e.g., 1024 tokens) compared to standard transformers, especially with larger patch sizes, all within a generalized autoregressive framework. We also trained a discrete autoregressive language model with a two-token prediction objective and compared its generative perplexity (measured by GPT-2 Large) to our model, conditioning on varying prefix lengths. The results show a large gap, highlighting how standard AR LMs break down when the next-single-token assumption is violated. See Fig. 5 for generative PPL with patch size 2. See Table 3 for unconditional samples from TarFlowLM.

**Flexible vocabulary size.**    A key flexibility of our model is the ability to freely choose the number of mixture components in each stacked autoregressive flow layer, especially when using mixture-based transformations (see Sections 3.2.1 and 3.2.2). This number, denoted as $V$ in Eq. equation 5 (1D) and Eq. equation 7 ($d$-dimensional), serves as an internal "vocabulary" for the flow, and can be set independently for each layer.

This design enables strategies like coarse-to-fine modeling: earlier layers can use fewer components to capture broad structure, while later layers use more to refine details. Importantly, the internal mixture size does not need to match the data's discrete vocabulary.

Empirically, this flexibility is robust. On TEXT8, as shown in Fig. 4, the model performs consistently well across a wide range of mixture counts (2 to 27), even though the dataset's vocabulary is 27 characters. On OPENWEBTEXT, using 64 or more mixture components in the mix-1d coupling layers

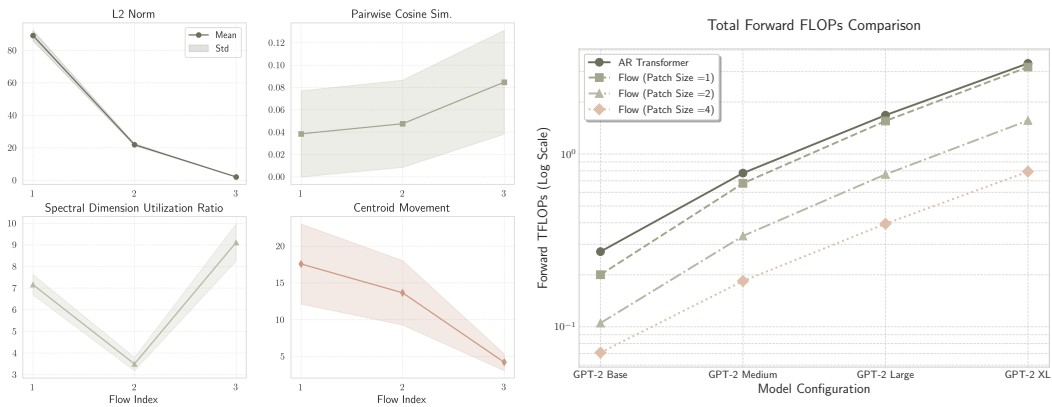

Figure 7: Left: Latent evolution analysis. Right: Transformer FLOPs comparison.

matches the performance of using the full GPT-2 tokenizer size (50257). In summary, our framework allows the internal "vocabulary" of mixture components to be tuned for the task, decoupling it from the data's vocabulary and providing a practical lever for model design and efficiency.

**Flexible text editing in continuous space.** Our framework models language in a continuous latent space using stacked autoregressive normalizing flows, enabling us to decode intermediate representations at each flow layer. This provides a unique, step-by-step view of how the model incrementally edits and refines latent states into coherent text—a process that is difficult to realize in discrete token spaces. A key advantage of this approach is its natural support for coarse-to-fine generation: early flow layers make broad, global changes to the latent sequence, while later layers focus on increasingly fine-grained adjustments. We quantify this progression using several metrics, including the mean L2 norm of token embeddings, mean pairwise cosine similarity, Participation Ratio (PR), and centroid movement between layers. Typically, we observe that the mean L2 norm decreases and cosine similarity increases across layers, indicating that representations become more compact and internally coherent. The centroid movement diminishes with depth, reflecting the transition from coarse to fine editing. Alternating flow directions (e.g., left-to-right and right-to-left) further enrich the representations by integrating information from both past and future context. This hierarchical, coarse-to-fine refinement is a direct benefit of operating in a continuous space, where smooth, learnable edits are possible at every stage. Fig. 6 visualizes per-layer edits on decoded text. For further details and quantitative results, see Appendix G.

**Model Ablation** We performed ablation studies on OPENWEBTEXT to evaluate key architectural choices (see Fig. 2). Adding dimension-wise autoregressive affine flows to a base model yields only minor NELBO improvement. Replacing affine couplings with 1-D mixture-CDF transforms greatly improves NELBO, and using full $d$-dimensional Mixture-Rosenblatt flow further boosts performance. Progressive layer-wise training—freezing earlier flow blocks—achieves the best results, demonstrating the benefit of a simple curriculum over flow depth. We also examined the effect of the number of Gaussian components $V$ in 1-D mixture-CDF couplings on TEXT8 (Fig. 4). We did not use MAF module to isolate the mixture coupling effect. A single component (affine flow) gives high BPC, while all other $> 1$ number of mixtures can work well.

## 5  Conclusion

This work introduced `TarFlowLM`, a novel framework that recasts language modeling in continuous latent spaces using Transformer-based autoregressive normalizing flows. Our approach achieves strong likelihood performance while enabling significant flexibility, including bi-directional context, block-wise generation, and hierarchical refinement. These findings highlight the potential of continuous latent variable models with mixture-based transformations to advance flexible and expressive sequence generation.

## Limitations

While our framework, `TarFlowLM`, demonstrates strong likelihood performance and notable modeling flexibility, we acknowledge certain limitations inherent to the current instantiation, particularly concerning sampling efficiency. Autoregressive normalizing flows, by their nature, involve a sequential generation process. Each step in the generation depends on the previously generated ones, which can lead to slower sampling speeds compared to models that allow for more parallelized generation, such as some non-autoregressive or diffusion-based approaches.

In this work, our primary focus was on establishing the viability of transformer-based autoregressive flows for flexible language modeling in continuous latent spaces, emphasizing likelihood estimation and the exploration of novel modeling capabilities like block-wise generation and hierarchical refinement. Consequently, a systematic investigation into optimizing sampling speed or exploring advanced sampling techniques (e.g., parallel decoding strategies, distillation) for this class of flow-based language models was beyond the scope of the current paper.

Improving the sampling efficiency of autoregressive flow models for text generation remains an important and active area of research. We consider this a promising direction for future work, which could involve developing specialized architectural modifications, exploring alternative flow parametrizations, or adapting techniques from other generative modeling paradigms to accelerate the sampling process without compromising the expressive power and flexibility demonstrated in this paper.

## Broader Impacts

The research presented in this paper, `TarFlowLM`, introduces a novel framework for language modeling in continuous latent spaces using transformer-based autoregressive normalizing flows. As with many advancements in machine learning and artificial intelligence, particularly in the domain of generative models, this work has the potential for a range of societal impacts, both positive and negative.

**Potential Positive Societal Impacts.** The enhanced flexibility offered by our approach—such as bi-directional context modeling, block-wise generation, and hierarchical refinement—could lead to significant positive developments.

- **Improved Creative Tools:** Models capable of more nuanced and controllable generation could serve as powerful assistive tools for writers, artists, and designers, helping to brainstorm, draft, and refine creative content. The ability to edit text in a continuous latent space and observe intermediate generation steps might offer new paradigms for human-AI collaboration in creative tasks.

- **Enhanced Controllability in NLP Applications:** The flexible architectural components, like adaptable patch sizes and independent mixture component selection per layer, might enable finer-grained control over the generation process. This could be beneficial for applications requiring specific stylistic attributes, content constraints, or conditional generation tasks (e.g., personalized dialogue systems, summarization with specific focuses).

- **Advancements in Fundamental Understanding:** By exploring language modeling in continuous latent spaces, this research contributes to a broader understanding of how complex sequential data can be represented and manipulated. Such foundational insights can spur further innovation in machine learning for sequences beyond just text, potentially impacting fields like bioinformatics or time-series analysis.

- **Potential for More Efficient Modeling:** While sampling efficiency is noted as a limitation, the architectural flexibility (e.g., block-wise processing potentially reducing effective sequence length for attention mechanisms) hints at avenues for developing more computationally efficient models for certain tasks or sequence lengths, which could make advanced NLP capabilities more accessible.

**Potential Negative Societal Impacts and Risks.** Advancements in generative language modeling, including the techniques explored in this paper, also carry inherent risks that warrant careful consideration.

- **Economic Disruption:** As generative AI tools become more capable, there is potential for economic disruption in professions that involve content creation or communication. While these tools can also augment human capabilities, the long-term societal adjustments need to be considered.
- **Security Concerns:** The potential for generating human-like text could be used in social engineering attacks, to create more convincing phishing emails, or to automate an abusive online presence.

**Considerations for Responsible Development.** While this work is primarily foundational, focusing on a novel modeling paradigm, it is crucial for the research community to engage in ongoing discussions about the responsible development and deployment of such technologies. Future research building upon `TarFlowLM` should ideally incorporate investigations into:

- Techniques for detecting synthetically generated text from such continuous-space flow models.
- Methods for identifying and mitigating biases in the learned representations and generated outputs.
- Developing frameworks for controllable and ethical AI, ensuring that generative capabilities are aligned with human values and safety.

Our aim is to contribute to the open scientific exploration of generative models. We believe that by understanding the capabilities and limitations of new approaches like `TarFlowLM`, the community is better equipped to foresee potential impacts and work towards harnessing these technologies for societal benefit while mitigating potential harms.

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

# Appendix

## Table of Contents

## A    Details for $1$-D Mixture-CDF Flow Layer

This section details the derivation for the 1D Mixture of Gaussians CDF (Cumulative Distribution Function) normalizing flow layer, as introduced in Section 3.2.1 of the main text. The conditional mixture density for a single scalar component $z_{t,i}$ given its context $(\mathbf{z}_{<t}, \mathbf{z}_{t,<i})$ is defined in Eq. equation 5 as:

$$p(z_{t,i}|\mathbf{z}_{<t}, \mathbf{z}_{t,<i}) = \sum_{k=1}^{V} \boldsymbol{\pi}[k] \mathcal{N}(z_{t,i}; \mathbf{m}[k], \boldsymbol{\sigma}^2[k])$$

where the parameter bundle $\left[\boldsymbol{\pi}, \mathbf{m}, \boldsymbol{\sigma}^2\right]$ (containing $V$ mixture weights, means, and variances respectively) is predicted by a Transformer model from the context $(\mathbf{z}_{<t}, \mathbf{z}_{t,<i})$. We denote this 1D mixture probability density function (PDF) as $p_{\text{mix-1}}(z_{t,i}|\mathbf{z}_{<t}, \mathbf{z}_{t,<i})$. This conditional PDF can be used to construct an invertible transformation suitable for normalizing flows.

The CDF of $p_{\text{mix-1}}(z_{t,i}|\mathbf{z}_{<t}, \mathbf{z}_{t,<i})$, denoted $F_{\text{mix-1}}(z_{t,i}; \mathbf{z}_{<t}, \mathbf{z}_{t,<i})$, is a weighted sum of the CDFs of the individual Gaussian components:

$$F_{\text{mix-1}}(z_{t,i}; \mathbf{z}_{<t}, \mathbf{z}_{t,<i}) = \sum_{k=1}^{V} \boldsymbol{\pi}[k] \Phi\left(\frac{z_{t,i} - \mathbf{m}[k]}{\boldsymbol{\sigma}[k]}\right)$$

where $\Phi(\cdot)$ is the CDF of the standard normal distribution $\mathcal{N}(0,1)$, and $\boldsymbol{\sigma}[k] = \sqrt{\boldsymbol{\sigma}^2[k]}$ is the standard deviation of the $k$-th component. The mixture parameters $\boldsymbol{\pi}[k], \mathbf{m}[k], \boldsymbol{\sigma}[k]$ are functions of $(\mathbf{z}_{<t}, \mathbf{z}_{t,<i})$.

Using the probability integral transform, if a random variable $Z_{t,i}$ is drawn from $p_{\text{mix-1}}(Z_{t,i}|\mathbf{z}_{<t}, \mathbf{z}_{t,<i})$, then $F_{\text{mix-1}}(Z_{t,i}; \mathbf{z}_{<t}, \mathbf{z}_{t,<i})$ is uniformly distributed on $[0, 1]$. To map $z_{t,i}$ to a variable $u_{t,i}$ that follows a standard normal distribution $\mathcal{N}(0,1)$, we compose the mixture CDF with the inverse standard normal CDF, $\Phi^{-1}(\cdot)$:

$$u_{t,i} = \Phi^{-1}\big(F_{\text{mix-1}}(z_{t,i}; \mathbf{z}_{<t}, \mathbf{z}_{t,<i})\big) \tag{10}$$

This transformation is invertible. The variable $z_{t,i}$ can be recovered from $u_{t,i}$ using $z_{t,i} = F_{\text{mix-1}}^{-1}(\Phi(u_{t,i}); \mathbf{z}_{<t}, \mathbf{z}_{t,<i})$, where $F_{\text{mix-1}}^{-1}$ is the inverse CDF (quantile function) of the mixture. $F_{\text{mix-1}}^{-1}$ generally does not have a closed-form expression and may require numerical methods for its evaluation.

For this transformation to serve as a normalizing flow layer, its Jacobian determinant is needed. In this 1D case, this is the absolute value of the derivative $\frac{\partial u_{t,i}}{\partial z_{t,i}}$. We compute this using the chain rule. Let $y_{\text{cdf}} = F_{\text{mix-1}}(z_{t,i}; \mathbf{z}_{<t}, \mathbf{z}_{t,<i})$. Then $u_{t,i} = \Phi^{-1}(y_{\text{cdf}})$, and

$$\frac{\partial u_{t,i}}{\partial z_{t,i}} = \left.\frac{d\Phi^{-1}(y_{\text{cdf}})}{dy_{\text{cdf}}}\right|_{y_{\text{cdf}}=F_{\text{mix-1}}(z_{t,i};\mathbf{z}_{<t},\mathbf{z}_{t,<i})} \cdot \frac{\partial F_{\text{mix-1}}(z_{t,i}; \mathbf{z}_{<t}, \mathbf{z}_{t,<i})}{\partial z_{t,i}}$$

We evaluate each term:

1. The derivative of the inverse standard normal CDF, $\frac{d\Phi^{-1}(y_{\text{cdf}})}{dy_{\text{cdf}}}$: If $u_{t,i} = \Phi^{-1}(y_{\text{cdf}})$, then $y_{\text{cdf}} = \Phi(u_{t,i})$. Differentiating $y_{\text{cdf}} = \Phi(u_{t,i})$ with respect to $u_{t,i}$ yields $\frac{dy_{\text{cdf}}}{du_{t,i}} = \mathcal{N}(u_{t,i}; 0, 1)$, where $\mathcal{N}(u_{t,i}; 0, 1)$ is the PDF of the standard normal distribution. Therefore, $\frac{du_{t,i}}{dy_{\text{cdf}}} = \frac{d\Phi^{-1}(y_{\text{cdf}})}{dy_{\text{cdf}}} = \frac{1}{\mathcal{N}(u_{t,i};0,1)}$.

2. The derivative of the mixture CDF with respect to $z_{t,i}$, $\frac{\partial F_{\text{mix-1}}(z_{t,i};\mathbf{z}_{<t},\mathbf{z}_{t,<i})}{\partial z_{t,i}}$:

$$\frac{\partial F_{\text{mix-1}}(z_{t,i}; \mathbf{z}_{<t}, \mathbf{z}_{t,<i})}{\partial z_{t,i}} = \frac{\partial}{\partial z_{t,i}} \sum_{k=1}^{V} \boldsymbol{\pi}[k] \Phi\left(\frac{z_{t,i} - \mathbf{m}[k]}{\boldsymbol{\sigma}[k]}\right)$$

$$= \sum_{k=1}^{V} \boldsymbol{\pi}[k] \mathcal{N}\left(\frac{z_{t,i} - \mathbf{m}[k]}{\boldsymbol{\sigma}[k]}; 0, 1\right) \cdot \frac{1}{\boldsymbol{\sigma}[k]}$$

Recognizing that $\frac{1}{\boldsymbol{\sigma}[k]} \mathcal{N}\left(\frac{z_{t,i}-\mathbf{m}[k]}{\boldsymbol{\sigma}[k]}; 0, 1\right)$ is the PDF $\mathcal{N}(z_{t,i}; \mathbf{m}[k], \boldsymbol{\sigma}^2[k])$, the sum becomes:

$$\frac{\partial F_{\text{mix-1}}(z_{t,i}; \mathbf{z}_{<t}, \mathbf{z}_{t,<i})}{\partial z_{t,i}} = \sum_{k=1}^{V} \boldsymbol{\pi}[k] \mathcal{N}(z_{t,i}; \mathbf{m}[k], \boldsymbol{\sigma}^2[k]) = p_{\text{mix-1}}(z_{t,i}|\mathbf{z}_{<t}, \mathbf{z}_{t,<i})$$

Thus, the derivative of the mixture CDF is its PDF.

Combining these results, the derivative of the transformation is:

$$\frac{\partial u_{t,i}}{\partial z_{t,i}} = \frac{p_{\text{mix-1}}(z_{t,i}|\mathbf{z}_{<t}, \mathbf{z}_{t,<i})}{\mathcal{N}(u_{t,i}; 0, 1)}$$

The log absolute Jacobian determinant is therefore:

$$\log\left|\frac{\partial u_{t,i}}{\partial z_{t,i}}\right| = \log p_{\text{mix-1}}(z_{t,i}|\mathbf{z}_{<t}, \mathbf{z}_{t,<i}) - \log \mathcal{N}(u_{t,i}; 0, 1) \tag{11}$$

This equation relates the log-density of $z_{t,i}$ under the mixture model $p_{\text{mix-1}}$ to the log-density of $u_{t,i}$ under the standard normal base distribution and the log-Jacobian term. Rearranging Eq. equation 11 according to the change of variables formula ($\log p(z) = \log p(u) + \log|\det J_f|$):

$$\log p_{\text{mix-1}}(z_{t,i}|\mathbf{z}_{<t}, \mathbf{z}_{t,<i}) = \log \mathcal{N}(u_{t,i}; 0, 1) + \log\left|\frac{\partial u_{t,i}}{\partial z_{t,i}}\right|$$

This confirms that learning the mixture parameters $\left[\boldsymbol{\pi}, \mathbf{m}, \boldsymbol{\sigma}^2\right]$ is equivalent to learning this 1D Mixture-CDF flow layer.

When such a transformation is used as a layer $\ell$ in a stack of flows, $z_{t,i}$ corresponds to the $i$-th scalar component of the input to this layer, $h_{t,i}^{(\ell-1)}$. The output $u_{t,i}$ corresponds to the $i$-th scalar component $h_{t,i}^{(\ell)}$. The parameters for the mixture density $p_{\text{mix-1}}(h_{t,i}^{(\ell-1)}|\text{cond}_{t,i}^{(\ell)})$ for layer $\ell$ are determined by a neural network conditioned on $\text{cond}_{t,i}^{(\ell)}$, which is the layer-specific conditioning information. The term $\log p_{\text{mix-1}}(h_{t,i}^{(\ell-1)}|\text{cond}_{t,i}^{(\ell)})$ in the log-determinant calculation (Eq. equation 11, adapted for layer $\ell$) represents the log-probability of the layer's input under the specific 1D mixture model implemented by that layer.

## A.1 Proof of Proposition 1

**Proposition 1.** *Let $p_{\mathrm{mix}-1}(z) = \sum_{k=1}^{V} \pi_k \mathcal{N}(z; m_k, \sigma_k^2)$ be a 1D MoG probability density function (PDF), and let $F_{\mathrm{mix}-1}(z)$ be its corresponding cumulative distribution function (CDF). The transformation $f : \mathbb{R} \to \mathbb{R}$ defined by $u = f(z) = \Phi^{-1}(F_{\mathrm{mix}-1}(z))$, where $\Phi$ is the standard normal CDF, is an exact normalizing flow between the density $p_{\mathrm{mix}-1}(z)$ and the standard normal density $\mathcal{N}(u; 0, 1)$.*

*Proof.* The proof demonstrates the equivalence from both forward and inverse directions. A key property is that $F_{\mathrm{mix}-1}(z)$ is strictly increasing, since its derivative, $F'_{\mathrm{mix}-1}(z) = p_{\mathrm{mix}-1}(z)$, is a sum of strictly positive Gaussian densities and is therefore strictly positive. Thus, $F_{\mathrm{mix}-1}$ is a bijection from $\mathbb{R}$ to $(0, 1)$.

**Forward Direction ($Z \to U$).** Assume a random variable $Z \sim p_{\mathrm{mix}-1}(z)$. We show that the transformed variable $U = f(Z)$ follows a standard normal distribution by computing its CDF:

$$
\begin{aligned}
P(U \le u) &= P\big(\Phi^{-1}(F_{\mathrm{mix}-1}(Z)) \le u\big) \\
&= P\big(F_{\mathrm{mix}-1}(Z) \le \Phi(u)\big) && \text{(since } \Phi \text{ is strictly increasing)} \\
&= F_{\mathrm{mix}-1}\big(F_{\mathrm{mix}-1}^{-1}(\Phi(u))\big) && \text{(by definition, } P(Z \le z') = F_{\mathrm{mix}-1}(z')\text{)} \\
&= \Phi(u).
\end{aligned}
$$

The CDF of $U$ is $\Phi(u)$, which is the CDF of the standard normal distribution. Hence, $U \sim \mathcal{N}(0, 1)$.

**Inverse Direction ($U \to Z$).** Assume a random variable $U \sim \mathcal{N}(0, 1)$. We show that $Z = f^{-1}(U) = F_{\mathrm{mix}-1}^{-1}(\Phi(U))$ is distributed according to $p_{\mathrm{mix}-1}(z)$ by computing its CDF:

$$
\begin{aligned}
P(Z \le z) &= P\big(F_{\mathrm{mix}-1}^{-1}(\Phi(U)) \le z\big) \\
&= P\big(\Phi(U) \le F_{\mathrm{mix}-1}(z)\big) && \text{(since } F_{\mathrm{mix}-1} \text{ is strictly increasing).}
\end{aligned}
$$

By the probability integral transform, the random variable $\Phi(U)$ is uniformly distributed on $[0, 1]$. Therefore, $P\big(\Phi(U) \le v\big) = v$ for any $v \in [0, 1]$. Since $F_{\mathrm{mix}-1}(z) \in (0, 1)$, we have:

$$
P(Z \le z) = F_{\mathrm{mix}-1}(z).
$$

The CDF of $Z$ is $F_{\mathrm{mix}-1}(z)$. Differentiating with respect to $z$ yields the PDF of $Z$ as $p_Z(z) = \frac{d}{dz} F_{\mathrm{mix}-1}(z) = p_{\mathrm{mix}-1}(z)$, which completes the proof. $\square$

# B Details for $d$-D Mixture-Rosenblatt Flow Layer

We use $\mathcal{C}$ to denote the conditioning context variable. A $d$-dimensional conditional mixture density, such as $p_{\mathrm{mix}}(\mathbf{z}|\mathcal{C}) = \sum_{k=1}^{V} \pi_k(\mathcal{C}) \mathcal{N}(\mathbf{z}; \mathbf{m}_k(\mathcal{C}), s_k^2(\mathcal{C})\boldsymbol{I}_d)$, can be realized as an exact and invertible normalizing flow layer. This construction is used for modeling complex conditional distributions within the flow framework. For example, it can represent the prior's conditional distribution $p(\mathbf{z}_t|\mathbf{z}_{<t})$ as defined in Eq. equation 7, or a general conditional density $p_{\mathrm{mix},t}^{(\ell)}(\mathbf{h}_t^{(\ell-1)}|\mathrm{cond}_t^{(\ell)})$ for an intermediate layer $\ell$ in a stack of flows (see Section 3.3).

In this description, $\mathbf{z} \in \mathbb{R}^d$ denotes the input variable to this flow layer. If this layer is the first transformation applied to a latent variable $\mathbf{z}_t$ from the VAE, then $\mathbf{z}$ corresponds to $\mathbf{z}_t$ (which is also $\mathbf{h}_t^{(0)}$ in the notation of stacked flows). If it is an intermediate layer $\ell$ in the stack, $\mathbf{z}$ corresponds to $\mathbf{h}_t^{(\ell-1)}$. The term $\mathcal{C}$ represents the conditioning information, such as $\mathbf{z}_{<t}$ for the prior, or a layer-specific context $\mathrm{cond}_t^{(\ell)}$. The parameters $\mathbf{m}_k$ and $s_k^2$ of the Gaussian components can either be predicted based on $\mathcal{C}$, or, as in Eq. equation 7, they can be part of a shared codebook (e.g., $\{\boldsymbol{\mu}_k, \sigma_k^2\}_{k=1}^{V}$) where only the mixture weights $\pi_k(\mathcal{C})$ are predicted.

The goal is to define an invertible transformation $\mathbf{g}_d : \mathbb{R}^d \to \mathbb{R}^d$, denoted $\mathbf{u} = \mathbf{g}_d(\mathbf{z}; \mathcal{C})$, such that the output variable $\mathbf{u} \in \mathbb{R}^d$ follows a standard $d$-dimensional Gaussian distribution, i.e., $\mathbf{u} \sim \mathcal{N}(\mathbf{0}, \boldsymbol{I}_d)$. This transformation is constructed by processing the components of $\mathbf{z} = (z_1, \dots, z_d)$ sequentially, from $i = 1$ to $d$, using a method related to the Rosenblatt transformation. The forward pass is detailed in Algorithm 1.

**Forward Transformation $\mathbf{z} \mapsto \mathbf{u}$:** The transformation from $\mathbf{z}$ to $\mathbf{u}$ is built sequentially, processing one component at a time. At each step $i$, $u_i$ is computed from $z_i$ using $\mathcal{C}$ and the previously processed components $\mathbf{z}_{<i} = (z_1, \ldots, z_{i-1})$. Mixture component weights are maintained and updated throughout this process.

The procedure begins with initialization: for the first component $z_1$, the initial mixture weights are set by the parameters of $p_{\text{mix}}(\mathbf{z}|\mathcal{C})$,

$$\alpha_k^{(1)}(\mathcal{C}) = \pi_k(\mathcal{C}) \quad \text{for } k = 1, \ldots, V. \tag{12}$$

For each dimension $i = 1, \ldots, d$, the following steps are performed: The forward transformation proceeds as follows. For each dimension $i = 1, \ldots, d$:

**Conditional Density of $z_i$.** The conditional probability density of $z_i$ given $\mathcal{C}$ and $\mathbf{z}_{<i}$ is a 1D mixture of Gaussians, denoted $p_{\text{mix},i}(z_i|\mathcal{C}, \mathbf{z}_{<i})$. This is computed by marginalizing over the mixture components using the current weights $\alpha_k^{(i)}$:

$$p_{\text{mix},i}(z_i|\mathcal{C}, \mathbf{z}_{<i}) = \sum_{k=1}^{V} \alpha_k^{(i)}(\mathcal{C}, \mathbf{z}_{<i}) \mathcal{N}(z_i; m_{k,i}(\mathcal{C}), s_k^2(\mathcal{C})) \tag{13}$$

Here, $m_{k,i}(\mathcal{C})$ is the $i$-th element of the mean vector $\mathbf{m}_k(\mathcal{C})$ for component $k$, and $s_k^2(\mathcal{C})$ is the (isotropic) variance for component $k$. By construction, $p_{\text{mix},i}(z_i|\mathcal{C}, \mathbf{z}_{<i})$ is equivalent to $p(z_i|\mathcal{C}, \mathbf{z}_{<i})$.

**Conditional CDF of $z_i$.** The corresponding 1D conditional cumulative distribution function (CDF) for $z_i$ is

$$F_i(z_i|\mathcal{C}, \mathbf{z}_{<i}) = \int_{-\infty}^{z_i} p_{\text{mix},i}(\xi|\mathcal{C}, \mathbf{z}_{<i}) d\xi = \sum_{k=1}^{V} \alpha_k^{(i)}(\mathcal{C}, \mathbf{z}_{<i}) \Phi\left(\frac{z_i - m_{k,i}(\mathcal{C})}{s_k(\mathcal{C})}\right) \tag{14}$$

where $\Phi(\cdot)$ is the CDF of the standard normal distribution and $s_k(\mathcal{C}) = \sqrt{s_k^2(\mathcal{C})}$.

**Transformation to $u_i$.** The $i$-th component $u_i$ of the transformed variable $\mathbf{u}$ is obtained by applying the probability integral transform to $z_i$ using $F_i$, followed by the inverse CDF of the standard normal:

$$u_i = \Phi^{-1}(F_i(z_i|\mathcal{C}, \mathbf{z}_{<i})) \tag{15}$$

If $z_i$ is drawn from $p_{\text{mix},i}(z_i|\mathcal{C}, \mathbf{z}_{<i})$, then $F_i(z_i|\mathcal{C}, \mathbf{z}_{<i})$ is uniformly distributed on $[0, 1]$, and thus $u_i$ is standard normal. The sequential conditioning ensures that $u_1, \ldots, u_d$ are mutually independent and standard normally distributed.

**Update Mixture Weights (if $i < d$).** After processing $z_i$ and obtaining $u_i$, if $i$ is not the last dimension, the mixture weights for the next dimension, $\alpha_k^{(i+1)}$, are updated using Bayes' rule to incorporate the information from $z_i$:

$$\begin{aligned}
\alpha_k^{(i+1)}(\mathcal{C}, \mathbf{z}_{\leq i}) &= p(\text{component} = k|\mathcal{C}, \mathbf{z}_{\leq i}) \\
&= \frac{p(z_i|\text{component} = k, \mathcal{C}, \mathbf{z}_{<i}) \cdot p(\text{component} = k|\mathcal{C}, \mathbf{z}_{<i})}{p(z_i|\mathcal{C}, \mathbf{z}_{<i})} \\
&= \frac{\mathcal{N}(z_i; m_{k,i}(\mathcal{C}), s_k^2(\mathcal{C})) \cdot \alpha_k^{(i)}(\mathcal{C}, \mathbf{z}_{<i})}{p_{\text{mix},i}(z_i|\mathcal{C}, \mathbf{z}_{<i})}
\end{aligned} \tag{16}$$

where $\mathbf{z}_{\leq i} = (z_1, \ldots, z_i)$. These updated weights are then used for the next dimension.

**Inverse Transformation $\mathbf{u} \mapsto \mathbf{z}$:** The transformation $\mathbf{g}_d$ is invertible. To compute the inverse, $\mathbf{z} = \mathbf{g}_d^{-1}(\mathbf{u}; \mathcal{C})$, we proceed as follows, iterating from $i = 1$ to $d$:

**Initialization (for $i = 1$):** Set the initial mixture weights $\alpha_k^{(1)}(\mathcal{C})$ as in Eq. equation 12.

**Sequential Inversion (for $i = 1, \ldots, d$):** For each dimension $i$, given $u_i$ and the current weights $\alpha_k^{(i)}(\mathcal{C}, \mathbf{z}_{<i})$ (where $\mathbf{z}_{<i}$ have been determined in previous steps), solve for $z_i$ by finding the value that satisfies

$$F_i(z_i|\mathcal{C}, \mathbf{z}_{<i}) = \Phi(u_i) \tag{17}$$

where $F_i$ is defined in Eq. equation 14. Since $F_i$ is a strictly monotonic function (as the CDF of a continuous distribution with positive density), a unique solution for $z_i$ exists. This inversion typically requires a numerical 1D root-finding algorithm.

After determining $z_i$, if $i < d$, update the mixture weights to $\alpha_k^{(i+1)}(\mathcal{C}, \mathbf{z}_{\leq i})$ using Eq. equation 16, now with the newly found $z_i$. This process is repeated sequentially for each dimension until all components of $\mathbf{z}$ are recovered.

**Jacobian Determinant**: The Jacobian matrix $J_{\mathbf{g}_d}(\mathbf{z})$ of the forward transformation $\mathbf{u} = \mathbf{g}_d(\mathbf{z}; \mathcal{C})$ has entries $(J_{\mathbf{g}_d})_{\ell i} = \frac{\partial u_\ell}{\partial z_i}$. Due to the sequential construction, where $u_\ell$ depends only on $z_1, \ldots, z_\ell$ (and $\mathcal{C}$), the Jacobian matrix is lower-triangular. Its determinant is the product of its diagonal entries: $\det J_{\mathbf{g}_d}(\mathbf{z}) = \prod_{i=1}^{d} \frac{\partial u_i}{\partial z_i}$.

To find $\frac{\partial u_i}{\partial z_i}$, we differentiate Eq. equation 15, or its form $\Phi(u_i) = F_i(z_i | \mathcal{C}, \mathbf{z}_{<i})$. Differentiating both sides with respect to $z_i$ (treating $\mathcal{C}$ and $\mathbf{z}_{<i}$ as constant for this partial derivative):

$$\frac{\partial(\Phi(u_i))}{\partial z_i} = \frac{\partial(F_i(z_i | \mathcal{C}, \mathbf{z}_{<i}))}{\partial z_i} \tag{18}$$

Using the chain rule on the left side yields $\phi(u_i) \frac{\partial u_i}{\partial z_i}$, where $\phi(\cdot)$ is the PDF of $\mathcal{N}(0, 1)$. The right side is the derivative of a CDF with respect to its variable, which is its PDF: $\frac{\partial F_i}{\partial z_i} = p_{\text{mix},i}(z_i | \mathcal{C}, \mathbf{z}_{<i})$. Thus,

$$\phi(u_i) \frac{\partial u_i}{\partial z_i} = p_{\text{mix},i}(z_i | \mathcal{C}, \mathbf{z}_{<i}) \tag{19}$$

Since $\phi(u_i) > 0$, we can write:

$$\frac{\partial u_i}{\partial z_i} = \frac{p_{\text{mix},i}(z_i | \mathcal{C}, \mathbf{z}_{<i})}{\phi(u_i)} \tag{20}$$

The log absolute Jacobian determinant is then:

$$\log |\det J_{\mathbf{g}_d}(\mathbf{z})| = \sum_{i=1}^{d} \log \left| \frac{\partial u_i}{\partial z_i} \right|$$
$$= \sum_{i=1}^{d} \left( \log p_{\text{mix},i}(z_i | \mathcal{C}, \mathbf{z}_{<i}) - \log \phi(u_i) \right) \tag{21}$$

By the chain rule of probability, the log-density of the input variable $\mathbf{z}$ under the conditional mixture $p_{\text{mix}}(\mathbf{z}|\mathcal{C})$ is:

$$\log p_{\text{mix}}(\mathbf{z}|\mathcal{C}) = \log \left( \prod_{i=1}^{d} p(z_i | \mathcal{C}, \mathbf{z}_{<i}) \right) = \sum_{i=1}^{d} \log p(z_i | \mathcal{C}, \mathbf{z}_{<i}) \tag{22}$$

In our construction, $p(z_i | \mathcal{C}, \mathbf{z}_{<i})$ is precisely $p_{\text{mix},i}(z_i | \mathcal{C}, \mathbf{z}_{<i})$ from Eq. equation 13. So, $\sum_{i=1}^{d} \log p_{\text{mix},i}(z_i | \mathcal{C}, \mathbf{z}_{<i}) = \log p_{\text{mix}}(\mathbf{z}|\mathcal{C})$. The log-density of the transformed variable $\mathbf{u}$ under the standard $d$-dimensional Gaussian base distribution $p_{\text{base}}(\mathbf{u}) = \mathcal{N}(\mathbf{u}; \mathbf{0}, \mathbf{I}_d)$ is $\log p_{\text{base}}(\mathbf{u}) = \sum_{i=1}^{d} \log \mathcal{N}(u_i; 0, 1) = \sum_{i=1}^{d} \log \phi(u_i)$. Substituting these into Eq. equation 21, we arrive at the expression for the log-determinant in normalizing flows:

$$\log |\det J_{\mathbf{g}_d}(\mathbf{z})| = \log p_{\text{mix}}(\mathbf{z}|\mathcal{C}) - \log \mathcal{N}(\mathbf{u}; \mathbf{0}, \mathbf{I}_d) \tag{23}$$

This confirms that the sequential transformation $\mathbf{g}_d$ correctly implements the desired conditional mixture density $p_{\text{mix}}(\mathbf{z}|\mathcal{C})$ as a normalizing flow layer. When such a layer is used, for example as layer $f_t^{(\ell)}$ in a stack, it transforms an input $\mathbf{h}_t^{(\ell-1)}$ (which plays the role of $\mathbf{z}$ in this derivation) to an output $\mathbf{h}_t^{(\ell)}$ (which plays the role of $\mathbf{u}$). The parameters defining the mixture (initial weights $\pi_k(\text{cond}_t^{(\ell)})$, means $\mathbf{m}_k(\text{cond}_t^{(\ell)})$, and variances $s_k^2(\text{cond}_t^{(\ell)})$) are predicted by a neural network based on a conditioning context $\text{cond}_t^{(\ell)}$.

## B.1  Proof of Proposition 2

Here we provide formal proofs for the normalizing flow layers based on the Mixture of Gaussians (MoG) CDF, as described in Appendices A and B. We establish that these transformations are exact, invertible, and correctly model the target mixture densities.

The core of the $d$-dimensional Mixture-Rosenblatt flow lies in sequentially computing the true conditional probability distribution $p(z_i|\mathbf{z}_{<i})$ for each dimension. The following lemma formally derives this distribution and shows that it takes the form of a 1D Mixture of Gaussians, which is analytically tractable.

**Lemma 1** (Conditional Distribution of an Isotropic MoG). *Let* $\mathbf{z} \in \mathbb{R}^d$ *be a random vector whose probability density function is a mixture of* $V$ *isotropic Gaussians:*

$$p(\mathbf{z}) = \sum_{k=1}^{V} \pi_k \mathcal{N}(\mathbf{z}; \mathbf{m}_k, \sigma_k^2 \mathbf{I}_d),$$

*where* $\sum_k \pi_k = 1$, $\pi_k > 0$. *Then for any dimension* $i \in \{1, \ldots, d\}$, *the true conditional probability density of its component* $z_i$ *given the preceding components* $\mathbf{z}_{<i} = (z_1, \ldots, z_{i-1})$ *is also a 1D Mixture of Gaussians, given by:*

$$p(z_i|\mathbf{z}_{<i}) = \sum_{k=1}^{V} \alpha_k^{(i)}(\mathbf{z}_{<i}) \mathcal{N}(z_i; m_{k,i}, \sigma_k^2),$$

*where* $m_{k,i}$ *is the* $i$-*th element of* $\mathbf{m}_k$, *and the mixture weights* $\alpha_k^{(i)}(\mathbf{z}_{<i})$ *are the posterior probabilities of component membership given* $\mathbf{z}_{<i}$:

$$\alpha_k^{(i)}(\mathbf{z}_{<i}) \triangleq P(K = k|\mathbf{z}_{<i}) = \frac{\pi_k \mathcal{N}(\mathbf{z}_{<i}; \mathbf{m}_{k,<i}, \sigma_k^2 \mathbf{I}_{i-1})}{\sum_{j=1}^{V} \pi_j \mathcal{N}(\mathbf{z}_{<i}; \mathbf{m}_{j,<i}, \sigma_j^2 \mathbf{I}_{i-1})}.$$

*(For the base case* $i = 1$, $\mathbf{z}_{<1}$ *is empty and* $\alpha_k^{(1)} = \pi_k$).

*Proof.* We derive the conditional distribution $p(z_i|\mathbf{z}_{<i})$ by introducing a latent categorical variable $K \in \{1, \ldots, V\}$ that indicates which Gaussian component $\mathbf{z}$ is drawn from. The hierarchical model is:

1. Sample a component index $K = k$ with probability $P(K = k) = \pi_k$.

2. Sample $\mathbf{z}$ from the chosen component: $p(\mathbf{z}|K = k) = \mathcal{N}(\mathbf{z}; \mathbf{m}_k, \sigma_k^2 \mathbf{I}_d)$.

The target conditional density can be found by marginalizing over this latent variable using the law of total probability:

$$p(z_i|\mathbf{z}_{<i}) = \sum_{k=1}^{V} p(z_i|\mathbf{z}_{<i}, K = k) P(K = k|\mathbf{z}_{<i}). \tag{24}$$

We now analyze the two terms in the summation separately.

**1. The Component Density:** $p(z_i|\mathbf{z}_{<i}, K = k)$. This term is the conditional density of $z_i$ given $\mathbf{z}_{<i}$, under the condition that we know the sample originates from component $k$. For a given component $k$, $\mathbf{z}$ follows a single multivariate Gaussian distribution $\mathcal{N}(\mathbf{m}_k, \sigma_k^2 \mathbf{I}_d)$. The covariance matrix $\sigma_k^2 \mathbf{I}_d$ is diagonal, which implies that all dimensions of $\mathbf{z}$ are mutually independent *when conditioned on $K = k$*. Therefore, the value of $z_i$ is independent of the preceding values $\mathbf{z}_{<i}$:

$$p(z_i|\mathbf{z}_{<i}, K = k) = p(z_i|K = k).$$

The distribution $p(z_i|K = k)$ is the $i$-th marginal of the multivariate Gaussian $\mathcal{N}(\mathbf{z}; \mathbf{m}_k, \sigma_k^2 \mathbf{I}_d)$, which is a 1D Gaussian:

$$p(z_i|K = k) = \mathcal{N}(z_i; m_{k,i}, \sigma_k^2).$$

**2. The Mixture Weights:** $P(K = k|\mathbf{z}_{<i})$**.** This term represents the posterior probability of being in component $k$ after observing the first $i - 1$ dimensions. We denote this by $\alpha_k^{(i)}(\mathbf{z}_{<i})$ and compute it using Bayes' rule:

$$P(K = k|\mathbf{z}_{<i}) = \frac{p(\mathbf{z}_{<i}|K = k)P(K = k)}{p(\mathbf{z}_{<i})} = \frac{p(\mathbf{z}_{<i}|K = k)P(K = k)}{\sum_{j=1}^{V} p(\mathbf{z}_{<i}|K = j)P(K = j)}.$$

The required terms are:

- $P(K = k) = \pi_k$ (the prior probability of selecting component $k$).

- $p(\mathbf{z}_{<i}|K = k)$ is the marginal density of the first $i - 1$ dimensions of the $k$-th Gaussian. Due to the isotropic structure, this is $\mathcal{N}(\mathbf{z}_{<i}; \mathbf{m}_{k,<i}, \sigma_k^2 \mathbf{I}_{i-1})$, where $\mathbf{m}_{k,<i}$ contains the first $i - 1$ elements of $\mathbf{m}_k$.

Substituting these into the formula gives the expression for the weights as stated in the lemma.

**Conclusion.** Substituting the derived component density and mixture weights back into Eq. equation 24, we obtain:

$$p(z_i|\mathbf{z}_{<i}) = \sum_{k=1}^{V} \underbrace{\alpha_k^{(i)}(\mathbf{z}_{<i})}_{\text{Mixture Weight}} \cdot \underbrace{\mathcal{N}(z_i; m_{k,i}, \sigma_k^2)}_{\text{1D Gaussian Component}}.$$

This confirms that the true conditional distribution is a 1D Mixture of Gaussians with analytically computable parameters. $\qquad \square$

With the analytical form of the conditional distribution established, we can now state and prove the proposition for the $d$-dimensional flow in a more direct manner.

**Proposition 2.** *Let* $p_{\text{mix}}(\mathbf{z}) = \sum_{k=1}^{V} \pi_k \mathcal{N}(\mathbf{z}; \mathbf{m}_k, \sigma_k^2 \mathbf{I}_d)$ *be a $d$-dimensional isotropic MoG PDF. Let the transformation* $\mathbf{g} : \mathbb{R}^d \to \mathbb{R}^d$ *be defined by the sequential application for* $i = 1, \ldots, d$*:*

$$u_i = g_i(\mathbf{z}) = \Phi^{-1}\big(F_i(z_i|\mathbf{z}_{<i})\big),$$

*where* $F_i(z_i|\mathbf{z}_{<i})$ *is the cumulative distribution function (CDF) of the 1D MoG conditional density* $p(z_i|\mathbf{z}_{<i})$ *as derived in Lemma 1. This transformation is an exact normalizing flow between* $p_{\text{mix}}(\mathbf{z})$ *and the standard $d$-dimensional normal distribution* $\mathcal{N}(\mathbf{u}; \mathbf{0}, \mathbf{I}_d)$*.*

*Proof.* The proof relies on the chain rule of probability, $p_{\text{mix}}(\mathbf{z}) = \prod_{i=1}^{d} p(z_i|\mathbf{z}_{<i})$, and the result of Lemma 1.

**Forward Direction ($\mathbf{Z} \to \mathbf{U}$).** Assume a random vector $\mathbf{Z} \sim p_{\text{mix}}(\mathbf{z})$. We show by induction that the components of the transformed vector $\mathbf{U} = \mathbf{g}(\mathbf{Z})$ are independent and identically distributed as $\mathcal{N}(0, 1)$. By Lemma 1, $F_i(Z_i|\mathbf{Z}_{<i})$ is the true conditional CDF of $Z_i$. By the conditional probability integral transform, the random variable $W_i = F_i(Z_i|\mathbf{Z}_{<i})$ is uniformly distributed on $[0, 1]$ and is independent of the conditioning random vector $\mathbf{Z}_{<i}$. Consequently, $U_i = \Phi^{-1}(W_i)$ is distributed as $\mathcal{N}(0, 1)$ and is also independent of $\mathbf{Z}_{<i}$. Since $(U_1, \ldots, U_{i-1})$ is an invertible function of $\mathbf{Z}_{<i}$, $U_i$ is also independent of $(U_1, \ldots, U_{i-1})$. By induction, all components of $\mathbf{U}$ are i.i.d. $\mathcal{N}(0, 1)$, so $\mathbf{U} \sim \mathcal{N}(\mathbf{0}, \mathbf{I}_d)$.

**Inverse Direction ($\mathbf{U} \to \mathbf{Z}$).** Assume a random vector $\mathbf{U} \sim \mathcal{N}(\mathbf{0}, \mathbf{I}_d)$. The inverse transformation $\mathbf{Z} = \mathbf{g}^{-1}(\mathbf{U})$ is computed by sequentially solving $F_i(Z_i|\mathbf{Z}_{<i}) = \Phi(U_i)$ for $Z_i$. This procedure is equivalent to ancestral sampling from the factorized distribution: for each $i = 1, \ldots, d$, one samples $Z_i$ from the true conditional distribution $p(z_i|\mathbf{Z}_{<i})$ via inverse transform sampling. The joint density of the resulting vector $\mathbf{Z}$ is therefore $\prod_{i=1}^{d} p(z_i|\mathbf{z}_{<i}) = p_{\text{mix}}(\mathbf{z})$.

**Jacobian Determinant.** The Jacobian of $\mathbf{g}$ is lower-triangular, as $u_i$ depends only on $\mathbf{z}_{\leq i}$. Its determinant is the product of the diagonal entries $\frac{\partial u_i}{\partial z_i}$. Treating $\mathbf{z}_{<i}$ as constant, we have:

$$\frac{\partial u_i}{\partial z_i} = \frac{d}{dz_i}\Phi^{-1}\big(F_i(z_i|\mathbf{z}_{<i})\big) = \frac{F_i'(z_i|\mathbf{z}_{<i})}{\phi(u_i)} = \frac{p(z_i|\mathbf{z}_{<i})}{\phi(u_i)},$$

where $\phi$ is the standard normal PDF and we used the fact that the derivative of a CDF is its PDF. The log-determinant is:

$$\log|\det J_{\mathbf{g}}(\mathbf{z})| = \sum_{i=1}^{d} \log\left|\frac{\partial u_i}{\partial z_i}\right| = \sum_{i=1}^{d} \left(\log p(z_i|\mathbf{z}_{<i}) - \log\phi(u_i)\right).$$

Using the chain rule of probability for the first term and the definition of the multivariate normal for the second term, this simplifies to:

$$\log|\det J_{\mathbf{g}}(\mathbf{z})| = \log p_{\mathrm{mix}}(\mathbf{z}) - \log\mathcal{N}(\mathbf{u}; \mathbf{0}, \mathbf{I}_d).$$

This confirms that the change of variables formula is satisfied exactly. The recursive Bayesian update algorithm described in Appendix B is the practical implementation of the analytical derivation in Lemma 1. $\qquad\square$

## B.2  Proof of Proposition 3

We now prove that the multidimensional transformation is a global diffeomorphism, which is a stronger condition than mere invertibility and differentiability, ensuring the transformation is smooth and stable.

**Proposition 3.** *Let* $p_{\mathrm{mix}}(\mathbf{z}) = \sum_{k=1}^{V} \pi_k \mathcal{N}(\mathbf{z}; \mathbf{m}_k, \sigma_k^2 \mathbf{I}_d)$ *be a $d$-dimensional MoG PDF. The Rosenblatt transformation* $\mathbf{g} : \mathbb{R}^d \to \mathbb{R}^d$, *defined sequentially by*

$$u_i = g_i(\mathbf{z}) = \Phi^{-1}\big(F_i(z_i|\mathbf{z}_{<i})\big) \quad for\ i = 1, \ldots, d,$$

*where $F_i$ is the CDF of the true conditional density* $p(z_i|\mathbf{z}_{<i})$, *is a global $C^\infty$-diffeomorphism.*

*Proof.* Our proof strategy is to first show that $\mathbf{g}$ is a local diffeomorphism everywhere on $\mathbb{R}^d$ and then establish that it is also a global bijection. These two properties together imply that $\mathbf{g}$ is a global diffeomorphism.

*Local Diffeomorphism.* A map is a local $C^\infty$-diffeomorphism at a point if it is smooth ($C^\infty$) in a neighborhood of that point and its Jacobian determinant is non-zero at that point.

*(a) Smoothness of $\mathbf{g}$:* The map $\mathbf{g}$ is smooth if each component function $u_i = g_i(\mathbf{z})$ is smooth. The function $g_i = \Phi^{-1} \circ F_i$ is a composition of functions. The conditional density $p(z_i|\mathbf{z}_{<i})$ is a ratio of marginal densities of the MoG. Since marginals of a MoG are themselves finite sums of Gaussian PDFs, they are smooth functions. As the denominator is strictly positive, $p(z_i|\mathbf{z}_{<i})$ is a smooth function of all its arguments, $(z_i, \mathbf{z}_{<i})$. By differentiation under the integral sign, its CDF $F_i(z_i|\mathbf{z}_{<i})$ is also smooth. Since $\Phi^{-1}$ is smooth on its domain, the composition $g_i$ is smooth. Thus, the overall map $\mathbf{g}$ is smooth on $\mathbb{R}^d$.

*(b) Non-Vanishing Jacobian Determinant:* The Jacobian matrix $J_{\mathbf{g}}(\mathbf{z})$ is lower-triangular due to the sequential construction of $\mathbf{g}$. Its determinant is the product of its diagonal entries:

$$\det J_{\mathbf{g}}(\mathbf{z}) = \prod_{i=1}^{d} \frac{\partial u_i}{\partial z_i} = \prod_{i=1}^{d} \frac{p(z_i|\mathbf{z}_{<i})}{\phi(u_i)}.$$

Since both the conditional density $p(z_i|\mathbf{z}_{<i})$ and the standard normal PDF $\phi(u_i)$ are strictly positive everywhere, each diagonal term is strictly positive. Consequently, $\det J_{\mathbf{g}}(\mathbf{z}) > 0$ for all $\mathbf{z} \in \mathbb{R}^d$.

By the Inverse Function Theorem, since $\mathbf{g}$ is smooth and its Jacobian determinant is non-zero everywhere, $\mathbf{g}$ is a local $C^\infty$-diffeomorphism at every point in $\mathbb{R}^d$.

*Global Bijectivity.* We show that $\mathbf{g}$ is a bijection by constructing a unique inverse $\mathbf{z} = \mathbf{g}^{-1}(\mathbf{u})$ for any $\mathbf{u} \in \mathbb{R}^d$. The inverse is found by solving $\Phi(u_i) = F_i(z_i|\mathbf{z}_{<i})$ for each $z_i$ sequentially. For any $i = 1, \ldots, d$, and for any fixed, previously determined context $\mathbf{z}_{<i}$, the function $F_i(\cdot|\mathbf{z}_{<i})$ is a strictly increasing bijection from $\mathbb{R} \to (0, 1)$. Thus, a unique solution $z_i = F_i^{-1}(\Phi(u_i)|\mathbf{z}_{<i})$ exists at each step. This inductive construction yields a unique $\mathbf{z}$ for any $\mathbf{u}$, proving that $\mathbf{g}$ is a global bijection.

A map that is a local diffeomorphism at every point and is also a global bijection is a global diffeomorphism. Having established both properties for $\mathbf{g}$, we conclude that it is a global $C^\infty$-diffeomorphism from $\mathbb{R}^d$ to $\mathbb{R}^d$. This confirms that the transformation is smooth, globally invertible with a smooth inverse, and possesses a well-defined, non-zero Jacobian determinant, making it a robust and theoretically sound building block for normalizing flows. $\qquad\square$

## C   Detailed Derivation of a Flow Layer's Learning Objective

This appendix provides a detailed, step-by-step derivation of the learning objective for a single layer within the normalizing flow prior. The aim is to elucidate how the optimization process shapes the behavior of each transformation in the flow.

We consider a generic invertible transformation $f_t^{(\ell)} : \mathbb{R}^d \to \mathbb{R}^d$ representing a single layer $\ell$ at time step $t$ in the flow. Let $\mathbf{x} = \mathbf{h}_t^{(\ell-1)}$ be the input to this layer and $\mathbf{y} = \mathbf{h}_t^{(\ell)}$ be its output, so $\mathbf{y} = f_t^{(\ell)}(\mathbf{x}; \text{cond}_t^{(\ell)})$. The function $f_t^{(\ell)}$ is parameterized, and its parameters (and thus its behavior) are learned based on a conditioning context $\text{cond}_t^{(\ell)}$. For brevity in the derivations that follow, we will often drop the explicit indices $(t, \ell)$ and the conditioning context, writing $f(\mathbf{x})$, $p_{\text{model}}(\mathbf{x})$, etc., with the understanding that these are specific to a layer and its context.

**The Log-Determinant of the Jacobian**   The change of variables formula relates the probability density of $\mathbf{x}$ to the probability density of $\mathbf{y}$:

$$p_X(\mathbf{x}) = p_Y(f(\mathbf{x})) \left| \det J_f(\mathbf{x}) \right| \tag{25}$$

where $p_X$ is the density of $\mathbf{x}$, $p_Y$ is the density of $\mathbf{y}$, and $J_f(\mathbf{x})$ is the Jacobian matrix of $f$ evaluated at $\mathbf{x}$.

Normalizing flow layers like those discussed in Sections 3.2.1 and 3.2.2 (e.g., Mixture-of-CDFs or transformations based on $d$-dimensional mixtures) are constructed such that they define a conditional model density for their input. Let $p_{\text{model}}(\mathbf{x}|\text{cond})$ be this parameterized density that the layer learns for its input $\mathbf{x}$. The transformation $f$ is designed such that if $\mathbf{x}$ were drawn from $p_{\text{model}}(\mathbf{x}|\text{cond})$, its output $\mathbf{y} = f(\mathbf{x})$ would be drawn from a simpler, fixed target density, $p_{\text{target}}(\mathbf{y})$ (typically a standard Gaussian). Substituting $p_X(\mathbf{x}) = p_{\text{model}}(\mathbf{x}|\text{cond})$ and $p_Y(\mathbf{y}) = p_{\text{target}}(\mathbf{y})$ into Eq. equation 25, we get:

$$p_{\text{model}}(\mathbf{x}|\text{cond}) = p_{\text{target}}(f(\mathbf{x})) \left| \det J_f(\mathbf{x}) \right| \tag{26}$$

Rearranging and taking the logarithm gives the expression for the log absolute Jacobian determinant:

$$\log \left| \det J_f(\mathbf{x}) \right| = \log p_{\text{model}}(\mathbf{x}|\text{cond}) - \log p_{\text{target}}(f(\mathbf{x})) \tag{27}$$

This equation matches the form of equation 9 in the main text. It shows that the log-determinant term, crucial for computing the density of $\mathbf{z}_{1:T}$, is composed of the log-likelihood of the input $\mathbf{x}$ under the layer's learned model $p_{\text{model}}$ and the log-likelihood of the output $\mathbf{y} = f(\mathbf{x})$ under the fixed target density $p_{\text{target}}$.

**The Expected Log-Jacobian in the ELBO**   The parameters of the entire model, including those defining $p_{\text{model}}(\mathbf{x}|\text{cond})$ for each layer, are learned by maximizing the Evidence Lower Bound (ELBO), as shown in Eq. equation 3. The ELBO includes the term $\mathbb{E}_{\mathbf{z}_{1:T} \sim q(\cdot|\mathbf{x}_{1:T})}[\log p(\mathbf{z}_{1:T})]$. The log-prior $\log p(\mathbf{z}_{1:T})$ is given by a sum of log-base-density terms and log-determinant terms from each layer:

$$\log p(\mathbf{z}_{1:T}) = \sum_{t=1}^{T} \left( \log p_{\text{base}}(\mathbf{u}_t) + \sum_{\ell=1}^{L} \log \left| \det J_{f_t^{(\ell)}}(\mathbf{h}_t^{(\ell-1)}) \right| \right) \tag{28}$$

Let $q_{\text{in}}(\mathbf{x})$ denote the actual distribution of the input $\mathbf{x} = \mathbf{h}_t^{(\ell-1)}$ to a specific layer $f \equiv f_t^{(\ell)}$. This distribution $q_{\text{in}}(\mathbf{x})$ is induced by the input data $\mathbf{x}_{1:T}$, the encoder $q(\mathbf{z}_{1:T}|\mathbf{x}_{1:T})$, and all preceding flow transformations $f^{(1)}, \ldots, f^{(\ell-1)}$. The ELBO maximization objective, with respect to the contribution of this layer's Jacobian to the prior, involves maximizing:

$$\mathcal{J}_{\text{layer}} = \mathbb{E}_{q_{\text{in}}(\mathbf{x})} \left[ \log \left| \det J_f(\mathbf{x}) \right| \right] \tag{29}$$

Substituting Eq. equation 27:

$$\mathcal{J}_{\text{layer}} = \mathbb{E}_{q_{\text{in}}(\mathbf{x})} \left[ \log p_{\text{model}}(\mathbf{x}|\text{cond}) - \log p_{\text{target}}(f(\mathbf{x})) \right] \tag{30}$$

$$= \mathbb{E}_{q_{\text{in}}(\mathbf{x})} \left[ \log p_{\text{model}}(\mathbf{x}|\text{cond}) \right] - \mathbb{E}_{q_{\text{in}}(\mathbf{x})} \left[ \log p_{\text{target}}(f(\mathbf{x})) \right] \tag{31}$$

Let $q_{\text{out}}(\mathbf{y})$ be the actual distribution of the output $\mathbf{y} = f(\mathbf{x})$ when $\mathbf{x} \sim q_{\text{in}}(\mathbf{x})$. The second term in Eq. equation 31 can be rewritten using this definition:

$$\mathbb{E}_{q_{\text{in}}(\mathbf{x})} \left[ \log p_{\text{target}}(f(\mathbf{x})) \right] = \mathbb{E}_{q_{\text{out}}(\mathbf{y})} \left[ \log p_{\text{target}}(\mathbf{y}) \right] \tag{32}$$

So, Eq. equation 31 becomes:

$$\mathcal{J}_{\text{layer}} = \mathbb{E}_{q_{\text{in}}(\mathbf{x})} \left[ \log p_{\text{model}}(\mathbf{x}|\text{cond}) \right] - \mathbb{E}_{q_{\text{out}}(\mathbf{y})} \left[ \log p_{\text{target}}(\mathbf{y}) \right] \tag{33}$$

**Relating to KL Divergence and Differential Entropy**   We use the definitions of differential entropy $H(q)$ and Kullback-Leibler (KL) divergence $\text{KL}(q\|p)$:

$$H(q) = -\mathbb{E}_{q(\mathbf{z})}[\log q(\mathbf{z})] = -\int q(\mathbf{z}) \log q(\mathbf{z}) d\mathbf{z} \tag{34}$$

$$\text{KL}(q\|p) = \mathbb{E}_{q(\mathbf{z})}\left[\log \frac{q(\mathbf{z})}{p(\mathbf{z})}\right] = \int q(\mathbf{z}) \log \frac{q(\mathbf{z})}{p(\mathbf{z})} d\mathbf{z} \tag{35}$$

From the definition of KL divergence, we have $\text{KL}(q\|p) = \mathbb{E}_{q(\mathbf{z})}[\log q(\mathbf{z})] - \mathbb{E}_{q(\mathbf{z})}[\log p(\mathbf{z})] = -H(q) - \mathbb{E}_{q(\mathbf{z})}[\log p(\mathbf{z})]$. Rearranging this gives a useful identity for an expected log-likelihood term:

$$\mathbb{E}_{q(\mathbf{z})}[\log p(\mathbf{z})] = -H(q) - \text{KL}(q\|p) \tag{36}$$

Applying Eq. equation 36 to the two terms in Eq. equation 33:

$$\mathbb{E}_{q_{\text{in}}(\mathbf{x})}[\log p_{\text{model}}(\mathbf{x}|\text{cond})] = -H(q_{\text{in}}) - \text{KL}(q_{\text{in}}\|p_{\text{model}}) \tag{37}$$

$$\mathbb{E}_{q_{\text{out}}(\mathbf{y})}[\log p_{\text{target}}(\mathbf{y})] = -H(q_{\text{out}}) - \text{KL}(q_{\text{out}}\|p_{\text{target}}) \tag{38}$$

Substituting these back into the expression for $\mathcal{J}_{\text{layer}}$:

$$\mathcal{J}_{\text{layer}} = [-H(q_{\text{in}}) - \text{KL}(q_{\text{in}}\|p_{\text{model}})] - [-H(q_{\text{out}}) - \text{KL}(q_{\text{out}}\|p_{\text{target}})] \tag{39}$$

$$= H(q_{\text{out}}) - H(q_{\text{in}}) - \text{KL}(q_{\text{in}}\|p_{\text{model}}) + \text{KL}(q_{\text{out}}\|p_{\text{target}}) \tag{40}$$

This matches the main text.

**The KL Divergence Identity for Normalizing Flows**   A key property of normalizing flows is the invariance of KL divergence under invertible transformations, specifically relating the KL divergence at the input of a layer to the KL divergence at its output. We aim to show:

$$\text{KL}(q_{\text{in}}\|p_{\text{model}}) = \text{KL}(q_{\text{out}}\|p_{\text{target}}) \tag{41}$$

Recall that $q_{\text{in}}(\mathbf{x})$ is the actual density of the layer's input and $q_{\text{out}}(\mathbf{y})$ is the actual density of the layer's output, where $\mathbf{y} = f(\mathbf{x})$. These are related by the change of variables formula:

$$q_{\text{in}}(\mathbf{x}) = q_{\text{out}}(f(\mathbf{x})) |\det J_f(\mathbf{x})| \tag{42}$$

Similarly, $p_{\text{model}}(\mathbf{x}|\text{cond})$ is the density parameterized by the layer for its input, and $p_{\text{target}}(\mathbf{y})$ is the fixed target density for the output. The transformation $f$ is constructed such that these are also related by the change of variables formula:

$$p_{\text{model}}(\mathbf{x}|\text{cond}) = p_{\text{target}}(f(\mathbf{x})) |\det J_f(\mathbf{x})| \tag{43}$$

Now, consider the ratio $\frac{q_{\text{in}}(\mathbf{x})}{p_{\text{model}}(\mathbf{x}|\text{cond})}$:

$$\frac{q_{\text{in}}(\mathbf{x})}{p_{\text{model}}(\mathbf{x}|\text{cond})} = \frac{q_{\text{out}}(f(\mathbf{x})) |\det J_f(\mathbf{x})|}{p_{\text{target}}(f(\mathbf{x})) |\det J_f(\mathbf{x})|} = \frac{q_{\text{out}}(f(\mathbf{x}))}{p_{\text{target}}(f(\mathbf{x}))} \tag{44}$$

The KL divergence $\text{KL}(q_{\text{in}}\|p_{\text{model}})$ is defined as:

$$\text{KL}(q_{\text{in}}\|p_{\text{model}}) = \int q_{\text{in}}(\mathbf{x}) \log \frac{q_{\text{in}}(\mathbf{x})}{p_{\text{model}}(\mathbf{x}|\text{cond})} d\mathbf{x} \tag{45}$$

Substitute Eq. equation 44 into the logarithm:

$$\text{KL}(q_{\text{in}}\|p_{\text{model}}) = \int q_{\text{in}}(\mathbf{x}) \log \frac{q_{\text{out}}(f(\mathbf{x}))}{p_{\text{target}}(f(\mathbf{x}))} d\mathbf{x} \tag{46}$$

Now, perform a change of variables in the integration from $\mathbf{x}$ to $\mathbf{y} = f(\mathbf{x})$. We have $\mathbf{x} = f^{-1}(\mathbf{y})$, and $d\mathbf{x} = |\det J_{f^{-1}}(\mathbf{y})| d\mathbf{y}$. The integral becomes:

$$\text{KL}(q_{\text{in}}\|p_{\text{model}}) = \int q_{\text{in}}(f^{-1}(\mathbf{y})) \log \frac{q_{\text{out}}(\mathbf{y})}{p_{\text{target}}(\mathbf{y})} |\det J_{f^{-1}}(\mathbf{y})| d\mathbf{y} \tag{47}$$

From the change of variables for $q_{\text{in}}$ (Eq. equation 42, rearranged for $f^{-1}$), we know that $q_{\text{in}}(f^{-1}(\mathbf{y})) |\det J_{f^{-1}}(\mathbf{y})| = q_{\text{out}}(\mathbf{y})$. Substituting this into the integral:

$$\text{KL}(q_{\text{in}}\|p_{\text{model}}) = \int q_{\text{out}}(\mathbf{y}) \log \frac{q_{\text{out}}(\mathbf{y})}{p_{\text{target}}(\mathbf{y})} d\mathbf{y} \tag{48}$$

The right-hand side is, by definition, $\text{KL}(q_{\text{out}}\|p_{\text{target}})$. Thus, the identity in Eq. equation 41 is established.

**Final Expression for Expected Log-Jacobian**  Substituting the KL divergence identity (Eq. equation 41) into the expanded expression for $\mathcal{J}_{\text{layer}}$ (Eq. equation 40):

$$\mathcal{J}_{\text{layer}} = H(q_{\text{out}}) - H(q_{\text{in}}) - \text{KL}(q_{\text{in}}\|p_{\text{model}}) + \text{KL}(q_{\text{in}}\|p_{\text{model}}) \tag{49}$$

$$= H(q_{\text{out}}) - H(q_{\text{in}}) \tag{50}$$

This confirms the results in the main text: the expected log-determinant of the Jacobian for an invertible transformation is the change in differential entropy from its input distribution to its output distribution.

**Interpretation of the Learning Objective for a Single Layer**  The overall ELBO maximization drives the learning of parameters $\boldsymbol{\theta}$, which include the parameters defining $p_{\text{model}}(\mathbf{x}|\text{cond})$ for each layer. The contribution of a single layer's Jacobian to the ELBO is $\mathcal{J}_{\text{layer}} = H(q_{\text{out}}) - H(q_{\text{in}})$.

However, the parameters of the current layer $f$ directly influence $p_{\text{model}}(\mathbf{x}|\text{cond})$. To maximize $\mathcal{J}_{\text{layer}}$, consider its expression before the KL terms cancel: $\mathcal{J}_{\text{layer}} = H(q_{\text{out}}) - H(q_{\text{in}}) - \text{KL}(q_{\text{in}}\|p_{\text{model}}) + \text{KL}(q_{\text{out}}\|p_{\text{target}})$. Using the identity $\text{KL}(q_{\text{in}}\|p_{\text{model}}) = \text{KL}(q_{\text{out}}\|p_{\text{target}})$, this is also: $\mathcal{J}_{\text{layer}} = \mathbb{E}_{q_{\text{in}}(\mathbf{x})}\left[\log p_{\text{model}}(\mathbf{x}|\text{cond})\right] - \mathbb{E}_{q_{\text{out}}(\mathbf{y})}\left[\log p_{\text{target}}(\mathbf{y})\right]$.

The optimization process adjusts the parameters of $f$ (which define $p_{\text{model}}$) to maximize this quantity. Maximizing $\mathbb{E}_{q_{\text{in}}(\mathbf{x})}\left[\log p_{\text{model}}(\mathbf{x}|\text{cond})\right]$ is equivalent to minimizing $\text{KL}(q_{\text{in}}\|p_{\text{model}})$ (since $-H(q_{\text{in}})$ does not depend on the parameters of the current layer $f$, but rather on the data and preceding layers/encoder). When $\text{KL}(q_{\text{in}}\|p_{\text{model}})$ is minimized, $p_{\text{model}}(\mathbf{x}|\text{cond})$ becomes a good approximation of the actual input distribution $q_{\text{in}}(\mathbf{x})$.

Due to the identity $\text{KL}(q_{\text{in}}\|p_{\text{model}}) = \text{KL}(q_{\text{out}}\|p_{\text{target}})$, minimizing $\text{KL}(q_{\text{in}}\|p_{\text{model}})$ simultaneously implies minimizing $\text{KL}(q_{\text{out}}\|p_{\text{target}})$. This means that the actual output distribution $q_{\text{out}}(\mathbf{y})$ is driven to match the fixed target distribution $p_{\text{target}}(\mathbf{y})$.

In summary, each flow layer learns to:

1. **Model its input distribution**: The layer's parameters (defining $p_{\text{model}}(\mathbf{x}|\text{cond})$) are adjusted so that $p_{\text{model}}(\mathbf{x}|\text{cond})$ accurately represents the distribution $q_{\text{in}}(\mathbf{x})$ of the data it receives. This corresponds to minimizing $\text{KL}(q_{\text{in}}\|p_{\text{model}})$.

2. **Transform its input to a target distribution**: As a consequence of (1) and the construction of $f$, the layer transforms its input $\mathbf{x}$ into an output $\mathbf{y}$ whose distribution $q_{\text{out}}(\mathbf{y})$ closely matches the predefined target distribution $p_{\text{target}}(\mathbf{y})$. This corresponds to minimizing $\text{KL}(q_{\text{out}}\|p_{\text{target}})$.

This step-by-step process, repeated through the stack of flow layers, allows the overall normalizing flow to transform a complex data distribution into a simple base distribution (e.g., standard Gaussian), while enabling exact density calculation.

# D  1-D Mixture-of-Gaussian and Mixture-of-Logistics Coupling

The analysis in the main text shows that an AR MoG probability distribution can be interpreted as an implicit normalizing flow layer. This layer utilizes the cumulative distribution function (CDF) of the Gaussian mixture to map between the data space $\mathbf{u}_t$ and a base space (implicitly standard normal). Similar transformations, built explicitly around mixture CDFs, are employed in various normalizing flow architectures. This section provides a detailed comparison between the transformation implicitly defined by our model and common explicit parameterizations, establishing their functional equivalence. We will consistently analyze the transformations in the **data-to-latent** direction for clarity, and then briefly discuss the inverse (latent-to-data) direction.

For conciseness within this section, we may drop the time index $t$ and dimension index $i$ from variables and parameters when the context is clear (e.g., using $x$ for a single data dimension, $\epsilon$ or $w$ for the corresponding latent dimension, and $\pi_k, \mu_k, \sigma_k$ for the mixture parameters predicted based on history). We maintain the standard normal PDF $\varphi(\cdot)$ and CDF $\Phi(\cdot)$, and the standard logistic CDF $\sigma(r) = 1/(1 + e^{-r})$ and its inverse, the logit function $\sigma^{-1}(p) = \log(p/(1-p))$.

**The Core Component: Gaussian Mixture CDF.**  Both the implicit and explicit layers rely on the same core component: the CDF of the Gaussian mixture distribution. For a single data dimension $x$,

the mixture CDF $F_{\text{mix}}(x)$, conditioned on history (parameters $\pi_k, \mu_k, \sigma_k$), is:

$$F_{\text{mix}}(x) = \sum_{k=1}^{V'} \pi_k \Phi \left( \frac{x - \mu_k}{\sigma_k} \right) \tag{51}$$

The derivative of this CDF with respect to $x$ is the probability density function (PDF) of the mixture:

$$m(x) = \frac{dF_{\text{mix}}(x)}{dx} = \sum_{k=1}^{V'} \pi_k \varphi \left( \frac{x - \mu_k}{\sigma_k} \right) \frac{1}{\sigma_k} \tag{52}$$

This mixture PDF $m(x)$ plays a key role in the Jacobian determinant calculations.

**Layer Definitions and Jacobians (Data-to-Latent Direction).** We now define two primary variants of the mixture CDF transformation layer, both mapping the data variable $x$ to a latent variable, but differing in the choice of outer non-linearity and corresponding base distribution.

**MoG Variant: ProbitMixtureCDF** This layer corresponds to the transformation $g_{t,i}$ identified in Section Sec. 3.2.2. It maps the data $x$ to a latent variable $z$, assumed to follow a standard normal distribution $\mathcal{N}(0, 1)$.

$$z = \Phi^{-1}(F_{\text{mix}}(x)) \tag{53}$$

Its Jacobian determinant, $\partial z / \partial x$, is calculated using the chain rule and properties of $\Phi^{-1}$ and $F_{\text{mix}}$:

$$\frac{\partial z}{\partial x} = \frac{m(x)}{\varphi(z)} \tag{54}$$

The log-Jacobian determinant is therefore:

$$\log \left| \frac{\partial z}{\partial x} \right| = \log m(x) - \log \varphi(z) \tag{55}$$

**MoL Variant: LogitMixtureCDF** Explicit flow layers often use the logit function $\sigma^{-1}$ as the outer non-linearity, mapping the probability $F_{\text{mix}}(x)$ to a latent variable $w$, typically assumed to follow a standard logistic distribution. We consider the core transformation without additional affine terms for now:

$$w = \sigma^{-1}(F_{\text{mix}}(x)) \tag{56}$$

Its Jacobian determinant, $\partial w / \partial x$, is calculated using the chain rule. The derivative of $\sigma^{-1}(p)$ is $1/(p(1-p))$. The PDF of the standard logistic distribution is $p_{\text{logistic}}(w) = \sigma'(w) = \sigma(w)(1 - \sigma(w))$. Since $F_{\text{mix}}(x) = \sigma(w)$, we have $F_{\text{mix}}(x)(1 - F_{\text{mix}}(x)) = p_{\text{logistic}}(w)$.

$$\frac{\partial w}{\partial x} = \frac{d\sigma^{-1}(p)}{dp} \bigg|_{p=F_{\text{mix}}(x)} \cdot \frac{dF_{\text{mix}}(x)}{dx} = \frac{1}{F_{\text{mix}}(x)(1 - F_{\text{mix}}(x))} \cdot m(x) = \frac{m(x)}{p_{\text{logistic}}(w)} \tag{57}$$

The log-Jacobian determinant is therefore:

$$\log \left| \frac{\partial w}{\partial x} \right| = \log m(x) - \log p_{\text{logistic}}(w) \tag{58}$$

**Comparison in the Data-to-Latent Direction.** Table 4 summarizes the two variants when viewed in the data-to-latent direction.

Table 4: Comparison of Mixture CDF Flow Variants (Data $x \to$ Latent).

| Layer Variant | Transformation | Latent Base Dist. | Log-Jacobian $\log \lvert \partial(\text{latent})/\partial x \rvert$ |
|---|---|---|---|
| ProbitMixtureCDF | $z = \Phi^{-1}(F_{\text{mix}}(x))$ | $\mathcal{N}(0,1)$ | $\log m(x) - \log \varphi(z)$ |
| LogitMixtureCDF (core) | $w = \sigma^{-1}(F_{\text{mix}}(x))$ | Logistic(0,1) | $\log m(x) - \log p_{\text{logistic}}(w)$ |

Both transformations share the same initial structure involving $F_{\text{mix}}(x)$ and yield a log-Jacobian determinant of the form $\log m(x) - \log p_{\text{base}}(\text{latent})$, where $p_{\text{base}}$ is the PDF of the corresponding base distribution (Gaussian or Logistic). The difference lies solely in the choice of the outer bijection ($\Phi^{-1}$ vs $\sigma^{-1}$) and the associated base distribution.

**Connecting the Variants via Re-parameterization.** The Probit and Logit variants are functionally equivalent because the standard normal and standard logistic distributions can be transformed into one another via a simple, smooth, invertible function. Define the map $g : \mathbb{R} \to \mathbb{R}$ that converts a standard logistic variable $w$ into a standard normal variable $z$:

$$z = g(w) = \Phi^{-1}(\sigma(w)) \tag{59}$$

The inverse map $g^{-1} : \mathbb{R} \to \mathbb{R}$ converts a standard normal variable $z$ into a standard logistic variable $w$:

$$w = g^{-1}(z) = \sigma^{-1}(\Phi(z)) \tag{60}$$

These functions act as translators between the two latent spaces.

We can demonstrate the equivalence by composing one layer type with the appropriate function $g$ or $g^{-1}$:

- **LogitMixtureCDF followed by $g$**: If we take the output $w$ from the LogitMixtureCDF layer (Eq. equation 56) and apply $g$, we get:

$$g(w) = g(\sigma^{-1}(F_{\text{mix}}(x))) = \Phi^{-1}(\sigma(\sigma^{-1}(F_{\text{mix}}(x)))) = \Phi^{-1}(F_{\text{mix}}(x)) \tag{61}$$

This result is exactly the output $z$ of the ProbitMixtureCDF layer (Eq. equation 53).

- **ProbitMixtureCDF followed by $g^{-1}$**: If we take the output $z$ from the ProbitMixtureCDF layer (Eq. equation 53) and apply $g^{-1}$, we get:

$$g^{-1}(z) = g^{-1}(\Phi^{-1}(F_{\text{mix}}(x))) = \sigma^{-1}(\Phi(\Phi^{-1}(F_{\text{mix}}(x)))) = \sigma^{-1}(F_{\text{mix}}(x)) \tag{62}$$

This result is exactly the output $w$ of the LogitMixtureCDF layer (Eq. equation 56).

This shows that the two layer types are related by composition with the fixed, invertible function $g$. In the context of normalizing flows, composing with such a function simply constitutes another valid flow layer, effectively re-parameterizing the latent space without changing the transformation's overall expressive power on the data $x$.

**Likelihood Equivalence.** The equivalence is further confirmed by examining the data log-likelihood $\log p(x)$ induced by each transformation. Using the change of variables formula for the data-to-latent direction $z = T^{-1}(x)$: $\log p(x) = \log p_{\text{base}}(z) + \log |\det J_{T^{-1}}(x)|$.

- **ProbitMixtureCDF:**

$$\log p(x) = \log p_{\text{gaussian}}(z) + \log \left| \frac{\partial z}{\partial x} \right|$$
$$= \log \varphi(z) + (\log m(x) - \log \varphi(z)) = \log m(x) \tag{63}$$

- **LogitMixtureCDF:**

$$\log p(x) = \log p_{\text{logistic}}(w) + \log \left| \frac{\partial w}{\partial x} \right|$$
$$= \log p_{\text{logistic}}(w) + (\log m(x) - \log p_{\text{logistic}}(w)) = \log m(x) \tag{64}$$

Both formulations yield $\log p(x) = \log m(x)$, the log-PDF of the Gaussian mixture model defining the transformation. The contribution from the base density's log-PDF always exactly cancels the second term in the layer's log-Jacobian determinant. This confirms that both variants define the *exact same probability distribution* over the data $x$, given the same mixture parameters $(\pi_k, \mu_k, \sigma_k)$. Consequently, the gradients with respect to these parameters during training will be identical.

**Handling Affine Transformations.** Explicit layers like the GaussMixCDF snippet often include an affine transformation $y = e^a w + b$ applied after the core logit transformation $w = \sigma^{-1}(F_{\text{mix}}(x))$. This affine transformation is itself an invertible flow layer with a simple constant log-Jacobian determinant:

$$\log \left| \frac{\partial y}{\partial w} \right| = \log |e^a| = a \tag{65}$$

This term simply adds $a$ to the total log-determinant of the composed flow ($x \to w \to y$). Such affine transformations do not change the core functionality related to the mixture CDF and can be treated as separate layers (like Batch Normalization or ActNorm layers) or fused for implementation. Their presence or absence does not affect the fundamental equivalence discussed above.

**Latent-to-Data Direction.** For completeness, we consider the inverse (generative) direction, mapping the latent variable back to the data $x$.

*MoG Variant: ProbitMixtureCDF Inverse* This corresponds to the transformation $h$:

$$x = h(z) = F_{\text{mix}}^{-1}(\Phi(z)) \tag{66}$$

The log-Jacobian is the negative of the forward log-Jacobian (Eq. equation 55):

$$\log \left| \frac{\partial x}{\partial z} \right| = -(\log m(x) - \log \varphi(z)) = \log \varphi(z) - \log m(x) \tag{67}$$

*MoL Variant: LogitMixtureCDF Inverse* Inverting $w = \sigma^{-1}(F_{\text{mix}}(x))$ gives $F_{\text{mix}}(x) = \sigma(w)$, so:

$$x = F_{\text{mix}}^{-1}(\sigma(w)) \tag{68}$$

The log-Jacobian is the negative of the forward log-Jacobian (Eq. equation 58):

$$\log \left| \frac{\partial x}{\partial w} \right| = -(\log m(x) - \log p_{\text{logistic}}(w)) = \log p_{\text{logistic}}(w) - \log m(x) \tag{69}$$

Table 5 summarizes the inverse transformations.

Table 5: Comparison of Mixture CDF Flow Variants (Latent $\rightarrow$ Data $x$).

| Layer Variant | Transformation | Latent Base Dist. | Log-Jacobian $\log \lvert \partial x / \partial (\text{latent}) \rvert$ |
|---|---|---|---|
| ProbitMixtureCDF Inv. | $x = F_{\text{mix}}^{-1}(\Phi(z))$ | $\mathcal{N}(0,1)$ | $\log \varphi(z) - \log m(x)$ |
| LogitMixtureCDF Inv. | $x = F_{\text{mix}}^{-1}(\sigma(w))$ | Logistic(0,1) | $\log p_{\text{logistic}}(w) - \log m(x)$ |

The transformation implicitly defined by our AR-MDN base prior is functionally equivalent to explicit Gaussian Mixture CDF flow layers found in the literature. Both leverage the mixture CDF $F_{\text{mix}}$ to perform a complex, data-dependent monotonic transformation. Differences in the choice of outer bijection (probit vs. logit) and associated base distributions (Gaussian vs. logistic), as well as optional affine components, amount to implementation choices or re-parameterizations connected by the simple diffeomorphism $g$. They represent the same class of transformations and induce the same probability density on the data $x$ for identical mixture parameters. Our VAE approach benefits from this expressive power implicitly by optimizing the mixture density $p(\mathbf{u}_t|\mathbf{u}_{<t}) = m(\mathbf{u}_t)$ directly, avoiding the potentially costly computation of $F_{\text{mix}}^{-1}$, $\Phi^{-1}$, or $\sigma^{-1}$ during training.

# E   Additional Experimental Details

This section provides supplementary experimental details and results that support the findings presented in the main body of the paper.

**Generative evaluation.** For MAUVE, we generate 5,000 samples for each configuration. We use GPT-2 Large as the embedding model, and set the MAUVE scaling hyperparameter to 5. For generative perplexity, we also use GPT-2 Large as the external language model to evaluate generated texts. We additionally report the average per-sequence entropy as a simple diversity metric.

Fig. 8 offers a more complete view of the training and validation loss curves, expanding on the comparison shown in Fig. 3 of the main text. These curves illustrate the learning progression for different model configurations. The results underscore the benefit of incorporating mixture-based coupling layers, which exhibit improved convergence and final performance (lower negative ELBO) compared to configurations relying solely on standard affine coupling layers. This observation reinforces the point made in the main text regarding the importance of mixture couplings for effectively modeling discrete text data within our continuous latent space framework.

To further clarify the mechanics of our proposed model variants, Alg. 1 details the forward transformation process for the $d$-dimensional mixture flow layer. This transformation is a core component of the 'Mix-d' model configuration (discussed in Section 3.2.2), enabling the direct modeling of token-wise dependencies in a $d$-dimensional latent space using a mixture of Gaussians. The algorithm

| Method | Timesteps | Gen PPL ↓ | MAUVE ↑ | Entropy ↑ |
|---|---|---|---|---|
| **Data** | – | 14.7 | 1.0 | 5.44 |
| **Autoregressive** | 1024 | 12.1 | 0.76 | 5.22 |
| SUNDAE | 200 | 34.7 | – | 5.2 |
| Ssd-LM | >10000 | 99.2 | – | 4.8 |
| D3PM Absorb | 1024 | 842.3 | – | 7.6 |
| SEDD | 256/512/1024/2048 | 110.1 / 107.2 / 104.7 / 103.2 | 0.007 / 0.008 / 0.008 / 0.008 | 5.63 / 5.62 / 5.62 / 5.61 |
| MDLM | 256/512/1024/2048 | 55.8 / 53.0 / 51.3 / 51.3 | 0.023 / 0.031 / 0.042 / 0.037 | 5.49 / 5.48 / 5.46 / 5.46 |
| **TarFlowLM single-dim CDF (Ours)** | – | 20.7 | 0.47 | 5.58 |
| **TarFlowLM multi-dim Rosenblatt (Ours)** | – | 14.3 | 0.68 | 5.13 |

Table 6: Generative evaluation comparing generation quality and diversity. We report generative perplexity (Gen PPL; lower is better), MAUVE (higher is better), and average sequence entropy (higher indicates more diversity). Some baseline numbers are from [68, 71].

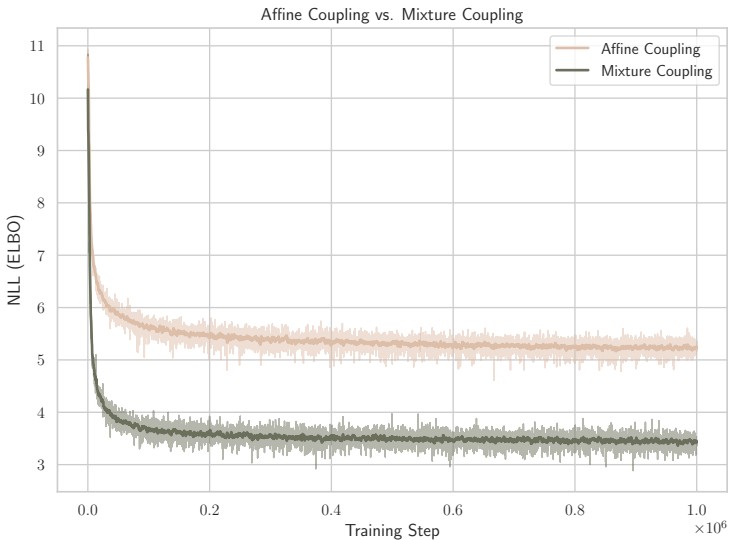

Figure 8: Comparison of training and validation loss curves across different model configurations. Our mixture-based coupling layers demonstrate improved convergence compared to standard affine coupling.

outlines how an input vector $\mathbf{z}$ is mapped to an output vector $\mathbf{u}$ that follows a standard Gaussian distribution, through a sequence of conditional 1D mixture-CDF transformations.

Alg. 2 describes the channel mixing and unmixing operations. These procedures are employed in the context of block-wise multi-token generation, as discussed in the "Flexible Patch Size" paragraph of the experiments section. By permuting latent dimensions between flow layers, channel mixing facilitates information exchange across different token positions within a patch, allowing the model to capture intra-patch dependencies more effectively as data propagates through the stack of flow transformations. The ChannelUnmix procedure is the exact inverse, ensuring the overall transformation remains bijective.

## F Connection to Discrete Autoregressive Language Models

A specific configuration of our continuous latent space framework reveals a direct relationship with conventional discrete autoregressive language models. This connection arises when we consider the model structure outlined in Section 3 under particular assumptions for the prior $p(\mathbf{z}_{1:T})$ and the encoder/decoder.

Specifically, we focus on the token-wise autoregressive prior for $p(\mathbf{z}_{1:T})$ as factorized in Section 3.2, where each conditional $p(\mathbf{z}_t|\mathbf{z}_{<t})$ is modeled as a $d$-dimensional mixture of Gaussians. We adopt the formulation from Eq. equation 7, where a shared codebook of Gaussian components is used. The key assumptions for this connection are: **Tied Prior and Encoder Components:** The $V$ Gaussian components $\{\boldsymbol{\mu}_k^{\text{prior}}, (\sigma_k^{\text{prior}})^2\}_{k=1}^V$ used in the prior $p(\mathbf{z}_t|\mathbf{z}_{<t})$ are identical to the VAE

---

**Algorithm 1** Forward Transformation for $d$-Dimensional Mixture Flow Layer: $\mathbf{u} = \mathbf{g}_d(\mathbf{z}; \mathcal{C})$

---

1: **Input:** Data $\mathbf{z} = (z_1, \ldots, z_d) \in \mathbb{R}^d$; context $\mathcal{C}$; mixture parameters $\{\pi_j(\mathcal{C}), \mathbf{m}_j(\mathcal{C}), s_j^2(\mathcal{C})\}_{j=1}^K$.
2: **Output:** Transformed variable $\mathbf{u} = (u_1, \ldots, u_d) \in \mathbb{R}^d$.
3: Initialize weights: $\alpha_j^{(1)} \leftarrow \pi_j(\mathcal{C})$ for $j = 1, \ldots, K$.
4: **for** $i = 1, \ldots, d$ **do**
5:     $m_{j,i} \leftarrow i$-th component of $\mathbf{m}_j(\mathcal{C})$; $s_j \leftarrow \sqrt{s_j^2(\mathcal{C})}$
6:     $p_i(z_i|\mathcal{C}, \mathbf{z}_{<i}) \leftarrow \sum_{j=1}^K \alpha_j^{(i)} \mathcal{N}(z_i; m_{j,i}, s_j^2)$
7:     $F_i(z_i|\mathcal{C}, \mathbf{z}_{<i}) \leftarrow \sum_{j=1}^K \alpha_j^{(i)} \Phi\left(\frac{z_i - m_{j,i}}{s_j}\right)$
8:     $u_i \leftarrow \Phi^{-1}(F_i(z_i|\mathcal{C}, \mathbf{z}_{<i}))$
9:     **if** $i < d$ **then**
10:         $\alpha_j^{(i+1)} \leftarrow \frac{\alpha_j^{(i)} \mathcal{N}(z_i; m_{j,i}, s_j^2)}{p_i(z_i|\mathcal{C}, \mathbf{z}_{<i})}$ for $j = 1, \ldots, K$.
11: **Return u**

---

**Algorithm 2** Channel Mixing and Unmixing Operations

---

1: **procedure** CHANNELMIX($\mathbf{X}$)
2:     $D \leftarrow$ size of the last dimension of $\mathbf{X}$
3:     $\mathbf{X}_{\text{even}} \leftarrow$ elements of $\mathbf{X}$ at even indices along the last dimension
4:     $\mathbf{X}_{\text{odd}} \leftarrow$ elements of $\mathbf{X}$ at odd indices along the last dimension
5:     $\mathbf{X}' \leftarrow$ Concatenate($\mathbf{X}_{\text{even}}, \mathbf{X}_{\text{odd}}$) along the last dimension.
6:     **return** $\mathbf{X}'$

7: **procedure** CHANNELUNMIX($\mathbf{X}'$)                 $\triangleright$ $\mathbf{X}'$ is a tensor, e.g., $\in \mathbb{R}^{B \times T \times \text{PDim}}$ or $\in \mathbb{R}^{\cdots \times D}$
8:     $D \leftarrow$ size of the last dimension of $\mathbf{X}'$
9:     **Assert** $D$ is an even number.
10:     $D_{\text{half}} \leftarrow D/2$
11:     $\mathbf{X}'_{\text{first\_half}} \leftarrow$ the first $D_{\text{half}}$ elements of $\mathbf{X}'$ along the last dimension
12:     $\mathbf{X}'_{\text{second\_half}} \leftarrow$ the last $D_{\text{half}}$ elements of $\mathbf{X}'$ along the last dimension
13:     Initialize $\mathbf{X}_{\text{out}}$ with the same shape as $\mathbf{X}'$.
14:     Place elements of $\mathbf{X}'_{\text{first\_half}}$ into the even-indexed positions of $\mathbf{X}_{\text{out}}$ along the last dimension.
15:     Place elements of $\mathbf{X}'_{\text{second\_half}}$ into the odd-indexed positions of $\mathbf{X}_{\text{out}}$ along the last dimension.
16:     **return** $\mathbf{X}_{\text{out}}$

---

encoder's Gaussian components $\{\boldsymbol{\mu}_k, \sigma_k^2\}_{k=1}^V$ defined in Section 3.1 (Eq. equation 3.1). Thus, $V$ equals the vocabulary size, $\boldsymbol{\mu}_k^{\text{prior}} = \boldsymbol{\mu}_k$, and $(\sigma_k^{\text{prior}})^2 = \sigma_k^2$. The prior conditional then becomes:

$$p(\mathbf{z}_t|\mathbf{z}_{<t}) = \sum_{k=1}^V \boldsymbol{\pi}_t(\mathbf{z}_{<t})[k]\mathcal{N}(\mathbf{z}_t; \boldsymbol{\mu}_k, \sigma_k^2\boldsymbol{I}) = \sum_{k=1}^V \boldsymbol{\pi}_t(\mathbf{z}_{<t})[k]\mathcal{N}_k(\mathbf{z}_t) \tag{70}$$

where $\boldsymbol{\pi}_t(\mathbf{z}_{<t})$ are the mixture weights predicted by a Transformer based on the history $\mathbf{z}_{<t}$. **Tied Encoder-Decoder:** The decoder $p(x_t|\mathbf{z}_t)$ uses the tied Bayesian formulation from Eq. equation 3.1:

$$p(x_t = k|\mathbf{z}_t) = \frac{\mathcal{N}_k(\mathbf{z}_t)}{\sum_{j=1}^V \mathcal{N}_j(\mathbf{z}_t)} \tag{71}$$

**No Additional Flow Transformations for Prior:** We consider the case where the prior $p(\mathbf{z}_{1:T}) = \prod_t p(\mathbf{z}_t|\mathbf{z}_{<t})$ with $p(\mathbf{z}_t|\mathbf{z}_{<t})$ defined by Eq. equation 70 is used directly in the ELBO (Eq. equation 3). This corresponds to setting $L = 0$ in the context of stacked flow layers (Section 3.3), meaning $\mathbf{z}_{1:T}$ is not further transformed into $\mathbf{u}_{1:T}$ via additional flow layers for this analysis.

Under these conditions, the ELBO (Eq. equation 3) is maximized. Let's analyze the terms in the ELBO related to a single token $x_t$, given its history $\mathbf{x}_{<t}$. The relevant part of the ELBO expectation, for a sample $\mathbf{z}_{1:T} \sim q(\cdot|\mathbf{x}_{1:T})$, can be written per token $t$ as:

$$\mathcal{L}_t^{\text{eff}} = \log p(x_t|\mathbf{z}_t) + \log p(\mathbf{z}_t|\mathbf{z}_{<t}) - \log q(\mathbf{z}_t|x_t) \tag{72}$$

Substituting the definitions:

$$\mathcal{L}_t^{\text{eff}} = \log\left(\frac{\mathcal{N}_{x_t}(\mathbf{z}_t)}{\sum_{j=1}^{V}\mathcal{N}_j(\mathbf{z}_t)}\right) + \log\left(\sum_{k=1}^{V}\boldsymbol{\pi}_t(\mathbf{z}_{<t})[k]\mathcal{N}_k(\mathbf{z}_t)\right) - \log\mathcal{N}_{x_t}(\mathbf{z}_t) \tag{73}$$

$$= \log\mathcal{N}_{x_t}(\mathbf{z}_t) - \log\left(\sum_{j=1}^{V}\mathcal{N}_j(\mathbf{z}_t)\right) + \log\left(\sum_{k=1}^{V}\boldsymbol{\pi}_t(\mathbf{z}_{<t})[k]\mathcal{N}_k(\mathbf{z}_t)\right) - \log\mathcal{N}_{x_t}(\mathbf{z}_t) \tag{74}$$

$$= \log\left(\sum_{k=1}^{V}\boldsymbol{\pi}_t(\mathbf{z}_{<t})[k]\mathcal{N}_k(\mathbf{z}_t)\right) - \log\left(\sum_{j=1}^{V}\mathcal{N}_j(\mathbf{z}_t)\right)$$

$$= \log\left(\sum_{k=1}^{V}\boldsymbol{\pi}_t(\mathbf{z}_{<t})[k]\frac{\mathcal{N}_k(\mathbf{z}_t)}{\sum_{j=1}^{V}\mathcal{N}_j(\mathbf{z}_t)}\right)$$

$$= \log\left(\sum_{k=1}^{V}\boldsymbol{\pi}_t(\mathbf{z}_{<t})[k]p(x_t = k|\mathbf{z}_t)\right) \tag{75}$$

The full ELBO involves an expectation $\mathbb{E}_{\mathbf{z}_{1:T}\sim q(\cdot|\mathbf{x}_{1:T})}[\sum_t \mathcal{L}_t^{\text{eff}}]$.

**Proposition 4.** *(Connection to Cross-Entropy Discrete AR LM). Under the tying conditions specified above (tied prior/encoder components, tied decoder, and $L = 0$ for additional flows), in the limit where the encoder/prior Gaussian components become infinitely narrow (i.e., $\sigma_k^2 \to 0$ for all $k = 1,\ldots,V$), minimizing the negative of the ELBO terms related to prior prediction and reconstruction for token $x_t$ approaches minimizing $-\log\boldsymbol{\pi}_t(\boldsymbol{\mu}_{x_1},\ldots,\boldsymbol{\mu}_{x_{t-1}})[x_t]$. This is equivalent to the negative log-likelihood (cross-entropy) objective for a discrete autoregressive model predicting token $x_t$ given the (deterministic embedding of) previous tokens.*

*Proof.* Consider the term $\mathcal{L}_t^{\text{eff}}$ from Eq. equation 75. The latent variables $\mathbf{z}_t$ are sampled from $q(\mathbf{z}_t|x_t) = \mathcal{N}_{x_t}(\mathbf{z}_t) = \mathcal{N}(\mathbf{z}_t; \boldsymbol{\mu}_{x_t}, \sigma_{x_t}^2\boldsymbol{I})$. As $\sigma_k^2 \to 0$ for all $k$:

1. For a given true token $x_t$, the sample $\mathbf{z}_t \sim \mathcal{N}_{x_t}(\mathbf{z}_t)$ will converge in probability to its mean: $\mathbf{z}_t \to \boldsymbol{\mu}_{x_t}$.

2. Consequently, the history $\mathbf{z}_{<t} = (\mathbf{z}_1,\ldots,\mathbf{z}_{t-1})$ will converge to $(\boldsymbol{\mu}_{x_1},\ldots,\boldsymbol{\mu}_{x_{t-1}})$.

3. The decoder probability $p(x_t = k|\mathbf{z}_t)$ (Eq. equation 71) evaluated at $\mathbf{z}_t \approx \boldsymbol{\mu}_{x_t}$ will behave as follows: If $\boldsymbol{\mu}_k$ are distinct, then for $\mathbf{z}_t \approx \boldsymbol{\mu}_{x_t}$, $\mathcal{N}_{x_t}(\mathbf{z}_t)$ will be large, while $\mathcal{N}_j(\mathbf{z}_t)$ for $j \neq x_t$ will be very small. Thus, $p(x_t = k|\mathbf{z}_t \approx \boldsymbol{\mu}_{x_t}) \to \begin{cases} 1 & \text{if } k = x_t \\ 0 & \text{if } k \neq x_t \end{cases}$.

Substituting these limits into Eq. equation 75, the sum $\sum_{k=1}^{V}\boldsymbol{\pi}_t(\mathbf{z}_{<t})[k]p(x_t = k|\mathbf{z}_t)$ becomes:

$$\boldsymbol{\pi}_t(\boldsymbol{\mu}_{x_1},\ldots,\boldsymbol{\mu}_{x_{t-1}})[x_t] \cdot 1 + \sum_{k\neq x_t}\boldsymbol{\pi}_t(\boldsymbol{\mu}_{x_1},\ldots,\boldsymbol{\mu}_{x_{t-1}})[k] \cdot 0 = \boldsymbol{\pi}_t(\boldsymbol{\mu}_{x_1},\ldots,\boldsymbol{\mu}_{x_{t-1}})[x_t] \tag{76}$$

Therefore, in this limit, the ELBO contribution $\mathcal{L}_t^{\text{eff}}$ approaches $\log\boldsymbol{\pi}_t(\boldsymbol{\mu}_{x_1},\ldots,\boldsymbol{\mu}_{x_{t-1}})[x_t]$. Maximizing the ELBO thus involves maximizing this term, which is equivalent to minimizing its negative: $-\log\boldsymbol{\pi}_t(\boldsymbol{\mu}_{x_1},\ldots,\boldsymbol{\mu}_{x_{t-1}})[x_t]$. This is precisely the cross-entropy loss for predicting the true token $x_t$ given the sequence of means of the previous true tokens as context. $\square$

This connection illustrates that our continuous latent variable framework, under specific simplifying assumptions (most importantly, very low-variance, well-separated encoder components and direct use of the tied mixture prior), can recover the objective of standard discrete autoregressive models. It offers a perspective on how continuous space modeling can generalize or relate to established discrete paradigms. When $\sigma_k^2 > 0$, the objective involves a "soft" version of this cross-entropy, where the target $p(x_t = k|\mathbf{z}_t)$ is not one-hot.

## G Latent Space Evolution Metrics

Our framework's formulation of language modeling within a continuous latent space, processed by stacked autoregressive normalizing flows, allows for a step-by-step observation of text formation. At any intermediate stage $\ell$ of the $L$ flow transformations, the continuous latent sequence $\mathbf{h}_{1:T}^{(\ell)}$ can be decoded using the decoder to yield a corresponding textual output. This ability to materialize text from intermediate representations offers a view into how the model refines an initial latent state towards a final coherent sequence. The continuous nature of the latent variables naturally enables this step-wise refinement process. Each flow layer performs a smooth, differentiable transformation across the entire sequence representation, effectively functioning as a sophisticated editing operation. This type of fine-grained, fully learnable editing process is challenging to implement in discrete token spaces, where making intermediate adjustments typically involves solving complex discrete optimization problems.

To further characterize quantitatively the transformations, we analyze statistics of the latent codes $\mathbf{h}_{1:T}^{(\ell)}$ after each flow layer. These include: (1) the mean L2 norm of token embeddings $\mathbf{h}_t^{(\ell)}$; (2) the mean pairwise cosine similarity among these embeddings within a sequence; (3) the mean Participation Ratio (PR) of the set of token embeddings $\{\mathbf{h}_t^{(\ell)}\}_{t=1}^{T}$; and (4) the mean centroid movement, measuring the Euclidean distance between the average sequence representation $\mathrm{mean}(\mathbf{h}_{1:T}^{(\ell)})$ and that of the preceding layer, $\mathrm{mean}(\mathbf{h}_{1:T}^{(\ell-1)})$.

Observations of these metrics reveal a structured evolution. A consistent decrease in the mean L2 norm across flow layers suggests that the transformations guide the latent representations towards a more defined and compact manifold. Simultaneously, an increasing mean pairwise cosine similarity indicates that token embeddings within a sequence become more semantically clustered, enhancing internal coherence. The utilization of embedding dimensions can be particularly revealing when considering alternating flow directions. For example, an initial Left-to-Right (L2R) flow might establish a foundational representation with a certain value (e.g., 7). A subsequent Right-to-Left (R2L) flow could then see a drop. This reduction might occur as the R2L pass focuses on integrating suffix-based or future context, temporarily constraining the representations to a lower-dimensional manifold that captures these specific right-side dependencies. The final L2R flow, benefiting from the synthesis of both left-anchored and right-anchored information, might then exhibit a significant increase, indicating that it expands the representational capacity to integrate these now bi-directionally informed features into a richer, more expressive state. A diminishing mean centroid movement with each subsequent layer further points to a coarse-to-fine adjustment, naturally enabled by the stacked continuous transformations: initial flow layers induce larger global changes, while later layers perform more subtle, fine-tuning modifications. This hierarchical refinement is a direct benefit of operating in a continuous space where gradual adjustments are possible, unlike the often all-or-nothing choices in discrete generation.

This section provides detailed definitions and computation methods for the metrics employed to analyze the evolution of continuous latent representations $\mathbf{h}_{1:T}^{(\ell)}$ across $L$ stacked autoregressive flow layers. We consider a batch of $B$ sequences, where each sequence consists of $T$ tokens. The $d$-dimensional latent embedding of the $t$-th token in the $b$-th sequence, after processing by the $\ell$-th flow transformation, is denoted $\mathbf{h}_{b,t}^{(\ell)} \in \mathbb{R}^d$. The initial state of these embeddings, prior to any flow transformations (e.g., sampled noise or an encoder's output), is designated $\mathbf{h}_{b,t}^{(0)}$. The subsequent states, corresponding to the outputs of the $L$ flow layers, are $\mathbf{h}_{b,t}^{(1)}, \ldots, \mathbf{h}_{b,t}^{(L)}$. Our statistical analysis primarily focuses on characterizing these $L$ flow layer outputs.

**Mean L2 Norm**    The L2 norm (Euclidean norm) of a token embedding $\mathbf{h}_{b,t}^{(\ell)}$ measures its magnitude in the $d$-dimensional space:

$$\|\mathbf{h}_{b,t}^{(\ell)}\|_2 = \sqrt{\sum_{j=1}^{d}(h_{b,t,j}^{(\ell)})^2} \tag{77}$$

For each sequence $b$ at flow layer $\ell$, the average L2 norm of its token embeddings is calculated as:

$$\bar{N}_b^{(\ell)} = \frac{1}{T} \sum_{t=1}^{T} \|\mathbf{h}_{b,t}^{(\ell)}\|_2 \tag{78}$$

The Mean L2 Norm reported for flow layer $\ell$ is the average of these per-sequence values over all $B$ sequences in the batch:

$$\text{Mean L2 Norm}^{(\ell)} = \frac{1}{B} \sum_{b=1}^{B} \bar{N}_b^{(\ell)} \tag{79}$$

The standard deviation of the set $\{\bar{N}_b^{(\ell)}\}_{b=1}^{B}$ across the batch is also computed. This metric indicates whether the flow transformations tend to expand, contract, or preserve the overall scale of the token embeddings.

**Mean Pairwise Cosine Similarity (Intra-Sequence)**  Cosine similarity quantifies the angular relationship between two embedding vectors. For any two token embeddings $\mathbf{h}_{b,t_1}^{(\ell)}$ and $\mathbf{h}_{b,t_2}^{(\ell)}$ within the same sequence $b$ at flow layer $\ell$, their cosine similarity is:

$$\text{cos\_sim}(\mathbf{h}_{b,t_1}^{(\ell)}, \mathbf{h}_{b,t_2}^{(\ell)}) = \frac{\mathbf{h}_{b,t_1}^{(\ell)} \cdot \mathbf{h}_{b,t_2}^{(\ell)}}{\|\mathbf{h}_{b,t_1}^{(\ell)}\|_2 \|\mathbf{h}_{b,t_2}^{(\ell)}\|_2 + \epsilon_{\text{cos}}} \tag{80}$$

where $\epsilon_{\text{cos}}$ is a small constant (e.g., $10^{-9}$) added to the denominator for numerical stability. For each sequence $b$ at flow layer $\ell$ (assuming $T > 1$), the average cosine similarity is computed over all unique pairs of distinct tokens $(t_1, t_2)$ where $t_1 < t_2$:

$$\bar{C}_b^{(\ell)} = \frac{2}{T(T-1)} \sum_{t_1=1}^{T-1} \sum_{t_2=t_1+1}^{T} \text{cos\_sim}(\mathbf{h}_{b,t_1}^{(\ell)}, \mathbf{h}_{b,t_2}^{(\ell)}) \tag{81}$$

If $T \leq 1$, this average is typically considered undefined or assigned a default value (e.g., 1.0 if $T = 1$, as there are no distinct pairs). The Mean Pairwise Cosine Similarity for flow layer $\ell$ is the average of these per-sequence values over the batch:

$$\text{Mean Pairwise Cosine Sim.}^{(\ell)} = \frac{1}{B} \sum_{b=1}^{B} \bar{C}_b^{(\ell)} \tag{82}$$

The standard deviation of $\{\bar{C}_b^{(\ell)}\}_{b=1}^{B}$ is also reported. This metric provides insight into the internal coherence or degree of representational similarity among tokens within a sequence.

**Mean Participation Ratio (PR) (Intra-Sequence)**  The Participation Ratio (PR) estimates the effective number of dimensions utilized by a collection of embeddings. For a given sequence $b$ at flow layer $\ell$, let $\mathbf{H}_b^{(\ell)} \in \mathbb{R}^{T \times d}$ be the matrix where each row is a token embedding $\mathbf{h}_{b,t}^{(\ell)}$. This analysis requires $T > 1$ and $d > 0$. First, the embeddings are centered by subtracting their mean $\bar{\mathbf{h}}_b^{(\ell)} = \frac{1}{T} \sum_{t=1}^{T} \mathbf{h}_{b,t}^{(\ell)}$ from each $\mathbf{h}_{b,t}^{(\ell)}$ to obtain the centered matrix $\mathbf{H}_{b,\text{centered}}^{(\ell)}$. The $d \times d$ sample covariance matrix of these $T$ centered $d$-dimensional embeddings is:

$$\Sigma_b^{(\ell)} = \frac{1}{T-1} (\mathbf{H}_{b,\text{centered}}^{(\ell)})^\top \mathbf{H}_{b,\text{centered}}^{(\ell)} \tag{83}$$

Let $\{\lambda_1, \ldots, \lambda_d\}$ be the non-negative eigenvalues of $\Sigma_b^{(\ell)}$. The PR for sequence $b$ at layer $\ell$ is defined as:

$$\text{PR}_b^{(\ell)} = \frac{(\sum_{j=1}^{d} \lambda_j)^2}{\sum_{j=1}^{d} \lambda_j^2 + \epsilon_{\text{PR}}} \tag{84}$$

where $\epsilon_{\text{PR}}$ is a small constant to prevent division by zero if all eigenvalues are zero. The PR ranges from 1 (if all centered embeddings are collinear, indicating usage of one effective dimension) to $d$ (if

the variance is distributed isotropically across all $d$ dimensions). The Mean Participation Ratio for flow layer $\ell$ is the average over the batch:

$$\text{Mean PR}^{(\ell)} = \frac{1}{B} \sum_{b=1}^{B} \text{PR}_b^{(\ell)} \tag{85}$$

The standard deviation of $\{\text{PR}_b^{(\ell)}\}_{b=1}^{B}$ is also reported. This metric indicates the breadth of the dimensional subspace actively occupied by the token representations within a sequence.

**Mean Centroid Movement (Inter-Step)**  The centroid of a sequence $b$ at a specific original step $s \in \{0, \ldots, L\}$ (where $s = 0$ denotes the initial state, and $s = 1, \ldots, L$ denote the outputs of the successive flow layers) is the mean of its token embeddings at that step:

$$\boldsymbol{\mu}_b^{(s)} = \frac{1}{T} \sum_{t=1}^{T} \mathbf{h}_{b,t}^{(s)} \tag{86}$$

The centroid movement for sequence $b$ induced by the $\ell$-th flow layer (which transforms the representation from original step $\ell - 1$ to original step $\ell$) is the Euclidean distance between the centroids of these two consecutive states:

$$M_b^{(\ell)} = \|\boldsymbol{\mu}_b^{(\ell)} - \boldsymbol{\mu}_b^{(\ell-1)}\|_2, \quad \text{for } \ell = 1, \ldots, L \tag{87}$$

The Mean Centroid Movement for flow layer $\ell$ is the average of these per-sequence movement magnitudes over the batch:

$$\text{Mean Centroid Movement}^{(\ell)} = \frac{1}{B} \sum_{b=1}^{B} M_b^{(\ell)} \tag{88}$$

The standard deviation of $\{M_b^{(\ell)}\}_{b=1}^{B}$ is also reported. This metric measures the magnitude of change that each flow layer imparts on the average representation of a sequence.

## H  FLOPs Calculation Details

This appendix provides a detailed breakdown of how the Floating Point Operations (FLOPs) for the forward pass of an entire sequence are calculated for both the regular Transformer model and the Flow Transformer model featuring the MAF-MLP head, as presented in Fig. 7. We assume a multiply-accumulate operation constitutes 2 FLOPs. FLOPs from bias terms, normalization layers, and activation functions (like GELU or ELU in the main Transformer body or within the MAF-MLP module) are generally not counted, as their contribution is typically much smaller than that of matrix multiplications.

**Regular Transformer Model**  The calculation for a standard autoregressive Transformer model is as follows. Let $L$ be the number of Transformer layers, $d$ be the model dimension (embedding dimension, d_model), $S$ be the input sequence length, and $V$ be the vocabulary size. We assume an MLP (Feed-Forward Network) ratio of 4, meaning the inner dimension of the FFN is $4d$. We also assume that the sum of attention head dimensions $d_{\text{attn}}$ equals $d$.

The FLOPs for each component for processing an entire sequence of length $S$ are:

First, for *Input Embeddings (Token + Positional)*, each token's embedding lookup and addition of positional encoding is approximated. For $S$ tokens, this is $C_{\text{embed}} = 4\, d\, S$.

Second, for *Attention QKV Projections*, for each of $L$ layers, input of shape $(S, d)$ is projected to queries, keys, and values. This can be viewed as a multiplication by a weight matrix of effective shape $(d, 3d)$. FLOPs per layer: $2 \times S \times d \times 3d = 6\, S\, d^2$. Total for $L$ layers: $C_{\text{QKV}} = 6\, L\, d^2\, S$.

Third, for *Attention Logits (Query-Key Dot Products)*, in each of $L$ layers, each of $S$ query vectors (dimension $d$) computes dot products with $S$ key vectors (dimension $d$). FLOPs for one query vector to attend to all $S$ key vectors: $S \times (2d) = 2\, d\, S$. For all $S$ query vectors in a layer: $S \times (2\, d\, S) = 2\, d\, S^2$. Total for $L$ layers: $C_{\text{QK}} = 2\, L\, d\, S^2$.

Fourth, for *Attention Output Projection*, for each of $L$ layers, the attention output (shape $(S, d)$) is projected by a weight matrix of shape $(d, d)$. FLOPs per layer: $2 \times S \times d \times d = 2\,S\,d^2$. Total for $L$ layers: $C_{\text{proj}} = 2\,L\,d^2\,S$.

Fifth, for *Feed-Forward Network (FFN/MLP)*, each FFN block has two linear layers. The first linear layer ($d \rightarrow 4d$): $2 \times S \times d \times 4d = 8\,S\,d^2$. The second linear layer ($4d \rightarrow d$): $2 \times S \times 4d \times d = 8\,S\,d^2$. Total FFN FLOPs per layer: $16\,S\,d^2$. Total for $L$ layers: $C_{\text{FF}} = 16\,L\,d^2\,S$.

Finally, for *Output Linear Head (De-embedding)*, it maps final hidden states (shape $(S, d)$) to vocabulary logits (shape $(S, V)$) using a weight matrix of shape $(d, V)$. FLOPs: $C_{\text{head}} = 2 \times S \times d \times V = 2\,d\,V\,S$.

Summing these components, the total forward FLOPs for the entire sequence are:

$$
\begin{aligned}
C_{\text{fwd/sequence}}^{\text{regular}} &= C_{\text{embed}} + C_{\text{QKV}} + C_{\text{QK}} + C_{\text{proj}} + C_{\text{FF}} + C_{\text{head}} \\
&= 4\,d\,S + 6\,L\,d^2\,S + 2\,L\,d\,S^2 + 2\,L\,d^2\,S + 16\,L\,d^2\,S + 2\,d\,V\,S \\
&= (6 + 2 + 16)\,L\,d^2\,S + 2\,L\,d\,S^2 + 4\,d\,S + 2\,d\,V\,S \\
&= 24\,L\,d^2\,S + 2\,L\,d\,S^2 + 4\,d\,S + 2\,d\,V\,S
\end{aligned}
$$

This can be factored by $S$:

$$
C_{\text{fwd/sequence}}^{\text{regular}} = S \left[\, 24\,L\,d^2 + 2\,L\,d\,S + 4\,d + 2\,d\,V \,\right]
$$

For the calculations related to Fig. 7, the parameters $L$ (number of layers) and $d$ (model dimension) correspond to the GPT-2 configurations specified in the main text. The sequence length $S$ is set to 1024, and the vocabulary size $V$ is 50257.

**Flow Transformer Model (with MAF-MLP Head)**  This model uses the same Transformer body as the regular model but replaces the standard linear output head with the custom MAF-MLP module. The input sequence structure is also modified based on a patch size $P$.

Let $L$ be the number of Transformer layers in the main body, $d_{\text{tf}}$ be the model dimension of the main Transformer body (i.e., n_embd from GPT-2 configs), $S_{\text{base}}$ be the original sequence length (e.g., 1024), $P$ be the patch size, and $S_{\text{special}} = \lfloor S_{\text{base}}/P \rfloor$ be the effective sequence length for the Transformer body, calculated using integer division. If this value becomes 0 (e.g., if $P > S_{\text{base}}$), it is clamped to 1 for calculation purposes.

The FLOPs for the Transformer body (excluding any output head) are:

$$
C_{\text{body/no-head}} = 24\,L\,d_{\text{tf}}^2\,S_{\text{special}} + 2\,L\,d_{\text{tf}}\,S_{\text{special}}^2 + 4\,d_{\text{tf}}\,S_{\text{special}}
$$

To this, we add the FLOPs from the MAF-MLP module, calculated per token and then multiplied by $S_{\text{special}}$.

For the MAF-MLP module, let $c_{\text{in}}$ be the module's input channels, determined by $16P$, num_mixtures $= 64$, $c_{\text{out\_per\_in}}$ be the module's output channels per input channel, defined as $3 \times$ num_mixtures $= 3 \times 64 = 192$, $D_{\text{module}}$ (the module's hidden_size parameter, fixed at 128 for the plot) be the module's internal hidden dimension, and $E$ be the module's computed embed_size, calculated as $E = \min(\max(1, \text{int}(D_{\text{module}} \times 9.0/16.0/(c_{\text{in}} - 1))), 96)$, where $\text{int}(\cdot)$ denotes integer casting. (We ensure $c_{\text{in}} - 1 > 0$ as $P \geq 1 \implies c_{\text{in}} \geq 16$).

The FLOPs for the MAF-MLP module ($C_{\text{module}}$) per token, considering only its linear layers, are:

First, the in_to_features layer projects an input of effective dimension $3(c_{\text{in}} - 1)$ to $E(c_{\text{in}} - 1)$. FLOPs: $C_1 = 2 \times 3(c_{\text{in}} - 1) \times E(c_{\text{in}} - 1) = 6\,E\,(c_{\text{in}} - 1)^2$.

Second, the features_to_hidden layer projects an input of dimension $d_{\text{tf}} + E(c_{\text{in}} - 1)$ (concatenation of transformer output features of dimension $d_{\text{tf}}$ and module's internal in_features) to an output dimension of $(D_{\text{module}}/2)c_{\text{in}}$. FLOPs: $C_2 = 2 \times [d_{\text{tf}} + E(c_{\text{in}} - 1)] \times (D_{\text{module}}/2 \times c_{\text{in}}) = D_{\text{module}}\,c_{\text{in}}\,[d_{\text{tf}} + E(c_{\text{in}} - 1)]$.

Third, the hidden_to_out layer projects an input of dimension $(D_{\text{module}}/2)c_{\text{in}}$ to $c_{\text{out\_per\_in}}c_{\text{in}}$. FLOPs: $C_3 = 2 \times (D_{\text{module}}/2 \times c_{\text{in}}) \times (c_{\text{out\_per\_in}}c_{\text{in}}) = D_{\text{module}}\,c_{\text{in}}^2\,c_{\text{out\_per\_in}}$.

The total FLOPs per token for the module is $C_{\text{module}} = C_1 + C_2 + C_3$:

$$C_{\text{module}} = 6\,E\,(c_{\text{in}} - 1)^2 + D_{\text{module}}\,c_{\text{in}}\,[d_{\text{tf}} + E(c_{\text{in}} - 1)] + D_{\text{module}}\,c_{\text{in}}^2\,c_{\text{out\_per\_in}}$$

The total forward FLOPs for the entire sequence in the Flow Transformer model are:

$$C_{\text{fwd/seq}}^{\text{special}} = C_{\text{body/no-head}} + S_{\text{special}} \times C_{\text{module}}$$
$$= S_{\text{special}} \left[ (24\,L\,d_{\text{tf}}^2 + 2\,L\,d_{\text{tf}}\,S_{\text{special}} + 4\,d_{\text{tf}}) + C_{\text{module}} \right]$$

The specific parameter values used to generate the data for Fig. 7 are summarized below. For the regular Transformer, the base model parameters ($L$, $d$, $S = 1024$, $V = 50257$) are taken from standard GPT-2 configurations (e.g., `gpt2`, `gpt2-medium`, `gpt2-large`, `gpt2-xl`). For the Flow Transformer (Special Transformer), the main body parameters $L$ and $d_{\text{tf}}$ also correspond to these GPT-2 configurations. The parameters specific to its MAF-MLP head and sequence processing depend on the patch size $P \in \{1, 2, 4\}$ as follows: $S_{\text{special}} = \lfloor 1024/P \rfloor$ (clamped to 1 if 0); $c_{\text{in}} = 16P$; $D_{\text{module}}$ (the module's `hidden_size`) is 128; `num_mixtures` = 64 resulting in $c_{\text{out\_per\_in}} = 192$; and $E = \min(\max(1, \text{int}(\lfloor D_{\text{module}} \times 9/(16\,(c_{\text{in}} - 1))\rfloor)), 96)$, where $\text{int}(\cdot)$ denotes integer casting. These values directly correspond to those listed in Table 1 and Table 2 of the user-provided values document that informed these calculations.

# I   Related Works

**Coupling-based Normalizing Flows**   A substantial body of work has focused on designing expressive invertible transformations with tractable Jacobian determinants for normalizing flows. NICE [16] pioneered the use of additive coupling layers, which made the Jacobian determinant computation straightforward. RealNVP [17] built upon this by introducing scaling and shifting operations, thereby increasing model flexibility. Glow [37] further improved these models by incorporating invertible $1 \times 1$ convolutions, which led to better performance in image generation. Flow++ [29] enhanced expressiveness by integrating attention mechanisms. iResNet [2] showed that standard ResNet architectures [28] could be made invertible by adding normalization steps. Additionally, normalizing flows have been instrumental in improving Variational Autoencoders (VAEs) [38] by enabling more flexible posteriors [63, 78], and in diffusion models [62] by introducing flexible nonlinear drift and diffusion terms [36]. However, these approaches often require carefully designed and restrictive architectures, which can hinder scalability.

**Continuous Normalizing Flows**   Neural Ordinary Differential Equations [11] reinterpret the ResNet architecture as a deterministic ODE in the continuous-time limit, which naturally extends to the concept of Continuous Normalizing Flows. In this setting, invertibility is guaranteed, and the Jacobian determinant reduces to the trace of the Jacobian. The adjoint method, based on Pontryagin's Maximum Principle [54], enables efficient gradient computation with constant memory requirements. FFJORD [24] made Jacobian trace estimation more efficient by using Hutchinson's estimator [33]. Despite these advances, such models can be numerically unstable during training and sampling, as discussed in [43, 81]. Their expressiveness can be further increased by introducing auxiliary variables [9, 18].

**Autoregressive Normalizing Flows**   There has also been considerable progress in combining normalizing flows with autoregressive models. IAF [39] proposed dimension-wise affine transformations conditioned on previous dimensions for variational inference, while MAF [50] utilized the MADE [20] architecture to realize invertible mappings via autoregressive transformations. Neural autoregressive flow [32] replaced the affine transformation in MAF with a monotonic neural network per dimension, increasing expressiveness at the expense of analytical invertibility. T-NAF [52] extended NAF by employing a single autoregressive Transformer. Block Neural Autoregressive Flow [8] instead fits an end-to-end autoregressive monotonic neural network, as opposed to NAF's dimension-wise approach, but also loses analytical invertibility.

**Probability Flow in Diffusion**   Diffusion models [30, 60, 62] generate samples by simulating stochastic differential equations. Song et al. [62] introduced a deterministic ODE formulation, known as scoreflow [61], as a counterpart to the stochastic process. This scoreflow can be viewed as a

special case of continuous normalizing flows, where the learned score and base drift are combined into a new drift term. However, Lu et al. [46] showed that the standard training objective in diffusion models, which is based on a first-order score approximation, does not maximize the likelihood for the scoreflow.

**Diffusion models, other generative models**   Diffusion models [30, 62] have recently emerged as powerful generative models, achieving impressive results. Stable Diffusion [53] and OpenSora [80] have demonstrated the ability of diffusion models to generate extremely high-dimensional data. Other prominent generative models include Variational Autoencoders (VAEs) [38] and Generative Adversarial Networks (GANs) [23]. VQ-VAE [65] addresses posterior collapse and achieves strong generative performance, serving as a key component in later latent diffusion models [53]. In the GAN domain, works such as [4, 34, 35] have demonstrated the ability to generate high-resolution images with relatively low inference cost, though training GANs remains challenging due to stability issues [69].

**Continuous Diffusion for Discrete Data**   A common strategy for modeling discrete data is to operate in a continuous embedding space, adapting continuous diffusion techniques. This enables the use of powerful continuous models, but requires mapping back to the discrete domain. **Diffusion-LM** [40] applied continuous diffusion to word embeddings, allowing for controllable text generation via gradient-based guidance. **Plaid** [27] focused on likelihood-based text modeling, jointly optimizing embeddings and model parameters using the VLB, categorical reparameterization, an output prior, a learned conditional likelihood $p(x|z_0)$, and self-conditioning. **CDCD** [15] used a probability flow ODE on embeddings, employing score interpolation to jointly train embeddings and a denoising Transformer with a cross-entropy loss and time warping. **Bit Diffusion** [12] represented discrete data as continuous "analog bits," incorporating self-conditioning and asymmetric time intervals. While these methods are effective, they rely on continuous relaxations or embeddings, motivating the development of models that operate directly on discrete spaces. Many of these works also explore non-autoregressive, parallel generation approaches [3, 10, 13, 14, 26, 31, 40, 57, 72, 74, 75, 76], in contrast to sequential autoregressive models.

**Discrete Diffusion Models**   Another line of research focuses on diffusion processes that are inherently defined on discrete state spaces, typically using Markov chains. Building on early work [31, 60], **D3PM** [1] generalized discrete diffusion by employing various structured transition matrices (such as uniform, absorbing, and Gaussian-like) and training with a hybrid VLB/cross-entropy loss. Campbell et al. [6] extended this to Continuous-Time Markov Chains (CTMCs), deriving a continuous-time ELBO and proposing efficient sampling methods like tau-leaping and predictor-corrector schemes, leveraging factorization for scalability.

Rather than simulating Markov chains directly, some works define score-like quantities for discrete diffusion. The concrete score, given by the ratio of marginal probabilities $p_t(\mathbf{y})/p_t(\mathbf{x})$, serves as a discrete analogue to the continuous score [45, 48]. **SEDD** [45] trained models using a score entropy objective ($L_{DSE}$) based on this ratio, relating it to the ELBO and employing Tweedie $\tau$-leaping for sampling. Sun et al. [64] introduced categorical ratio matching in a CTMC framework, learning singleton conditionals $p_t(x^d|\mathbf{x}^{\backslash d})$ with a tractable loss and an analytical reverse sampler. Building on this, Ou et al. [49] showed that for absorbing diffusion, the concrete score decomposes into a time-independent conditional and a time-dependent scalar, simplifying the model (**RADD**) and leading to the Denoising Cross-Entropy (DCE) loss.

Masked (absorbing) diffusion, which replaces tokens with a special [MASK] token during the forward process, has proven highly effective. **MDLM** [56] introduced a substitution-based parameterization (SUBS) and derived a simplified Rao-Blackwellized ELBO equivalent to weighted Masked Language Modeling (MLM) losses, enabling generative training of encoder-only models. Shi et al. [58] (**MD4**) further unified this framework, deriving a simple ELBO with SNR invariance properties akin to continuous diffusion and generalizing to state-dependent masking schedules.

Further advances have refined the parameterization and mechanisms of discrete diffusion. **Reparameterized Discrete diffusion Models (RDM)** [79] identified a route-and-denoise mechanism, reducing the objective to cross-entropy on noisy tokens and enabling adaptive routing during sampling. Liu et al. [44] proposed **Discrete Diffusion with Planned Denoising (DDPD)**, which factorizes the

reverse process into a planner (predicting corruption) and a denoiser, allowing for adaptive sampling via the Gillespie algorithm guided by the planner.

Discrete Flow Matching provides another generalization. Gat et al. [19] defined probability paths interpolating between discrete distributions and derived corresponding probability velocities, analogous to continuous flow matching, yielding a unified sampling theory. [7] formulated discrete flows using CTMCs, learning scores via cross-entropy and enabling flexible inference by adjusting the rate matrix family at test time without retraining, also supporting multimodal generation. Discrete diffusion concepts have also been extended to structured data, such as graphs, as in **DiGress** [67], which uses specialized noise transitions, auxiliary features, and classifier guidance.

