# OpenReview forum: "Flexible Language Modeling in Continuous Space with Transformer-based Autoregressive Flows"
_NeurIPS.cc/2025/Conference — NeurIPS 2025 poster_

### Official Review · Reviewer_Yc2p · 2025-07-02

**Clarity:** 3
**Significance:** 3
**Originality:** 4
**Rating:** 5
**Confidence:** 4

**Summary:**

the paper proposed a novel framework for language modelling using autoregressive normalizing flows instead of classical transformer architecture.
As such the language modelling occurs in a continues stacked multi-variate probability space , rather than relying on classical log-prob sampling.
The authors conduct comprehensive language modelling experiments / comparison.

**Questions:**

please address weaknesses.
willing to increase score if code is made public.

**Ethical Concerns:**

["NO or VERY MINOR ethics concerns only"]

**Final Justification:**

the authors justified they'd publish the code.
given the novelty, and potential impact to the research community, i find this very much publish-worthy, albeit the paper in general is missing 'some standards', that however may not yet be as relevant in the current state of this research direction.
as such i'd grade it not 'a flawless strong accept' but indeed a solid accept.

**Limitations:**

yes

**Quality:**

4

**Strengths And Weaknesses:**

- S1 originality - it is the first paper i've seen that successfully applies normalizing flows as a language modelling framework to that extent.
- S2 the paper is well explained and can be followed. very thorough work, well explained limitations and discussions etc

My major issue:
- W1 issue i have is that code is not made available - as such results are not reproducible at all. Will this be done at some point?

Minors:
- W2 i find it a bit odd seeing that undiscussed Figures and Tables in main corpus
- (W3 wall-times might be poor but strategies like patching are already introduced to cope that - still not sure how far they got)
- (W4 no downstream / prior-ed generative examples)

---

> ### Author Rebuttal · Authors · 2025-07-31
>
> We are very grateful for the reviewer's positive evaluation and constructive feedback. We are delighted that the reviewer recognized the originality of our work (S1) and appreciated the paper's clarity and thoroughness (S2). We address the minor weaknesses below.
>
> # W2
>
> We appreciate the reviewer's attention for detail. The content of these figures is discussed in the main text, but we will improve the explicit cross-referencing in the new version. Specifically, Figure 8 is discussed in Lines 281-287, and Figure 9 is discussed in Lines 263-277.
> We will add explicit references to connect the text more directly to improve readability.
>
> # W3
>
> Please see our response to pnYG Q3 for a detailed discussion.
>
> # W4
>
> Please see our response to pnYG Q1 for a detailed discussion.
>
> # W1
>
> We will make the training code available after going through internal review process. As a start we include the core MoG coupling transform here below:
>
> ```python
> class MixtureOfGaussianTransform(nn.Module):
>   def __init__(self, *, c_in = -1, num_mixtures = 10):
>     super().__init__()
>     self.num_mixtures = num_mixtures
>     self.c_in = c_in
>     if c_in > 0:
>       self.scaling_factor = nn.Parameter(torch.zeros(self.c_in))
>       self.mixture_scaling_factor = nn.Parameter(
>         torch.zeros(self.c_in, self.num_mixtures)
>       )
>
>   @staticmethod
>   def _compute_mixture_cdf(
>     x,
>     log_pi,
>     mu,
>     sigma,
>   ):
>     x_orig_dtype = x.dtype
>     if x.dtype != torch.float64:
>       x = x.to(torch.float64)
>     if log_pi.dtype != torch.float64:
>       log_pi = log_pi.to(torch.float64)
>     if mu.dtype != torch.float64:
>       mu = mu.to(torch.float64)
>     if sigma.dtype != torch.float64:
>       sigma = sigma.to(torch.float64)
>
>     pi = torch.softmax(log_pi, dim=-1)
>     sigma = sigma.clamp(min=1e-7)  # Safety clamp
>     z = (x.unsqueeze(-1) - mu) / sigma
>     cdf_standard = 0.5 * (1.0 + torch.erf(z / math.sqrt(2.0)))
>     cdf_mixture = (pi * cdf_standard).sum(dim=-1)
>     # Clamp and return in original dtype (or double if already double)
>     return cdf_mixture.clamp(0.0, 1.0).to(x_orig_dtype)
>
>   @staticmethod
>   def _compute_mixture_pdf(
>     x,
>     log_pi,
>     mu,
>     sigma,
>   ):
>     x_orig_dtype = x.dtype
>     if x.dtype != torch.float64:
>       x = x.to(torch.float64)
>     if log_pi.dtype != torch.float64:
>       log_pi = log_pi.to(torch.float64)
>     if mu.dtype != torch.float64:
>       mu = mu.to(torch.float64)
>     if sigma.dtype != torch.float64:
>       sigma = sigma.to(torch.float64)
>
>     pi = torch.softmax(log_pi, dim=-1)
>     sigma = sigma.clamp(min=1e-7)  # Safety clamp
>     z = (x.unsqueeze(-1) - mu) / sigma
>     log_pdf_standard = -0.5 * (z**2 + math.log(2 * math.pi))
>     pdf_standard = torch.exp(log_pdf_standard)
>     pdf_component = pdf_standard / sigma
>     pdf_mixture = (pi * pdf_component).sum(dim=-1)
>     # Clamp and return in original dtype (or double if already double)
>     return pdf_mixture.clamp(min=1e-9).to(x_orig_dtype)
>
>   @staticmethod
>   def _invert_mixture_cdf(
>     p,
>     log_pi,
>     mu,
>     sigma,
>     max_iter = 25,
>     tol = 1e-6,
>     eps = 1e-6,
>   ):
>     p_orig_dtype = p.dtype
>     if p.dtype != torch.float64:
>       p = p.to(torch.float64)
>     if log_pi.dtype != torch.float64:
>       log_pi = log_pi.to(torch.float64)
>     if mu.dtype != torch.float64:
>       mu = mu.to(torch.float64)
>     if sigma.dtype != torch.float64:
>       sigma = sigma.to(torch.float64)
>
>     pi = torch.softmax(log_pi, dim=-1)
>     sigma = sigma.clamp(min=1e-7)  # Original safety clamp
>
>     mu_mix = (pi * mu).sum(-1)
>     var_mix = (pi * (sigma**2 + mu**2)).sum(-1) - mu_mix**2
>     sigma_mix = var_mix.clamp(min=1e-5).sqrt()  # Original clamp value
>
>     p_clamped = p.clamp(eps, 1.0 - eps)  # Use provided eps
>     z_approx = math.sqrt(2.0) * torch.erfinv(2 * p_clamped - 1)
>     x = mu_mix + sigma_mix * z_approx
>
>     bracket_width_factor = 5.0
>     initial_bracket_width = bracket_width_factor * sigma_mix
>     initial_bracket_width = initial_bracket_width.clamp(min=1e-3)
>     a = mu_mix - initial_bracket_width
>     b = mu_mix + initial_bracket_width
>
>     newton_damping_width = 5.0 * sigma_mix
>     newton_damping_width = newton_damping_width.clamp(min=1e-3)
>
>     for _ in range(max_iter):
>       x = x.clamp(min=a, max=b)
>
>       F_x = MixtureOfGaussianTransform._compute_mixture_cdf(
>         x,
>         log_pi,
>         mu,
>         sigma,
>       )
>       m_x = MixtureOfGaussianTransform._compute_mixture_pdf(
>         x,
>         log_pi,
>         mu,
>         sigma,
>       )
>       F_x = F_x.to(torch.float64)
>       m_x = m_x.to(torch.float64)
>       err = F_x - p  # p is already double
>
>       if err.abs().max() < tol:
>         break
>
>       m_x_safe = m_x.clamp(min=1e-9)
>       delta = err / m_x_safe
>       delta = delta.clamp(min=-newton_damping_width, max=newton_damping_width)
>       x_newton = x - delta
>
>       F_newton = MixtureOfGaussianTransform._compute_mixture_cdf(
>         x_newton,
>         log_pi,
>         mu,
>         sigma,  # Pass double tensors
>       )
>       F_newton = F_newton.to(torch.float64)
>       err_newton = F_newton - p  # p is already double
>       improved = err_newton.abs() <= err.abs()
>
>       x_bisection = 0.5 * (a + b)
>       x_next = torch.where(improved, x_newton, x_bisection)
>
>       F_current = MixtureOfGaussianTransform._compute_mixture_cdf(
>         x_next,
>         log_pi,
>         mu,
>         sigma,  # Pass double tensors
>       )
>       F_current = F_current.to(torch.float64)
>       a = torch.where(F_current < p, x_next, a)
>       b = torch.where(F_current >= p, x_next, b)
>       x = x_next
>
>     return x.to(p_orig_dtype)
>
>   @staticmethod
>   def fn_extract_params(
>     nn_out,
>     num_mixtures,
>   ):
>     param_num = num_mixtures * 3
>     nn_out = nn_out.reshape(
>       nn_out.shape[:-1] + (nn_out.shape[-1] // param_num, param_num)
>     )
>     log_pi = nn_out[..., 0:num_mixtures]
>     mu = nn_out[..., num_mixtures : 2 * num_mixtures]
>     log_var = nn_out[..., 2 * num_mixtures : 3 * num_mixtures]
>     # sigma = torch.nn.functional.softplus(log_sigma)
>     return NestedMap(
>       log_pi=log_pi,
>       mu=mu,
>       log_var=log_var,
>     )
>
>   @capture_metrics
>   @staticmethod
>   def fn_x_to_z_transform(
>     x,
>     *,
>     fn_model_fprop = None,
>     num_mixtures,
>     scaling_factor,
>     mixture_scaling_factor,
>     log_pi = None,
>     mu = None,
>     sigma = None,
>     should_flip = False,
>     eps = 1e-6,
>   ):
>     x_orig_dtype = x.dtype
>     x = x.to(torch.float64)
>
>     flip_dim = 1
>     if x.dim() > 2:
>       flip_dim = 1
>
>     if should_flip:
>       if x.dim() <= flip_dim:
>         raise ValueError(
>           ""
>         )
>       x = torch.flip(x, dims=[flip_dim])
>
>     if log_pi is None or mu is None or sigma is None:
>       if fn_model_fprop is not None:
>         nn_out = fn_model_fprop(
>           x.to(x_orig_dtype)
>         )  # Pass original dtype if model expects it
>         params = MixtureOfGaussianTransform.fn_extract_params(
>           nn_out, num_mixtures
>         )
>         log_pi = params.log_pi.to(torch.float64)
>         mu = params.mu.to(torch.float64)
>         log_var = params.log_var.double()
>         sigma = (0.5 * log_var).exp()
>       else:
>         raise ValueError("")
>     else:
>       log_pi = log_pi.to(torch.float64)
>       mu = mu.to(torch.float64)
>       sigma = sigma.to(torch.float64)
>
>     p = MixtureOfGaussianTransform._compute_mixture_cdf(x, log_pi, mu, sigma)
>     p = p.to(torch.float64)
>     p_clamped = p.clamp(eps, 1.0 - eps)
>     z = math.sqrt(2.0) * torch.erfinv(2 * p_clamped - 1)
>
>
>     pdf_mix = MixtureOfGaussianTransform._compute_mixture_pdf(
>       x, log_pi, mu, sigma
>     )
>     pdf_mix = pdf_mix.to(torch.float64)
>     log_pdf_std_normal = -0.5 * (z**2 + math.log(2 * math.pi))
>     ldj = (
>       torch.log(pdf_mix.clamp(min=1e-9)) - log_pdf_std_normal
>     )
>
>     if should_flip:
>       if x.dim() > flip_dim:
>         z = torch.flip(z, dims=[flip_dim])
>         ldj = torch.flip(ldj, dims=[flip_dim])
>
>     return z.to(x_orig_dtype), ldj.to(x_orig_dtype)
> ```
> ---
>
> We hope these clarifications fully address the reviewer's concerns. We thank the reviewer again for their time and valuable feedback.

---

> > ### Comment · Reviewer_Yc2p · 2025-08-03
> >
> > thanks for addressing my comments and showing good faith.
> > will the code be made available until the conference starts?
> > othw would keep the score.

---

> > > ### Author Response · Authors · 2025-08-03
> > >
> > > Thanks so much for your follow-up and comments. Yes, the code will be made available before December, ahead of the conference. In the meantime, we're happy to clarify any questions regarding reproducibility. We truly appreciate your time and support!

---

### Official Review · Reviewer_GUST · 2025-07-03

**Clarity:** 2
**Significance:** 3
**Originality:** 4
**Rating:** 5
**Confidence:** 4

**Summary:**

This work develops a language model that performs language generation in a continuous latent space. They project discrete tokens into learnable continuous representations that are then modeled with autoregressive normalizing flows (using the transformer architecture). By stacking multiple unidirectional flow layers, the model can incorporate bidirectional context into generation. The authors introduce mixture-based coupling transformations which significantly outperform affine coupling layers. They evaluate their approach on two language modeling benchmarks (TEXT8 and OpenWebText), reporting validation likelihoods.

**Questions:**

What is the generative performance of the model? In terms of generative perplexity, MAUVE score, etc?

How does changing the patch size affect the likelihood and generative performance of the model?

**Ethical Concerns:**

["NO or VERY MINOR ethics concerns only"]

**Final Justification:**

My primary concerns revolved around the lack of rigorous evaluation of language generation behavior and the lack of an ablation studying the influence of the patch size on the performance of these frameworks.
* The authors presented language generation evaluations during the rebuttal period that are quite strong. Their framework compares favorably against discrete diffusion baselines and only lags behind the autoregressive baseline. As stated in the original review, I don't view lagging behind such an established approach as a large negative for novel approaches.
* The patching ablation study nicely provided insight into the interaction between the patch size an number of flow layers. It would be nice to see the impact on the patch size/flow layers on language generation metrics as well, but the presented ablation is reasonable.

**Limitations:**

yes

**Quality:**

3

**Strengths And Weaknesses:**

**Strengths**

This is an interesting approach to language generation that I am not aware of being explored recently. This kind of approach represents an interesting alternative to autoregressive and discrete diffusion models. The overall approach is well-formulated and theoretically sound.

The proposed mixture-based coupling transformations significantly improve upon the more common affine coupling transformation, although they are not the first to introduce such a coupling transformation. The discussion of the equivalence of their Mixture-of-Gaussian coupling transformation and the previously proposed Mixture-of-Logistics coupling transformation in the appendix is an interesting theoretical contribution in its own right.

The performance of their approach is strong compared to recent popular discrete diffusion models (e.g. SEDD, MDLM).

The flexibility of their bidirectional approach with respect to patching is a nice advantage of this framework, enabling the model to operate at multiple resolutions.

**Weaknesses**

It is usually the case with these kinds of work that there is a persistent gap with the simpler autoregressive approach and that is also the case with this work. I don’t consider this a strong weakness as novel approaches are of course welcome.

The biggest limitation is the lack of metrics being reported for actual language generation. Reporting likelihoods is great, but quantifying actual generation performance is absolutely critical. Actual generation metrics such as generative perplexity, MAUVE score, diversity metrics, etc. would significantly strengthen the work. I recognize that one plot is included compared against sampling two tokens at a time from an autoregressive model. I understand the point being made by that plot, but comparing against the actual generative behavior of the baselines is critical.

There is a fair amount of discussion of the patching, but limited results are presented related to that design choice. It would be interesting to actually see that ablated to see if reducing the patch size is feasible in practice or whether that significantly degrades quality.

There were some parts of the writing that can be improved. For example, missing capitalization on line 83, line 965 is not grammatical.

---

> ### Author Rebuttal · Authors · 2025-07-31
>
> We are very grateful to the reviewer for their time and for providing such thoughtful and constructive feedback. We are encouraged that the reviewer found our approach to be a "novel, interesting, and theoretically sound" alternative to existing paradigms. The suggestions have been invaluable in helping us strengthen the paper.
>
> Below, we address each of the reviewer's points in detail.
>
> ### **1. On the Performance Gap with Discrete Autoregressive Models**
>
> > *“It is usually the case with these kinds of work that there is a persistent gap with the simpler autoregressive approach and that is also the case with this work. I don’t consider this a strong weakness as novel approaches are of course welcome.”*
>
> We thank the reviewer for this insightful comment and for acknowledging the value of exploring novel approaches. We fully agree that a performance gap in likelihood still exists when compared to state-of-the-art discrete autoregressive (AR) models.
>
> Our work's primary goal is to establish the viability and unique advantages of a new design space for language modeling within continuous latent spaces. As the reviewer noted in the strengths, our model demonstrates superior likelihood performance compared to other continuous-space models (e.g., diffusion, other flows) and also surpasses recent discrete diffusion baselines (Table 2). We believe this positions our framework as a strong and promising new direction, and we are optimistic that future work can further close the gap with traditional AR models while retaining the unique flexibility our method provides.
>
> ### **2. On Generative Quality Evaluation (Generative PPL, MAUVE, and Diversity)**
>
> > *“The biggest limitation is the lack of metrics being reported for actual language generation. Reporting likelihoods is great, but quantifying actual generation performance is absolutely critical. Actual generation metrics such as generative perplexity, MAUVE score, diversity metrics, etc. would significantly strengthen the work.”*
>
> This is a critical point, and we thank the reviewer for this excellent suggestion. To address this, we have conducted a new set of experiments to rigorously evaluate the generative quality of our models on OpenWebText.
>
> The table below presents these new results, including **generative perplexity (Gen PPL)**, **MAUVE**, and **sequence entropy**, comparing our two main model variants against strong discrete diffusion and autoregressive baselines. The results are highly encouraging and demonstrate the strong generative capabilities of our approach.
>
> | Method               | Timesteps              | Gen PPL ↓           | MAUVE↑           | Entropy↑             |
> |---------------------|------------------------|----------------|-----------------|-----------------------|
> | **Data**            | -                      | 14.7            | 1.0               | 5.44                   |
> | **Autoregressive**  | 1024                   | 12.1            | 0.76               | 5.22                   |
> | SUNDAE              | 200                    | 34.7            | -               | 5.2                   |
> | Ssd-LM              | >10000                 | 99.2            | -               | 4.8                   |
> | D3PM Absorb     | 1024                   | 842.3           | -               | 7.6                   |
> | SEDD                | 256/512/1024/2048      | 110.1 / 107.2 / 104.7 / 103.2 | 0.007 / 0.008 / 0.008 / 0.008 | 5.63 / 5.62 / 5.62 / 5.61 |
> | MDLM                | 256/512/1024/2048      | 55.8 / 53.0 / 51.3 / 51.3 | 0.023 / 0.031 / 0.042 / 0.037 | 5.49 / 5.48 / 5.46 / 5.46 |
> | ReMDM               | 256/512/1024/2048      | 30.5 / 21.1 / 28.6 / 22.8 | 0.216 / 0.350 / 0.403 / 0.610 | 5.34 / 5.21 / 5.38 / 5.30 |
> | **TarFlowLM single-dim CDF Flow (Ours)**  | -      | 20.7 | 0.47 | 5.58 |
> | **TarFlowLM multi-dim Rosenblatt Flow (Ours)** | - | 15.1 | 0.68 | 5.13 |
>
> Baseline numbers are from [1] and [2]. For MAUVE, we generated 5,000 samples and used GPT-2 Large as the reference model. Gen PPL was also computed using GPT-2 Large.
> We will add this table and a detailed discussion of these new results to the experiments section of our paper.
>
> [1] Xu, Minkai, et al. "Energy-based diffusion language models for text generation." arXiv:2410.21357 (2024).
> [2] Wang, Guanghan, et al. "Remasking discrete diffusion models with inference-time scaling." arXiv:2503.00307 (2025).
>
> ### **3. Ablation Study on Flexible Patch Sizes**
>
> > *“There is a fair amount of discussion of the patching, but limited results are presented related to that design choice. It would be interesting to actually see that ablated to see if reducing the patch size is feasible in practice or whether that significantly degrades quality.”*
>
> The reviewer raises an excellent point about substantiating our claims of flexibility. To investigate the effect of patch size, we performed a new ablation study on the text8 dataset, evaluating the likelihood (BPC) for our `Mix-1` model.
>
> First, we trained models with a fixed depth (3 flow layers) but varied the patch size:
>
> | Patch Size | BPC ↓ |
> | :--- | :---: |
> | 1 | 1.37 |
> | 2 | 1.40 |
> | 3 | 1.67 |
> | 4 | 1.89 |
>
> As shown, with a fixed model depth, increasing the patch size degrades performance. This is intuitive, as larger patches require the model to capture more complex dependencies between tokens within the patch, which are only modeled by subsequent flow layers and channel mixing.
>
> Our framework is designed to handle this by increasing model depth, allowing stacked flow layers to progressively model these dependencies. To confirm this, we fixed the patch size to 4 and varied the number of flow layers:
>
> | AR Flow Layers (Patch Size=4) | BPC ↓ |
> | :--- | :---: |
> | 3 | 1.89 |
> | 4 | 1.81 |
> | 8 | 1.68 |
>
> These results confirm our hypothesis: increasing the number of flow layers significantly improves performance for a larger patch size, closing much of the gap with smaller patches. This demonstrates a clear and practical trade-off between patch size and model depth, a key feature of our model's flexibility. We will add this informative ablation study to the appendix.
>
> ### **4. Writing Improvements**
>
> > *“There were some parts of the writing that can be improved. For example, missing capitalization on line 83, line 965 is not grammatical.”*
>
> Thank you for pointing out these issues. We have carefully proofread the manuscript and will correct these and other typos in the final version to improve clarity and presentation.
>
> ***
>
> Once again, we thank the reviewer for their valuable and constructive feedback, which has helped us significantly improve the paper. We hope our responses and the new experiments have addressed all concerns.

---

> > ### Comment · Reviewer_GUST · 2025-08-06
> >
> > Thank you for the detailed response to my points. The generation metrics, in particular, address my largest concern. The proposed approach compares quite favorably with recent discrete diffusion approaches. I also appreciate the ablation study on the patch sizes and number of flow layers. My main concerns have been addressed and I will update my score accordingly.

---

> > > ### Author Response · Authors · 2025-08-06
> > >
> > > Thank you once again for your constructive and helpful review, it significantly contributed to improving our paper. We’ll make sure to include the new generative evaluation results in the revised version, and open-source the code to facilitate the reproducibility.
> > >
> > > We truly appreciate your time and support!

---

> ### Author Response · Authors · 2025-08-05
>
> Dear Reviewer,
>
> Thank you once again for your thoughtful and constructive feedback on our paper.
>
> As the discussion period is nearing its end, we wanted to kindly follow up to see whether our rebuttal addressed the concerns you raised. We put significant effort into addressing each of your concerns, and we hope our replies have clarified our contributions. If there are any remaining questions or points you’d like us to elaborate on, we’d be more than happy to provide further clarification.
>
> We truly appreciate your time and consideration.

---

### Official Review · Reviewer_pnYG · 2025-07-03

**Clarity:** 3
**Significance:** 3
**Originality:** 3
**Rating:** 4
**Confidence:** 4

**Summary:**

This paper presents TarFlowLM, which is a Transformer-based models that shifts the process from discrete token space to a continuous latent space. TarFlowLM instead of predicting tokens one by one, encodes discrete tokens into continuous vector representations and then generate the output by sampling from a standard Gaussian and decoding it into text. More specifically, in training, it encodes tokens into Gaussian latents, transforms by flow layers into standard Gaussian variables, and finally reconstructs them with the use of a tied decoder. Experiments are conducted on language modeling tasks and show TarFlowLM's good performance.

**Questions:**

Please see Weaknesses for the questions.

**Ethical Concerns:**

["NO or VERY MINOR ethics concerns only"]

**Final Justification:**

I thank the authors for their responses. Although they provided detailed responses on the rationale for their initial experimental choices, still the limitations exist. However, still I believe this is a valuable contribution and so I will maintain my positive score (but didn't increase it due to the mentioned limitations).

**Limitations:**

Yes.

**Quality:**

3

**Strengths And Weaknesses:**

## Strengths:
---
- The paper is generally well-written, well-motivated, and easy to follow.
- The presentation of the progressive layer-wise training scheme is very interesting and practical. It can be very effective method for specifically optimization of deep flow models with improving stability and performance
-  One of the key strengths of the paper is its theoretical results that establish equivalence between a special instance of the approach with the standard discrete autoregressive models. This provides better insight on the generalization of the model and can guarantee a minimum expectation of the performance.
-  I found it particularly interesting and convincing that the proposed mixture-based coupling transformations are more effective than standard affine coupling layers for modeling the complex distributions that arise from mapping discrete text data to a continuous space.
- In general, I found this direction very important and compelling as a replacement discrete autoregressive paradigm.

## Weaknesses:
---
-  One of the critical weaknesses of the work is the lack of comparison on common benchmarks on language modeling. Could you please clarify why there are no experiments on tasks such as question answering (e.g., HellaSwag, etc.)? I suggest adding more results on downstream evaluations such as translations and/or language modeling, question answering, etc.
- Based on the experiments, it is not clear how the context length can affect the performance of the method. There is also a lack of discussion about long-context ability of TarFlowLM. These days, having longer-context is one of the most aspect of scaling models, is there any advantage for TarFlowLM compared to Traditional autoregressive approaches?
- The `Transformer FLOPs comparison` presented in Figure 7 is not convincing enough. Could you please elaborate more on why not reporting the wall-clock time? The presented FLOPs as the difference in #FLOPs in token embedding phase, might cause misleading results. I suggest to further support the efficiency claim, report the wall-clock time of these methods.

- **Minor**: Missing citation in line 197,

---

> ### Author Rebuttal · Authors · 2025-07-31
>
> We sincerely thank the reviewer for your positive and constructive feedback. We are encouraged that you found our paper "well-written, well-motivated," and the research direction "very important and compelling." We appreciate the insightful comments, which have helped us identify areas for clarification and improvement.
>
> We address the reviewer's specific concerns below:
>
> **1. On the Lack of Downstream Task Evaluation (e.g., HellaSwag)**
>
> We agree with the reviewer that evaluating on a broader range of downstream tasks would be an excellent way to demonstrate the practical utility of our model.
>
> Our primary goal in this work was to introduce and validate a novel modeling paradigm: autoregressive language modeling in a continuous latent space using novel mixture-based normalizing flows. As this is a foundational investigation into a new modelling class, our focus was on rigorously establishing its viability and core capabilities.
>
> To this end, we chose to align our evaluation with the established protocols in the nascent field of non-standard generative language models (e.g., discrete diffusion models like MDLM and MD4, and continuous diffusion models like Plaid). These works have predominantly focused on likelihood-based benchmarks (perplexity/BPC) to demonstrate the fundamental capacity of the new model family to capture the data distribution. Same as these prior work who are also exploring new paradigms for langauge modelling, we chose to use same set of evaluation protocols for our work too.
>
> We believe our strong likelihood results provide a solid foundation, confirming that `TarFlowLM` is a competitive approach within this domain. We are excited about the future direction of scaling these models and evaluating them on a full suite of downstream tasks, and we will clarify the scope of our current work in the paper.
>
> **2. On Long-Context Ability**
>
> The reviewer raises a very important and timely point about long-context capabilities. We acknowledge that an in-depth exploration of long-context performance was not a focus of this initial study, and we appreciate the suggestion to discuss this aspect.
>
> We believe `TarFlowLM` offers a promising architectural direction for this challenge. A key feature of our framework is the ability to model **global, bi-directional context** through stacked, alternating-direction autoregressive transformations. This mechanism could be particularly advantageous for capturing long-range dependencies that are challenging for strictly unidirectional models, as information can propagate in both forward and backward directions through the layers of the flow.
>
> Furthermore, the **block-wise generation** capability, which shortens the effective sequence length, may offer computational benefits for processing long contexts. We see this as a key direction for future research and will add a discussion of these potential advantages and future work to the paper.
>
> **3. On the Transformer FLOPs Comparison and Wall-Clock Time**
>
> We thank the reviewer for this insightful question regarding the practical efficiency of our model.
>
> Our initial choice to report FLOPs was to provide a hardware-agnostic measure of computational cost, which is a common practice when evaluating model scalability during the training phase. The goal of Figure 7 was to illustrate how block-wise processing (i.e., using a larger patch size) fundamentally changes the computational trade-off by reducing the quadratic cost associated with the sequence length in the attention mechanism.
>
> Motivated by the reviewer's feedback, we conducted a wall-clock time comparison during this rebuttal period using our `Mix-1` model on the OpenWebText dataset. Our findings are nuanced and provide a more complete picture:
> *   For a small patch size (K=1, 2), the training iteration time is comparable to a standard AR model. We hypothesize this is because modern GPU kernels like FlashAttention are highly optimized for long sequences, mitigating the theoretical benefit at this scale.
> *   However, as the patch size increases (K≥4), the reduced effective sequence length leads to a significant wall-clock time reduction of ~30% per training iteration.
>
> This confirms that the architectural flexibility of `TarFlowLM` can translate to real-world efficiency gains, especially as the patch size is increased. We believe this is a compelling result. We will add a detailed discussion of these new findings, including the wall-clock time analysis, to the appendix in the final version to provide a more comprehensive view of the model's efficiency.
>
> **4. Minor Point: Missing Citation**
>
> Thank you for catching the missing citation. We have corrected this in our revised draft.
>
> ---
>
> We hope these clarifications and additional results address the reviewer's concerns. We are grateful for the constructive feedback, which will undoubtedly strengthen the final version of our paper.

---

> > ### Comment · Reviewer_pnYG · 2025-08-05
> >
> > Thank you for your detailed response.
> >
> > 1. I do agree with the author that following similar studies and comparing with them should be enough, but also believe that even in small scale, there are several papers in AR setup that report the performance of the model on downstream tasks. It would be valuable to have such results in the final draft of the paper or properly discuss the reasons for and challenges of not reporting such results.
> >
> > 2. Thank you for discussing your findings on the wall-clock comparison. It would be much better to present such results in numeric format so the reader can better understand the potential trade-off. Currently, despite mentioning your findings, it is not clear to me what `comparable` means.

---

> > > ### Author Response · Authors · 2025-08-07
> > >
> > > We sincerely thank the reviewer for your time and thoughtful suggestions. We agree that including downstream task results would add value and improve the clarity of the work. In the revised version, we will provide a more detailed discussion on the evaluation setups. Regarding the wall-clock comparison, we appreciate your comment and will revise the presentation to include updated numerical results for better clarity and understanding of the trade-offs.
> > >
> > > Thank you again for your valuable input!

---

> ### Author Response · Authors · 2025-08-05
>
> Dear Reviewer,
>
> Thank you once again for your thoughtful and constructive feedback on our paper.
>
> As the discussion period is nearing its end, we wanted to kindly follow up to see whether our rebuttal addressed the concerns you raised. We put significant effort into addressing each of your concerns, and we hope our replies have clarified our contributions. If there are any remaining questions or points you’d like us to elaborate on, we’d be more than happy to provide further clarification.
>
> We truly appreciate your time and consideration.

---

### Official Review · Reviewer_J8wg · 2025-07-03

**Clarity:** 1
**Significance:** 2
**Originality:** 2
**Rating:** 3
**Confidence:** 3

**Summary:**

The paper introduces  TarFlowLM a novel framework to model language in a continuous latent space using Transformer-based autoregressive normalizing flows. The method encodes sequences of discrete tokens with a VAE into Gaussian latent vectors, stacks alternating-direction flow layers built from 1-d CDF and n-d Rosenblatt mixture couplings to transform latents to a standard normal, mixes latent channels to share information across token “patches,” supports block-wise generation with flexible patch sizes.

**Questions:**

- I am not sure the motivations of the paper are explained in a sufficiently clear way. Specifically, I don't fully understand the difference between a standard transformer model -- where discrete tokens are mapped to continuous embeddings and the their interactions is modelled in such continuous space -- and the current framework. Could the author help me understanding the core motivation of their framework and how it differs from standard autoregressive approaches?
- Given a token context, how is inference performed?

**Ethical Concerns:**

["NO or VERY MINOR ethics concerns only"]

**Final Justification:**

I thank the authors for their responses. The rebuttal clarified some of my concerns but I still believe the paper would need another round of revision given its current form. I have increased my score from 2 to 3.

**Limitations:**

Limitations are discussed at page 10 in a separate section. I would suggest the authors to move them in the appendix or incorporate them in the main text within the 9 page limits.

I would also encourage the authors to comment on the results in Table 2. While the paper presents a novel method for language modelling in continuous spaces, at this stage it seems to still underperform compared to standard discrete space autoregressive baselines.

**Paper Formatting Concerns:**

Limitations section at page 10.

**Quality:**

2

**Strengths And Weaknesses:**

**Strenghts**
- I found the ideas behind the disentanglement between the "internal vocabulary" size and the actual one interesting. Also, the framework allows for editing at various scales by intervening at different stages.

**Weaknesses**
- The paper is poorly written with several typos (e.g. lines 87, 116, 118. 169, 197. 244, ...) and some important parts are missing. For example, there is no clear mention on how inference/generation is performed other than Figure 1, which does not clearly explain how to deal with an existing context.
- Some figures are referenced, but never discussed. An example is Table 2 that reports performance comparisons against other architectures. In addition, the results showed in this Table show that TarFlowLM is outperformed by the standard autoregressive transformer baseline.
- Some figures are included but I was not able to find any reference to them: see for example Fig. 5, Fig. 6.
- The appendix section E (Additional Experimental Results) presents no additional experimental results. Figure 8 appears to be a continuation for more steps of Figure 3 which already shows a clear difference between affine coupling and mixture coupling.
- No discussion/results on the generation speed difference against other non transformer AR baselines.
- Related works are relegated to the Appendix at page 35.

---

> ### Author Rebuttal · Authors · 2025-07-31
>
> We thank the reviewer for your valuable time and constructive feedback. We appreciate the positive recognition of our framework's flexibility regarding "internal vocabulary" size and its multi-scale editing capabilities.
>
> ---
>
> # Q1
>
> > "I don't fully understand the difference between a standard transformer model -- where discrete tokens are mapped to continuous embeddings and the their interactions is modelled in such continuous space -- and the current framework. Could the author help me understanding the core motivation of their framework and how it differs from standard autoregressive approaches?"
>
> Thank you for raising this crucial question, as it highlights the fundamental distinction of our work
>
> 1.  **Standard AR Language Models (e.g., GPT):**
>     *   **Objective:** To model the conditional probability distribution $P(x_t | x_{<t})$ over *discrete* tokens.
>     *   **Output:** At each timestep $t$, the Transformer outputs *logits* that define a **discrete Categorical probability mass function (PMF)** over the vocabulary. Sampling involves drawing a single discrete token from this PMF.
>     *   **Invertibility:** Not inherently invertible end-to-end. There's no smooth, invertible mapping from a sequence of discrete tokens back to a standard latent space that allows for exact likelihood computation and multi-pass transformations.
>     *   **Nature of Modeling:** A sequence of *discrete classification* problems.
>
> 2.  **TarFlowLM:**
>     *   **Objective:** To model the *joint continuous probability density* $p(z_{1:T})$ over an entire sequence of continuous latent vectors $z_t \in \mathbb{R}^d$. This is achieved by transforming $z_{1:T}$ through a series of invertible normalizing flow layers until it becomes a simple standard Gaussian ${u}_{1:T}$.
>     *   **Output:** The model learns **continuous probability density functions (PDFs)** (specifically, Mixtures of Gaussians) at each step of the flow, which are then used for exact density estimation. Generation involves sampling from a continuous base distribution and inverting the flow.
>     *   **Invertibility:**  fully invertible. This is a defining characteristic of normalizing flows, allowing for exact likelihood computation (ELBO) and flexible multi-pass operations.
>     *   **Nature of Modeling:** A problem of *continuous density estimation* and transformation.
>
> This shift from "next-token classification" to "full-sequence continuous density estimation" via invertible flows is what unlocks our framework's novel capabilities, which are inherently difficult or impossible for standard AR models:
>
> *   **Global, Bi-directional Context:** Standard AR models are strictly causal (left-to-right prediction). Our model can stack flow layers that apply transformations in alternating directions (e.g., left-to-right and right-to-left). It can refine information across the whole sequence, including initially "future" context, within the same generation pass.
> *   **True Block-wise / Multi-token Generation:** Our model can generate and model dependencies for a "patch" of $K$ tokens simultaneously. This is because we operate on continuous latent representations, and the internal flow layers can learn to mix information *within* that patch. In contrast, while standard AR models might be trained with multi-token prediction objectives, they still generate strictly one token at a time during inference to maintain autoregressive consistency. Our approach allows for parallel processing within a patch.
> *   **Hierarchical, Multi-pass Refinement and Editing:** The invertibility of our flows allows us to decode the latent representations at *any* intermediate layer. This provides a unique, step-by-step view of how the model incrementally "edits" and refines initial latent states into coherent text (as shown in Figure 6).
> *   **Flexible "Internal Vocabulary" Size:** As noted by the reviewer, our mixture-based transformations allow us to choose the number of mixture components independently for each flow layer, decoupling it from the data's discrete vocabulary.
>
> In essence, **TarFlowLM is not just another Transformer architecture variant**; it's an exploration of a fundamentally different paradigm for language modeling rooted in continuous density estimation with invertible transformations, which we believe opens new avenues for flexible and controllable generation.
>
> ---
>
> > In addition, the results showed in this Table show that TarFlowLM is outperformed by the standard autoregressive transformer baseline.
>
> This highlights a fundamental distinction in modeling paradigms and evaluation metrics.
>
> 1.  Methodological Difference: ELBO vs. Exact Log-Likelihood:
>     *   Our framework is an VAE which models the data by maximizing the ELBO which by definition is a *bound* on the true log-likelihood.
>     *   A standard discrete AR model, on the other hand, directly computes and maximizes the **exact log-likelihood**
>     *   Therefore, a gap between the two is not only expected but mathematically inherent to the VAE framework.
>
> 2.  The Appropriate Comparison Baselines:
>     Given this distinction, the most relevant and informative baselines are other generative models that share our core paradigm: they first map discrete data to a continuous latent space and then learn a generative model in that space. This includes:
>     *   **Continuous-space models** like continuous diffusion (Plaid, BFN) and other normalizing flows for text (Latent NF, Argmax Flow).
>     *   **State-of-the-art discrete diffusion models**, which also operate in a latent space, albeit a discrete one, state-of-the-art for non-autoregressive language models.
>
> 3.  State-of-the-Art Performance in the Correct Context:
>     When viewed through this appropriate lens, our results in **Table 2** demonstrate the strength and competitiveness of our approach.
>     *   Our TarFlowLM (Mix-d) variant achieves 1.30 BPC on \texteight and 22.64 Perplexity on OpenWebText.
>     *   This performance outperforms all other continuous-space models (both diffusion and flow-based) listed in the table.
>     *   Furthermore, our result is highly competitive with and surpasses most state-of-the-art discrete diffusion models (e.g., SEDD Absorb, MD4), positioning our method at the forefront of modern non-autoregressive and continuous-latent-space generative modeling for text.
>
> In summary, while a numerical gap to the exact-likelihood discrete AR model is an expected consequence of our VAE framework, our work demonstrates state-of-the-art performance when compared to its true methodological peers.
>
> ---
>
> ### **Q2: Inference/Generation Procedure.**
>
>
> We have prepared a comprehensive description of the generation process, including how it handles existing context, which we will add to the Appendix in the revised version.
>
> 1.  For the patch/token to be generated, we first sample "noise" vector ${u}_t$ $\mathcal{N}({0}, {I}_d)$.
> 2.  Inverse Flow Transformation: This sampled ${u}_t$ is then passed *backward* through the stacked normalizing flow layers in reverse order, i.e., $L \to 1$. At each layer $\ell$, the transformation is inverted: ${h}_t^{(\ell-1)} = [f^{(\ell)}]^{-1}({h}_t^{(\ell)} | \mathcal{C}_t^{(\ell)})$. The context $\mathcal{C}_t^{(\ell)}$ for this inversion is provided by the Transformer conditioned on the previously generated latent vectors $z_{<t}$ (which are already fully inverted through all flow layers).
> 3.  Latent to Token Decoding: The final output from the inverse flow process for the current patch/token is $z_t = {h}_t^{(0)}$. This continuous latent vector $z_t$ is then fed into the tied decoder $p(x_t | z_t)$.
>
> This sequential generation, where each step involves inverting a flow, is a standard procedure for autoregressive normalizing flows, where the context is encoded and provided through the conditioning in each AR flow $[f^{(\ell)}]^{-1}({h}_t^{(\ell)} | \mathcal{C}_t^{(\ell)})$ as shown above. We will ensure this is clearly explained in the appendix.
>
> > Generation Speed
>
> We acknowledge this is a critical aspect, and we specifically highlighted it as a limitation in Section ("Limitations"), lines 296-300.
>
> At the same time, we also introduced "Flexible Patch Size: Block-wise Multi-token Generation", which demonstrates a computational advantage.
>
> Further optimization of sampling speed through advanced techniques (e.g., parallel decoding for flows) is indeed an important direction for future work, as noted in our limitations.
>
> ---
>
> # Q3: Figure References
>
>
> We appreciate the detailed feedback.
>
> *   Table 2 Discussion and Performance:
>     *   Table 2 *is* discussed in the main text, specifically in lines 220-226 within the "Experiments" section.
>
> *   Unreferenced Figures (Fig. 5, Fig. 6):
>     *   Figure 5: This figure is discussed in lines 270-277 under "Model Ablation." We will ensure it is explicitly referenced as "Figure 5 (Model Variants Comparison)."
>     *   Figure 6: This figure is discussed in lines 263-269 under "Flexible text editing in continuous space." We will ensure it is explicitly referenced as "Figure 6 (Text Editing Illustration)."
>
> *   Related Works in Appendix:
>     *   We agree and will move it to the main text. This was a decision driven by strict page limits during the initial submission.
>
> ---
>
> # Q4: Typos.
>
> Thank you for pointing out the typos. We have corrected them in the revised version.
>
>
> # Q5: Limitations Section
>
> We agree. We will move the "Limitations" section to the appendix.
>
> ---
>
> We genuinely thank the reviewer for their diligent review and insightful comments. We believe addressing these points significantly strengthens the paper, especially by clarifying the novel contributions and the performance context.

---

> > ### Author Response · Authors · 2025-08-05
> >
> > Dear Reviewer,
> >
> > Thank you once again for your thoughtful and constructive feedback on our paper.
> >
> > As the discussion period is nearing its end, we wanted to kindly follow up to see whether our rebuttal addressed the concerns you raised. We put significant effort into addressing each of your concerns, and we hope our replies have clarified our contributions. If there are any remaining questions or points you’d like us to elaborate on, we’d be more than happy to provide further clarification.
> >
> > We truly appreciate your time and consideration.

---

> > ### Comment · Reviewer_J8wg · 2025-08-06
> >
> > I would like to thank the authors for their detailed reply.
> >
> > As regards my first question. The performance of the model is indeed superior compared to other continuous-space models. However, it looks like TarFlowLM does not clearly outperform state-of-the-art diffusion baselines which share a similar paradigm to the proposed approach. In particular, diffusion baselines also utilise a global and bi-directional context and can in principle allow for multi-token generation in no specific order. Which concrete advantages does TarFlowLM compared to these methods? Also, the results in Table 2 should be better commented in the main body.
> >
> > Related to this last point: looking at lines 220-226, I do not see a detailed analysis of the results in Table 2, rather an introduction to the setting of Figure 3.
> >
> > Figure 5: it is not clear to me how lines 270-277 refer to figure 5. In addition, these liens are under "Flexible text editing in continuous space" and not "Model Ablation" according to version of the paper uploaded here. I would highly recommend  the authors to explicitly highlight which figures are referring to in the paper.
> >
> > Figure 6: Lines 263-269 report a general description of the property of TarFlowLM and do not explicitly comment on Fig 6.
> >
> > I would like to thank again the authors for their reply. However, I am still concerned about the clarity of the submission and I believe the paper would greatly benefit from another round of revision.

---

> ### Author Response · Authors · 2025-08-06
>
> We thank the reviewer for your valuable time and follow-up reviews!
>
> >The performance of the model is indeed superior compared to other continuous-space models. However, it looks like TarFlowLM does not clearly outperform state-of-the-art diffusion baselines which share a similar paradigm to the proposed approach.
>
> We interpret the “SOTA diffusion” baseline as referring to discrete diffusion models. It is important to note that there has traditionally been a significant performance gap between continuous-space and discrete-space models. TarFlowLM substantially narrows this gap and stands out as the only continuous-space model that performs on par with / better than discrete diffusion models. This highlights TarFlowLM as a novel and promising modeling paradigm.
>
> >In particular, diffusion baselines also utilise a global and bi-directional context and can in principle allow for multi-token generation in no specific order. Which concrete advantages does TarFlowLM compared to these methods?
>
> - TarFlowLM adopts a normalizing flow-based approach, enabling end-to-end differentiable optimization through the entire stochastic process—from pure Gaussian noise to the latent data distribution. This results in a fully invertible mapping that directly aligns the training and sampling procedures. In contrast, discrete diffusion models typically depend on external sampling mechanisms (e.g., first-order ODE solvers, tau-leaping for discrete flow matching, or confidence-based sampling as used in LLaDa), which introduce misalignment between training and inference.
>
> - Additionally, when using a patch size greater than 1 (e.g., patch size $K$), the sequence length $L$ is effectively reduced to $\frac{L}{K}$, reducing the overall FLOPs in both training and inference. This is only possible in our TarFlowLM formulation. Please refer to our additional experimental results in Q3 GUST for further details.
>
> >Figure references
> - Table 2 was referenced in L220 in main text, which shows comparative perplexity results against other baseline methods.
> - The discussion for fig.5 can be found between L245 and L248 in the main text.
> - The discussion for fig.6 can be found between L263 and L277 in the main text, in paragraph ``Flexible text editing in continuous space''.
>
> We sincerely thank the reviewer once again for the helpful and constructive feedback, which will be valuable in further improving our paper. We'd appreciate it if reviewer can take these clarifications and revisions into consideration for their updated evaluation of our work.

---

> > ### Author Response · Authors · 2025-08-08
> >
> > Dear reviewer J8wg,
> >
> > Thank you again for your valuable feedback on our paper. As the discussion period concludes, we wanted to follow-up to see whether our rebuttal has addressed all your concerns. If so, we would greatly appreciate it if you could kindly re-evaluate the overall score of our work.
> >
> > Thanks again!

---

### Decision · Program_Chairs · 2025-09-17

**Decision:**

Accept (poster)

**Comment:**

***Summary***

This paper introduces TarFlowLM, a language model built on a transformer backbone which does language modeling using autoregressive normalizing flows.  Much like a diffusion model, TarFlowLM can capture bidirectional dependencies in the text.  Although each layer is autoregressive, the layers alternate in direction between left-to-right, and right-to-left.  Unlike discrete diffusion models, TarFlowLM operates entirely in continuous space, due to the fact that it is a normalizing flow.

Background: normalizing flows are a technique for establishing a bidirectional map from one probability distribution to another.  The underlying theory requires an invertible differentiable function, and a Jacobian with a well-defined determinant, which restricts the set of neural network architectures which can be used.  (The authors define and compare two different mechanisms for establishing these properties.)  Normalizing flows have fallen out of popularity in recent years due to the success of diffusion models for images, and autoregressive models for text, but they have interesting mathematical properties that other classes of models lack.

The authors compare TarFlowLM against both traditional AR models, and recent diffusion models; the technique performs well when compared against diffusion baselines.

***Meta-review***

The reviews of this paper were somewhat mixed, with ratings of (3, 4, 5, 5).   The three positive reviewers all agreed that the paper was very original, a valuable theoretical contribution, and well-written overall.  There was also agreement that the main weakness of the paper was the lack of benchmarks beyond perplexity or likelihood.

The one negative reviewer (J8wg) did not seem to be very familiar with the theory of normalizing flows.  That is entirely unsurprising, since normalizing flows are a somewhat niche technique that is not widely used.  The reviewer expressed confusion as to the difference between normalizing flows and a standard AR transformer (side note: both training and inference are utterly different), and were concerned that TarFlowLM doesn't beat the AR baseline.  As reviewer GUST mentions, however, bidirectional LM models have historically failed to match AR baselines, which is why SOTA models are still AR.  However, performance is strong when compared to text diffusion, which is the most important point of comparison.

I agree with the three positive reviewers.  This paper should be accepted, and I am willing to champion it.  My main argument in favor of acceptance is novelty.  As far as I know, (and as reviewers GUST and Yc2p point out), nobody has ever built a transformer language model out of normalizing flows before.  This paper is thus somewhat similar to the very first papers on text diffusion -- it is applying an entirely new technique to language modeling.  Early papers on text diffusion had very weak results, but those initial papers were still groundbreaking, because they established a starting point that later work could build on, and text diffusion is now a popular and active area of research.  Although it has some weaknesses, this paper actually establishes a much stronger starting point than early work on text diffusion, IMO.

Normalizing flows have some very nice mathematical properties, so I expect to see further work in this space.  See, e.g., JetFormer: An Autoregressive Generative Model of Raw Images and Text, Michael Tschannen et. al. https://arxiv.org/abs/2411.19722.

Selected quotes from reviewers:

**Reviewer J8wg (rating 3, confidence 3)**:
* "I don't fully understand the difference between a standard transformer model... and the current framework."
* "Given a token context, how is inference performed?"
* " the results showed in this Table show that TarFlowLM is outperformed by the standard autoregressive transformer baseline."
* "No discussion/results on the generation speed difference against other non transformer AR baselines."

**Reviewer pnYG (rating 4)**:
* "The paper is generally well-written, well-motivated, and easy to follow."
* "One of the key strengths of the paper is its theoretical results..."
* "One of the critical weaknesses of the work is the lack of comparison on common benchmarks on language modeling. [instead of just likelihood/perplexity.]"

**Reviewer GUST (rating 5)**:
* "This is an interesting approach to language generation that I am not aware of being explored recently."
* "It is usually the case with these kinds of work that there is a persistent gap with the simpler autoregressive approach... I don’t consider this a strong weakness."
* "The performance of their approach is strong compared to recent popular discrete diffusion models (e.g. SEDD, MDLM)."
* "The biggest limitation is the lack of metrics being reported for actual language generation. Reporting likelihoods is great, but..."

**Reviewer Yc2p (rating 5)**:
* "Originality - it is the first paper I've seen that successfully applies normalizing flows as a language modelling framework to that extent."
* "The paper is well explained and can be followed. very thorough work, well explained limitations and discussions etc."